# Genome-wide identification and phenotypic characterization of seizure-associated copy number variations in 741,075 individuals

Ludovica Montanucci[1,170], David Lewis-Smith [2,3,4,5,170], Ryan L. Collins [6,7,170], Lisa-Marie Niestroj[8,170], Shridhar Parthasarathy[4,5], Julie Xian[4,5], Shiva Ganesan[4,5], Marie Macnee[8], Tobias Brünger[8], Rhys H. Thomas [2,3], Michael Talkowski[6,7], Epi25 Collaborative*, Ingo Helbig[4,5,9], Costin Leu [1,10,11,12,171] ✉ & Dennis Lal [1,7,11,12,171] ✉

Copy number variants (CNV) are established risk factors for neurodevelopmental disorders with seizures or epilepsy. With the hypothesis that seizure disorders share genetic risk factors, we pooled CNV data from 10,590 individuals with seizure disorders, 16,109 individuals with clinically validated epilepsy, and 492,324 population controls and identified 25 genome-wide significant loci, 22 of which are novel for seizure disorders, such as deletions at 1p36.33, 1q44, 2p21-p16.3, 3q29, 8p23.3-p23.2, 9p24.3, 10q26.3, 15q11.2, 15q12-q13.1, 16p12.2, 17q21.31, duplications at 2q13, 9q34.3, 16p13.3, 17q12, 19p13.3, 20q13.33, and reciprocal CNVs at 16p11.2, and 22q11.21. Using genetic data from additional 248,751 individuals with 23 neuropsychiatric phenotypes, we explored the pleiotropy of these 25 loci. Finally, in a subset of individuals with epilepsy and detailed clinical data available, we performed phenome-wide association analyses between individual CNVs and clinical annotations categorized through the Human Phenotype Ontology (HPO). For six CNVs, we identified 19 significant associations with specific HPO terms and generated, for all CNVs, phenotype signatures across 17 clinical categories relevant for epileptologists. This is the most comprehensive investigation of CNVs in epilepsy and related seizure disorders, with potential implications for clinical practice.

An epileptic seizure is a paroxysm of symptoms and signs due to abnormally excessive or synchronous neuronal activity[1]. Seizures are classified based on their characteristics and electroencephalogram (EEG) as focal-onset seizures (which start in a specific brain region) and generalized-onset seizures (which are rapidly seen across bihemispheric networks)[1,2]. The utility of this seizure classification is that it categorizes epilepsy into syndromes and allows clinicians to make implications about disease etiology, trajectory, and response to medication. Clinical manifestations vary from whole-body convulsions with loss of consciousness (tonic-clonic seizures), to movements involving only part of the body with variable levels of consciousness (focal motor seizure), to a brief loss of awareness (absence seizure)[1,2]. Seizures can be provoked by head trauma, infection, or acute toxic-metabolic imbalance, or they can be spontaneous and unprovoked. Individuals who exhibit at least one unprovoked seizure with an enduring elevated risk of further seizures or who have the electroclinical features of one of a few specific epilepsy syndromes that can be diagnosed without recurrent seizures fulfill the criteria for a diagnosis of epilepsy[1].

A full list of affiliations appears at the end of the paper. *A List of authors and their affiliations appears at the end of the paper.
✉e-mail: costin.leu@uth.tmc.edu; Dennis.Lal@uth.tmc.edu

Seizures and epilepsy are common in the general population. Neonatal seizures occur in 1.5% of neonates, febrile seizures in 2–4% of young children, and epilepsy in up to 1% of children and adolescents[3]. Seizures are common among individuals with neurodevelopmental disorders, affecting 21.5% of those with autism and intellectual disability and 8% with autism without intellectual disability[4].

Copy number variants (CNVs), such as deletions and duplications, change the dosage of genomic segments and are established risk factors for various types of epilepsy[5–14], seizures[15], and neuropsychiatric disorders[16–19]. Large CNVs can affect multiple dosage-sensitive genes, leading to complex clinical presentations. To date, only one hypothesis-free genome-wide CNV association study (CNV-GWAS) has been reported for epilepsy[20]. This CNV-GWAS in 10,712 individuals with epilepsy and 6,746 controls identified three genome-wide significant CNVs[20]. High-resolution CNV screening has become routine in clinical molecular diagnostics, leading to greater detection of chromosomal abnormalities in patients[21]. Diagnostic CNVs can be identified in 1–4% of individuals with epilepsy and >10% of those with seizures and neurodevelopmental disorders[13,20–22]. However, the pleiotropy of pathogenic CNVs, partially driven by structural properties (size, fixed vs. variable breakpoints, number of affected genes), represents a significant challenge in the clinical interpretation of CNVs, limiting their utility for disorder classification, prognostication, and the development of precision medicine treatments that specifically target the critical pathogenic gene(s) altered by the CNV. The majority of pathogenic and likely pathogenic CNVs are greater than 1 megabase (Mb) in size, and it is often unclear which gene(s) or genomic element(s) affected by the CNV contribute to one or more disorders[23,24]. A well-powered seizure CNV discovery screen combined with detailed genotype-phenotype analyses could identify genomic segments that confer risk for seizures, identify clinical characteristics in affected patients and consequently guide genetic test interpretation.

Although many individuals with neuropsychiatric and developmental disorders have comorbid seizures, genome-wide CNV association analyses across epilepsy and seizure have yet to be reported. We hypothesized that genetic risk for seizures is shared in individuals with epilepsy diagnosed according to International League Against Epilepsy (ILAE) criteria[1] and related neurological and neurodevelopmental disorders who also have seizures. Therefore, a joint analysis could add to the three epilepsy-associated CNV loci reported previously[20]. To explore this hypothesis, we performed a meta-analysis of GWAS studies comprising 26,699 individuals with diagnosed epilepsy or seizures and 492,324 controls. Since both definitions are based on the presence of seizures, we refer to individuals affected by either condition as individuals with seizures from here on forward. The effective sample size of this study ($N_{eff}$ = 101,302) provides adequate power to identify significant associations of risk CNVs that are present in the general healthy population, therefore, do not exhibit complete penetrance. However, the analytic setup restricts the frequency in the general population to up to 1% for quality purposes. We assessed the pleiotropy of any identified seizure-associated CNV in subsequent meta-analyses of epilepsy and 238,161 independent individuals affected by a range of 23 neuropsychiatric disorders. Finally, using a subset of the seizure cohort comprising 10,880 individuals with epilepsy detailed using 214,203 Human Phenotype Ontology (HPO) annotations[25], we evaluated the clinical features characterizing carriers of each seizure-associated CNV.

## Results

### Discovery of 25 genome-wide significant seizure-associated CNVs regions

We performed a meta-analysis of 16,109 individuals with epilepsy and 8545 population controls (the Epi25 Collaborative cohort) with 10,590 individuals with seizures (not explicitly meeting diagnostic criteria for epilepsy) and 483,779 population controls, derived from an aggregated CNV dataset of 17 cohorts (neuropsychiatric disorders cohort) (see all cohorts of this study in Supplementary Table 1). The genome was scanned using 267,237 genomic segments of 200 kb size in a 10 kb sliding window approach[26]. After applying Bonferroni correction of the threshold for a significant association in the meta-analysis and fine-mapping, we identified 25 loci associated with seizures at genome-wide significance ($P \le 3.74 \times 10^{-6}$). All 25 loci are shown in Fig. 1 and detailed in Table 1. The 25 identified loci included 15 deletion CNVs (size range: 230 kb to 5 Mb) and ten duplication CNVs (size range:

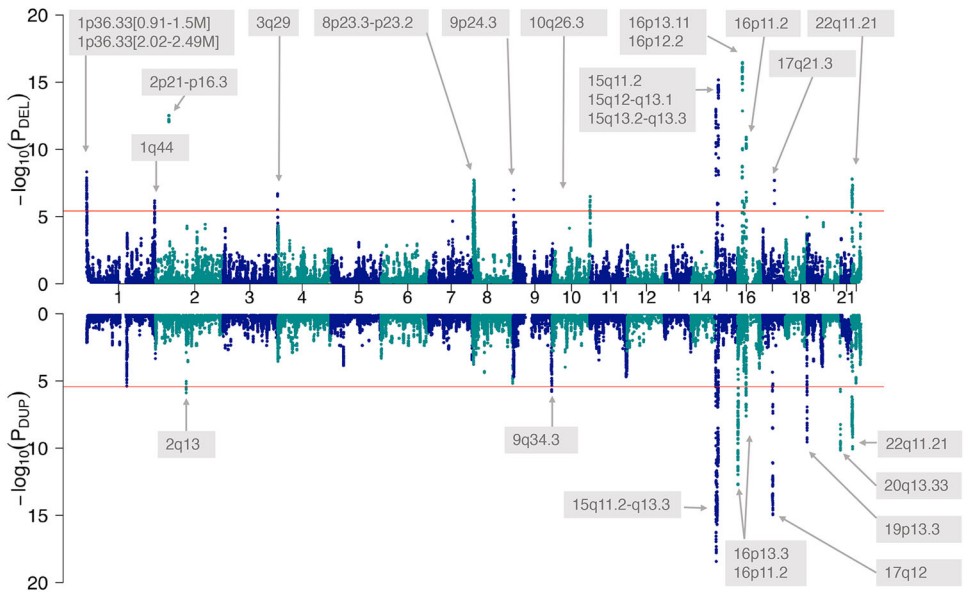

**Fig. 1 | Genome-wide meta-analysis identifies 25 CNVs associated with seizure disorders.** Miami plot of the meta-analysis of the CNV genome-wide association analyses of (1) 16,109 individuals with clinically validated epilepsy vs. 8545 controls and (2) 10,590 individuals with seizure disorders vs. 483,779 controls. Dots represent -log$_{10}$ of the meta-analysis P-values (P$_{DEL}$ and P$_{DUP}$ for deletions and duplications, respectively) of the cohort-specific Fisher exact tests for the enrichment of CNVs in cases vs. controls for each a 200 kb sliding window. Genomic regions that surpassed the Bonferroni-corrected threshold for significance (red line, α = 3.74 × 10$^{-6}$) were annotated with the genomic band containing the signal. Deletions (top) and duplications (mirrored) are shown.

**Table 1 | Genome-wide significantly associated CNV regions and credible intervals**

| Cytoband | CNV type | Hg19 Start (Mb) | Hg19 End (Mb) | Lowest P-value in region | OR [95% CI] | Credible interval containing the causal element/gene with 95% confidence | Associations with neuropsychiatric phenotypes [N] | Highest odds ratio in neuropsychiatric disorder/seizure meta-analyses (95% CI) |
|---|---|---|---|---|---|---|---|---|
| 1p36.33 | DEL | 0.91 | 1.51 | 4.65E-09 | 23 (10-54) | 910000-1510000 | 5 | 11 (5-24) CNS abnormality |
| 1p36.33 | DEL | 2.02 | 2.49 | 1.95E-07 | 44 (14-141) | 2020000-2490000 | 3 | 48 (14-170) Abnormalities of Cognition |
| 1q44 | DEL | 245.29 | 245.86 | 6.59E-07 | 41 (13-133) | 245290000-245860000 | 6 | 62 (19-207) Abnormal brain morphology |
| 2p21-p16.3 | DEL | 47.5 | 47.85 | 3.00E-13 | 12 (7-22) | 47500000-47850000 | 23 | 13 (8-24) Behavioral abnormalities |
| 2q13 | DUP | 110.77 | 111.06 | 1.33E-06 | 3 (2-5) | 110770000-111060000 | 6 | 24 (9-63) Hyperactivity |
| 3q29 | DEL | 195.76 | 196.24 | 2.01E-07 | 40 (13-122) | 195760000-196240000 | 0 | no significant association signal |
| 8p23.3-p23.2 | DEL | 0.4 | 5.47 | 1.83E-08 | 12 (6-25) | 400000-610000 3040000-3780000 4810000-5470000 | 4 | 16 (7-39) Intellectual disability |
| 9p24.3 | DEL | 0.33 | 0.56 | 1.08E-07 | 13 (6-29) | 330000-560000 | 1 | 7 (4-12) Neurodevelopmental abnormality |
| 9q34.3 | DUP | 139.21 | 140.12 | 1.67E-06 | 12 (5-27) | 139210000-139590000 139890000-140120000 | 21 | 12 (6-25) Schizophrenia |
| 10q26.3 | DEL | 133.41 | 134.68 | 3.16E-07 | 40 (13-125) | 133410000-133740000 134370000-134680000 | 2 | 32 (10-107) Sleep disorder |
| 15q11.2 | DEL | 22.74 | 23.28 | 1.02E-13 | 3 (2-3) | 22740000-23280000 | 12 | 3 (2-4) Abnormalities of Cognition |
| 15q11.2-q13.3 | DUP | 22.98 | 32.15 | 3.68E-19 | 27 (14-52) | 24750000-25080000 | 23 | 33 (16-67) Intellectual disability |
| 15q12-q13.1 | DEL | 27.93 | 28.23 | 9.85E-07 | 14 (6-34) | 27930000-28230000 | 3 | 24 (11-52) Intellectual disability |
| 15q13.2-q13.3 | DEL | 31.06 | 32.51 | 6.71E-16 | 14 (8-24) | 31060000-32510000 | 17 | 16 (6-44) Sleep disorder |
| 16p13.3 | DUP | 0.6 | 0.89 | 1.98E-13 | 9 (5-14) | 600000-890000 | 23 | 12 (6-26) Schizophrenia |
| 16p13.11 | DEL | 15.42 | 16.35 | 3.53E-17 | 9 (6-14) | 15420000-16350000 | 18 | 13 (4-44) CNS atrophy |
| 16p12.2 | DEL | 21.88 | 22.5 | 1.14E-06 | 4 (2-5) | 21880000-22500000 | 6 | 4 (2-6) Hyperactivity |
| 16p11.2 | DEL | 29.56 | 30.19 | 1.25E-11 | 9 (5-15) | 29560000-30190000 | 11 | 13 (8-21) Intellectual disability |
| 16p11.2 | DUP | 29.87 | 30.19 | 2.45E-08 | 6 (3-10) | 29870000-30190000 | 12 | 11 (5-24) Sleep disorder |
| 17q12 | DUP | 34.76 | 36.25 | 1.12E-15 | 15 (9-27) | 34760000-35510000 35960000-36250000 | 14 | 10 (5-20) Abnormal brain morphology |
| 17q21.31 | DEL | 41.08 | 41.45 | 1.98E-08 | 5 (3-9) | 41080000-41450000 | 20 | 6 (4-10) Abnormality of the nervous system |
| 19p13.3 | DUP | 1.04 | 1.34 | 2.85E-10 | 7 (4-12) | 1040000-1340000 | 23 | 7 (4-14) Schizophrenia |
| 20q13.33 | DUP | 62 | 62.35 | 7.12E-11 | 6 (4-10) | 62000000-62350000 | 23 | 8 (5-13) CNS abnormality |
| 22q11.21 | DUP | 18.99 | 21.54 | 8.73E-11 | 5 (3-7) | 18990000-1937000 20200000-21540000 | 9 | 5 (3-7) Hyperactivity |
| 22q11.21 | DEL | 18.99 | 21.54 | 1.57E-08 | 26 (10-65) | 18990000-19400000 19670000-19960000 20650000-21540000 | 12 | 26 (9-70) Central motor dysfunction |

Column 1: Cytoband localization of the CNV. Column 2: CNV type, either deletion (DEL, white background row) or duplication (DUP, grey background row). Columns 3 and 4: Genomic coordinates (in Mb) on the GRCh37 reference genome of the start and end position of the merged CNV region that is supported by genome-wide association signals. Columns 5 and 6: Lowest P-values in each CNV region and corresponding odds ratios (OR) (with 95% confidence interval) of the genome-wide CNV meta-analysis in 25,345 individuals with seizures and 492,324 controls. Column 7: GRCh37 coordinates of the credible interval(s) that contained the causal element(s) with 95% confidence. Column 8: Number of neuropsychiatric disorders that also show a significant genome-wide CNV-association in this locus. Column 9: Highest odds ratio for each locus in any of the 23 cross-disorder meta-analyses.

290 kb to 8.9 Mb). All the genome-wide associated deletions found in this study consisted of the loss of one copy, while all duplications consisted of the gain of one copy. Three of the 25 seizure-associated loci (15q11.2-q13.3 dup, 15q13.2-q13.3 del, 16p13.11 del) had previous genome-wide statistical support for an association with epilepsy from our previous study[20] that included 40% of the individuals with seizures of this study. All other identified CNVs (22/25, 88%) represent new genome-wide significant loci for seizures, with 10/22 (59%) loci previously implicated in neurological and psychiatric disorders, 6/22 (23%) specifically in epilepsy by studies without genome-wide statistical support, 2/22 (9%) reported in individuals without neurological or psychiatric disorders, and 4/22 (18%) not previously reported regions. We detailed in Table 2 all commonly reported disease phenotypes for the 25 identified seizure-associated loci. Our meta-analysis in seizure disorders was likely not powered enough to identify some of the known CNVs implicated in epilepsy (without genome-wide statistical support) associated with seizures (e.g., 1q21.1 del/dup). Reciprocal CNVs, defined by deletions and duplications associated with seizures involving overlapping genomic segments, were found at 15q11.2, 16p11.2, and 22q11.21. No overlap existed between the seizure-associated CNV regions identified in this study and the most recent SNP-based GWAS study in epilepsy[27].

**Fine-mapping and candidate genes**

Out of the three CNV regions with previous genome-wide statistical support, our fine-mapping approach narrowed down the critical

seizure-relevant region for the known 15q11-q13 duplication to the imprinted promoter/exon 1 region of SNPRN (Table 2, Supplementary Fig. 1). The SNRPN promoter/exon 1 region was suggested to regulate the imprinting of the critical region for Prader-Willi syndrome[28,29]. Overexpression of SNRPN, corresponding to the seizure-associated duplication of the region, was found to cause abnormal neural development in cultured primary cortical neurons[30]. Conversely, SNRPN knockdown was found in the same study to also cause subtle neuronal abnormalities, in line with reports of short SNRPN deletions in Prader-Willi syndrome[31]. For the other two CNV regions with previous genome-wide statistical support, we identified several genes with a brain phenotype in the minimal credible intervals. The 15q13.2-q13.3 deletion credible interval includes the haploinsufficient gene OTUD7A, shown to cause abnormal development of cortical dendritic spines and dendrite outgrowth in Otud7a[DEL/+] mice[32], and KLF13, shown to cause a layer-specific decrease of cortical interneurons in Klf13[DEL/+] mice[33]. The 16p13.11 deletion credible interval includes two haploinsufficient genes: MYH11, implicated in cerebrovascular disorders[34,35] that are a risk factor for seizures[36], and MARF1, involved in cortical neurogenesis[37].

Out of the six seizure-associated CNV regions previously implicated in epilepsy without genome-wide statistical support, we mapped the credible intervals of the two seizure-associated deletions at 1p36 to the first and third known critical regions for seizures within the phenotype spectrum of the 1p36 deletion syndrome[38]. Known disease genes in the credible intervals at 1p36 are DVL1 (Robinow syndrome[39]), TMEM240 (Spinocerebellar ataxia 21[40]), and SKI (Shprintzen-Goldberg

**Table 2 | Known disease genes in the credible intervals of the seizure-associated CNV regions**

| Cytoband | CNV type | Best overlapping syndrome | Credible interval containing the causal element/gene with 95% confidence | Brain-related disease genes (high confidence) | PMID |
|---|---|---|---|---|---|
| 15q11.2-q13.3 | DUP | 15q11-q13 duplication syndrome (Prader-Willi/Angelman critical region) | 15:24750000-25080000 | *SNRPN* overexpression (Neurodevelopmental phenotype) | 27430727 |
| | | | | *SNRPN* deletion (Prader-Willi) if the CNV is gene disrupting | 35956251 |
| 15q13.2-q13.3 | DEL | 15q13.3 deletion syndrome | 15:31060000-32510000 | - | - |
| 16p13.11 | DEL | 16p13.11 deletion syndrome | 16:15420000-16350000 | *MYH11* (Moyamoya-like cerebrovascular disease, cerebral artery aneurysm) | 29263223, 27367753 |
| 1p36.33 | DEL | 1p36 deletion syndrome (Seizures critical region 1) | 1:910000-1510000 | *DVL1* (Robinow syndrome) | 25817016 |
| | | | | *TMEM240* (Spinocerebellar ataxia 21) | 25070513 |
| 1p36.33 | DEL | 1p36 deletion syndrome (Seizures critical region 3) | 1:2020000-2490000 | *SKI* (Shprintzen-Goldberg syndrome) | 23023332 |
| 1q44 | DEL | *KIF26B* deletion | 1:245290000-245860000 | *KIF26B* (Pontocerebellar hypoplasia) | 30151950 |
| 16p12.2 | DEL | 16p12.1 deletion syndrome | 16:21880000-22500000 | - | - |
| 16p11.2 | DEL | 16p11.2 deletion syndrome (BP4-BP5) | 16:29560000-30190000 | *PRRT2* (Benign familial infantile seizures) | 33746883 |
| | | | | *TAOK2* (Autism spectrum disorder) | 29467497 |
| 17q12 | DUP | 17q12 duplication syndrome | 17:34760000-35510000 | - | - |
| | | | 17:35960000-36250000 | - | - |
| 2q13 | DUP | NPHP1 duplication | 2:110770000-111060000 | *NPHP1* duplication (Autism spectrum disorder, global developmental delay) | 25126106, 16892302 |
| 3q29 | DEL | 3q29 deletion syndrome | 3:195760000-196240000 | - | - |
| 8p23.3-p23.2 | DEL | 8p23.2-pter deletion syndrome | 8:400000-610000 | - | - |
| | | | 8:3040000-3780000 | - | - |
| | | | 8:4810000-5470000 | - | - |
| 9p24.3 | DEL | 9p24.3 *DOCK8 / KANK1* deletion | 9:330000-560000 | *KANK1* (Cerebral palsy spastic quadriplegic 2) | 16301218 |
| 9q34.3 | DUP | interstitial 9q34.3 duplication (not encompassing *EHMT1*) | 9:139210000-139590000 | - | - |
| | | | 9:139890000-140120000 | *GRIN1* gain of function (Polymicrogyria) | 29365063 |
| 10q26.3 | DEL | 10q26 deletion syndrome | 10:133410000-133740000 | - | - |
| | | | 10:134370000-134680000 | - | - |
| 15q11.2 | DEL | 15q11.2 deletion syndrome (BP1-BP2) | 15:22740000-23280000 | *NIPA1* (Spastic paraplegia 6, autosomal dominant) | 23897027 |
| 16p11.2 | DUP | 16p11.2 duplication syndrome (BP4-BP5) | 16:29870000-30190000 | - | - |
| 22q11.21 | DUP | 22q11.21 deletion syndrome (LCRA-LCRD) | 22:18990000-19370000 | - | - |
| | | | 22:20200000-21540000 | - | - |
| 22q11.21 | DEL | 22q11.21 deletion syndrome (LCRA-LCRD) | 22:18990000-19400000 | - | - |
| | | | 22:19670000-19960000 | *TBX1* (22q11.21 deletion syndrome) | 14585638 |
| | | | 22:20650000-21540000 | *LZTR1* (Noonan syndrome) | 30368668 |
| 2p21-p16.3 | DEL | Lynch syndrome locus | 2:47500000-47850000 | - | - |
| 19p13.3 | DUP | non-canonical 19p13.3 duplication | 19:1040000-1340000 | - | - |
| 15q12-q13.1 | DEL | *OCA2* deletion | 15:27930000-28230000 | - | - |
| 16p13.3 | DUP | non-canonical 16p13.3 duplication | 16:600000-890000 | *STUB1* gain of function (early onset dementia syndrome, autosomal dominant ataxia with cognitive decline and autism) | 35493319, 32211513 |
| 17q21.31 | DEL | non-canonical 17q21.31 deletion | 17:41080000-41450000 | *BRCA1* (Cancer) | 35393462 |
| 20q13.33 | DUP | novel 20q13.33 duplication | 20:62000000-62350000 | *KCNQ2* gain of function (Neurodevelopmental disability, neonatal encephalopathy) | 35780567, 28139826 |
| | | | | *EEF1A2* gain of function (Neurodevelopmental disorders) | 32160274 |

Highlighted are: (1) Darkest grey: three CNV regions with previous genome-wide statistical support for epilepsy (PMID: 32568404), (2) Medium-dark grey: six CNV regions previously implicated in epilepsy without genome-wide statistical support, (3) Medium-light grey: ten CNV regions previously reported in other neurological and psychiatric disorders, and (4) Light grey: four novel CNV regions never reported in neurological or psychiatric disorders. In the second column, DEL and DUP indicate deletions and duplications, respectively. Gene names are formatted in italic.

syndrome[41]). In the credible intervals of the remaining CNV regions, we identified the following known disease genes: (i) the haploinsufficient *KIF26B* gene (Pontocerebellar hypoplasia[42]) as the only gene affected by the 1q44 deletion, and (ii) *PRRT2* (self-limited familial infantile epilepsy, paroxysmal dyskinesia[43]) and the haploinsufficient *TAOK2* gene (Autism[44]) at the 16p11.2 BP4-BP5 deletion syndrome locus. Of note, single nucleotide variants in *PRRT2* are among the most frequent findings in clinical genetic testing of epilepsy[45].

Among the ten seizure-associated CNV regions previously reported in other neurological and psychiatric disorders, we identified one credible interval suggesting a different causal gene than previously reported: an interstitial 9q34.3 duplication not encompassing *EHMT1* that is considered as the causal gene based on one out of 22 reported 9q34.3 duplication carrier[46]. The top candidate gene within the credible interval identified by our meta-analysis is *GRIN1*, affected by 9q34.3 duplications in 21 of all reported carriers[46]. *GRIN1* gain of

function variants are known to cause a developmental epileptic encephalopathy, often with polymicrogyria[47]. In contrast, our fine-mapping analysis confirms *TBX1* as the (known) causal gene for the 22q11.21 deletion/DiGeorge syndrome[48]. We also found *LZTR1* (Noonan syndrome[49]) within the credible 22q11.21 deletion intervals. Other known disease genes in the credible intervals of the remaining CNV regions implicated in neurological and psychiatric disorders were: *NPHP1* inside a 2q13 duplication (Autism and global developmental delay[50,51]), *KANK1* (Cerebral palsy spastic quadriplegic 2[52]) inside a small 9p24.3 *DOCK8/KANK1* deletion, and *NIPA1* (Autosomal dominant spastic paraplegia 6[53]) inside the 15q11.2 BP1-BP2 deletion syndrome region.

Finally, we identified four novel CNV regions associated with seizures. Three out of four harbored known disease genes. The credible region of a non-canonical 16p13.3 duplication included *STUB1*. *STUB1* gain of function was reported to cause early onset dementia syndrome[54] and autosomal dominant ataxia with cognitive decline and autism[55]. The credible region of a non-canonical 17q21.31 deletion included *BRCA1*. *BRCA1* mutations are well-known in cancer[56], with *BRCA1* as a possible mediator of glioma cell proliferation, migration, and glioma stem cell self-renewal[57]. The credible region of a novel 20q13.33 duplication included *KCNQ2* and *EEF1A2*. *KCNQ2* gain of function is known to cause neurodevelopmental disability and neonatal encephalopathy[58,59]. *EEF1A2* gain of function was shown to cause neurodevelopmental disorders, including epilepsy and intellectual disability[60].

Significantly enriched Gene ontology (GO) Biological Processes among all known brain-related disease genes in the credible intervals were: chordate embryonic development (GO:0043009 [http://amigo.geneontology.org/amigo/search/ontology?q=GO%3A0043009&searchtype=ontology]), sensory organ morphogenesis (GO:0090596 [http://amigo.geneontology.org/amigo/search/ontology?q=GO%3A0090596&searchtype=ontology]), mitotic G2 DNA damage checkpoint signaling (GO:0007095 [http://amigo.geneontology.org/amigo/search/ontology?q=GO%3A0007095&searchtype=ontology]), neural tube closure (GO:0001843 [http://amigo.geneontology.org/amigo/search/ontology?q=GO%3A0001843&searchtype=ontology]), negative regulation of Ras protein signal transduction (GO:0046580 [http://amigo.geneontology.org/amigo/search/ontology?q=GO%3A0046580&searchtype=ontology]), dendrite morphogenesis (GO:0048813 [http://amigo.geneontology.org/amigo/search/ontology?q=GO%3A0048813&searchtype=ontology]), and mitotic G2/M transition checkpoint (GO:0044818 [http://amigo.geneontology.org/amigo/search/ontology?q=GO%3A0044818&searchtype=ontology]). No GO Biological Process was significantly enriched when considering all genes inside all credible intervals, pointing to likely heterogeneous disease mechanisms of the 25 seizure-associated CNV regions. All credible intervals and known brain-related disease genes are detailed in Table 2, additional candidate genes of lower confidence are detailed in Supplementary Data 1, and all genes inside the credible intervals are detailed in Supplementary Data 2.

## Most of the 25 identified risk CNVs are pleiotropic

We performed 23 meta-analyses of epilepsy with 23 other neuropsychiatric disorders (listed in Supplementary Table 2) in an additional 238,161 individuals with neuropsychiatric disorders and 492,324 controls to explore pleiotropy of the 25 identified CNVs. 24 out of 25 seizure-associated CNVs were significantly associated in at least one of the 23 meta-analyses with a neuropsychiatric disorder. The number of neuropsychiatric disorders with which a significant association was found and their greatest odds ratios are reported in Table 1. About two thirds (60%) of all CNVs were highly pleiotropic and showed significant associations with >10 epilepsy/neuropsychiatric disorder meta-analyses. The most frequently co-associated phenotype was

"Neurodevelopmental abnormality" (HP:0012759 [https://hpo.jax.org/app/browse/term/HP:0012759]); associated with 36% of all seizure-associated CNVs).

## Characterization of the clinical subphenotypes enriched in the carriers of each seizure-associated CNV in epilepsy patients with deep phenotypes

We performed phenome-wide association analyses for each of the 33 credible intervals identified across the 25 CNV regions to characterize the high-resolution clinical manifestations associated with each CNV. This analysis was performed on a subset of the Epi25 Collaborative cohort (Phenomic cohort, Supplementary Table 1) comprising 10,880 individuals with non-acquired epilepsy and deep phenotypic data (the clinical presentation of this cohort of 10,880 individuals and the frequencies of selected common and characteristic epilepsy phenotypes are provided in Supplementary Table 3). In the Phenomic cohort, 562 individuals (5.2%) carried at least one seizure-associated credible interval ($N = 498$ / 4.6% carried one credible interval, $N = 64$ / 0.6% carried 2–5 credible intervals). The most common credible interval (deletion at 2p21-p16.3) was carried by 114 (1.0%) individuals, and 18 credible intervals were found in at least 0.1% of the cohort (≥11 carriers). One CNV was not found (deletion at 9p24.3, containing a single credible interval). Across the 32 detected credible intervals and 1667 annotated HPO concepts, we identified 622 nominally significant associations (two-sided Fisher's exact test, Supplementary Data 3). Given the large number of associations tested and that HPO annotations describing the same clinical feature at different levels of precision are highly correlated, we applied the minP step-down procedure to aid interpretation[61], yielding 19 associations robust to multiple testing within each genetically defined group (minP-adjusted $P < 0.05$, Table 3, Figs. 2, 3, and Supplementary Fig. 2A–E).

Carriers of deletions at 1p36.33 [0.91–1.51 Mb] ($N = 25$, 0.23% of the Phenomic cohort), 1p36.33 [2.02–2.49 Mb] ($N = 17$, 0.16%), or 15q12-q13.1 ($N = 4$, 0.037%), and carriers of duplications at 15q11.2-q13.3 ($N = 46$, 0.42%) were enriched with clinical features suggestive of developmental and epileptic encephalopathies, such as epileptic spasms and tonic seizures, epileptic encephalopathy, and other neurodevelopmental disorders, sudden unexpected death in epilepsy, and morphological abnormalities[62]. Features characterizing genetic generalized epilepsy were associated with deletions at 2p21-p16.3 ($N = 114$, 1.05%, generalized tonic-clonic and absence seizures), 15q11.2 ($N = 56$, 0.52%, eyelid myoclonia and absence seizures), 16p13.11 ($N = 42$, 0.39%, generalized tonic-clonic seizures), 15q13.2-q13.3 ($N = 24$, 0.22%, absence seizures) or 22q11.21 [20.65–21.54 Mb] ($N = 6$, 0.055%, juvenile myoclonic epilepsy-like features). Duplications at 16p11.2 ($N = 8$, 0.074%) were associated with non-epileptic seizures comorbid with epilepsy (OR = 81.5, unadjusted $P = 4.82 \times 10^{-4}$, minP-adjusted $P = 0.0297$), and showed a nonsignificant greater frequency of microcephaly (OR = 31.5, unadjusted $P = 3.62 \times 10^{-2}$, minP-adjusted $P = 0.92$) that replicates the mirror microcephaly/macrocephaly phenotype of the reciprocal 16p11.2 CNVs[63].

We interrogated the phenotypic annotations of CNV carriers regarding the candidate genes prioritized in our fine-mapping analysis. *MSH2* was prioritized as the candidate gene for the most common deletion in the Phenomic cohort (2p21-p16.3). Heterozygous loss of function variants of the haploinsufficient gene *MSH2* cause Lynch syndrome 1[64], and complete knockout of paralog *Msh2* in *Ccm1*+/- mice causes multiple cavernoma through a presumed second hit[65]. We found that carriers had a nonsignificant greater frequency of neoplasms (OR = 2.35, unadjusted $P = 2.49 \times 10^{-2}$, minP-adjusted $P = 1.00$) and cerebral cavernomata (OR = 5.23, unadjusted $P = 6.58 \times 10^{-4}$, minP-adjusted $P = 0.157$) than non-carriers. Carriers of the 1p36.33 [2.02–2.49 Mb] deletion overlapping the gene *SKI* had features (hypotonia, talipes equinovarus, abnormalities of the globe and nose, osteoporosis, global developmental delay, and Chiari malformation)

**Table 3 | Significant individual CNV-HPO associations**

| Locus | CNV type | HPO | Odds ratio [95% CI] | Relative risk | P-value | | CNV carriers | | | CNV non-carriers | |
|---|---|---|---|---|---|---|---|---|---|---|---|
| | | | | | Raw | Adjusted | Prop | $N_{pheno}$ | $N_{tot}$ | $N_{pheno}$ | $N_{tot}$ |
| 15q13.2-q13.3 [31.06–32.51 Mb] | DEL | Generalized non-motor (absence) seizure [HP:0002121] | 10.5 [4.25–28.5] | 4.18 | 3.70E–08 | 1.00E–05 | 0.667 | 16 | 24 | 1731 | 10,856 |
| 15q13.2-q13.3 [31.06–32.51 Mb] | DEL | Typical absence seizure [HP:0011147] | 8.43 [3.48–21.3] | 4.1 | 6.94E–07 | 1.10E–04 | 0.583 | 14 | 24 | 1545 | 10,856 |
| 15q13.2-q13.3 [31.06–32.51 Mb] | DEL | EEG with spike-wave complexes [HP:0010850] | 7.84 [3.16–21.2] | 3.28 | 1.18E–06 | 2.00E–04 | 0.667 | 16 | 24 | 2205 | 10,856 |
| 15q13.2-q13.3 [31.06–32.51 Mb] | DEL | Generalized-onset seizure [HP:0002197] | 9.41 [3.15–37.9] | 2.4 | 1.41E–06 | 2.20E–04 | 0.833 | 20 | 24 | 3766 | 10,856 |
| 15q13.2-q13.3 [31.06–32.51 Mb] | DEL | EEG with generalized epileptiform discharges [HP:0011198] | 6.76 [2.44–23.2] | 2.2 | 1.98E–05 | 0.00379 | 0.792 | 19 | 24 | 3905 | 10,856 |
| 15q13.2-q13.3 [31.06–32.51 Mb] | DEL | Bilateral tonic-clonic seizure with focal onset [HP:0007334] | 0 [0–0.404] | 0 | 4.07E–04 | 0.0484 | 0 | 0 | 24 | 3168 | 10,856 |
| 1p36.33 [0.91–1.51 Mb] | DEL | Hypotonia [HP:0001252] | 12.2 [3.95–32] | 9.51 | 3.23E–05 | 0.00674 | 0.24 | 6 | 25 | 274 | 10,855 |
| 1p36.33 [0.91–1.51 Mb] | DEL | Epileptic spasm [HP:0011097] | 7.47 [2.78–18.4] | 5.4 | 6.85E–05 | 0.0108 | 0.32 | 8 | 25 | 643 | 10,855 |
| 1p36.33 [0.91–1.51 Mb] | DEL | Abnormal muscle tone [HP:0003808] | 8.65 [2.81–22.7] | 6.82 | 1.97E–04 | 0.0287 | 0.24 | 6 | 25 | 382 | 10,855 |
| 1p36.33 [0.91–1.51 Mb] | DEL | Infantile spasms [HP:0012469] | 8.34 [2.71–21.9] | 6.58 | 2.39E–04 | 0.0324 | 0.24 | 6 | 25 | 396 | 10,855 |
| 1p36.33 [0.91–1.51 Mb] | DEL | Abnormal muscle physiology [HP:0011804] | 8.21 [2.67–21.5] | 6.48 | 2.59E–04 | 0.0339 | 0.24 | 6 | 25 | 402 | 10,855 |
| 1p36.33 [0.91–1.51 Mb] | DEL | Abnormality of the musculature [HP:0003011] | 8.04 [2.61–21.1] | 6.35 | 2.87E–04 | 0.038 | 0.24 | 6 | 25 | 410 | 10,855 |
| 1p36.33 [0.91–1.51 Mb] | DEL | Plagiocephaly [HP:0001357] | 93.8 [9.48–482] | 86.8 | 3.30E–04 | 0.045 | 0.08 | 2 | 25 | 10 | 10,855 |
| 2p21-p16.3 [47.50–47.85 Mb] | DEL | Focal-onset seizure [HP:0007359] | 0.463 [0.313–0.681] | 0.708 | 4.79E–05 | 0.0086 | 0.456 | 52 | 114 | 6939 | 10,766 |
| 2p21-p16.3 [47.50–47.85 Mb] | DEL | Bilateral tonic-clonic seizure with generalized onset [HP:0025190] | 2.3 [1.5–3.46] | 1.88 | 9.09E–05 | 0.0157 | 0.325 | 37 | 114 | 1861 | 10,766 |
| 15q12-q13.1 [27.93–28.23 Mb] | DEL | Global developmental delay [HP:0001263] | 69.1 [5.55–3540] | 18.1 | 2.80E–04 | 0.0127 | 0.75 | 3 | 4 | 451 | 10,876 |
| 15q12-q13.1 [27.93–28.23 Mb] | DEL | Epileptic encephalopathy [HP:0200134] | Inf [4.43-Inf] | 7.72 | 2.83E–04 | 0.0127 | 1 | 4 | 4 | 1408 | 10,876 |
| 15q12-q13.1 [27.93–28.23 Mb] | DEL | Encephalopathy [HP:0001298] | Inf [4.41-Inf] | 7.69 | 2.87E–04 | 0.0129 | 1 | 4 | 4 | 1414 | 10,876 |
| 16p11.2 [29.87–30.19 Mb] | DUP | Psychogenic non-epileptic seizure [HP:0033052] | 81.5 [7.85–471] | 61.8 | 4.82E–04 | 0.0297 | 0.25 | 2 | 8 | 44 | 10,872 |

In the first column, the genomic band and coordinates of the considered CNV are reported. The CNV type is reported in column 2. In column 3, the HPO term name and identifier are reported. In column 4, the odds ratio with unadjusted two-sided 95% confidence interval is reported. In column 5, the relative risk is given to aid interpretation. In column 6, the unadjusted two-sided $P$-values from Fisher's exact test are reported. In column 7, the minP step-down $P$-value is given, which provides an adjustment for all 1,667 HPO term associations tested within each CNV group, while accounting for the correlation between harmonized HPO annotations (see Online Methods). In column 8, the proportion of CNV carriers annotated with the phenotype is given. In columns 9–10 and 11–12, $N_{pheno}$ and $N_{tot}$ are the number of individuals annotated with the phenotype and the total number of individuals carrying and not-carrying the CNV, respectively.

concordant to the Shprintzen-Goldberg craniosynostosis syndrome caused by $SKI$[41]. All 15 individuals with duplication of 9q34.3 had focal-onset seizures that were rarely drug-resistant, without any individual annotated with a neurodevelopmental disorder or polymicrogyria despite the presence of the $GRIN1$, which can cause polymicrogyria when affected by gain-of-function variants[47]. Sixteen of 24 individuals carrying deletions at 15q13.3 [31.06–32.51 Mb] had generalized absence seizures (OR = 10.5, unadjusted $P = 3.70 \times 10^{-8}$, minP-adjusted $P = 1 \times 10^{-5}$), in line with the primary seizure type reported in carriers of the 15q13.3 deletion[66]. Finding generalized myoclonic seizures in half of the carriers of the 22q11.2 [19.67–19.96 Mb] deletion further confirmed $TBX1$[67], the known causal gene for the 22q11.21 deletion/DiGeorge syndrome[48]. Features suggestive of juvenile myoclonic

epilepsy were also found among six people carrying deletions overlapping with the second credible interval at 22q11.2 [20.65–21.54 Mb] spanning the Noonan syndrome 10 locus containing in which a single individual was reported with seizures[49]. However, none of these six individuals had annotations beyond seizures and electro-encephalography phenotypes that would support a multisystemic syndrome.

Finally, clinicians may want to know the frequency of broad clinical features among carriers of the CNV identified in their patients to improve the interpretation of its clinical relevance and to facilitate genetically stratified prognostication. Therefore, we prioritized 17 common, conceptually broad, and important epilepsy manifestations and comorbidities for visualization, including the co-occurrence of

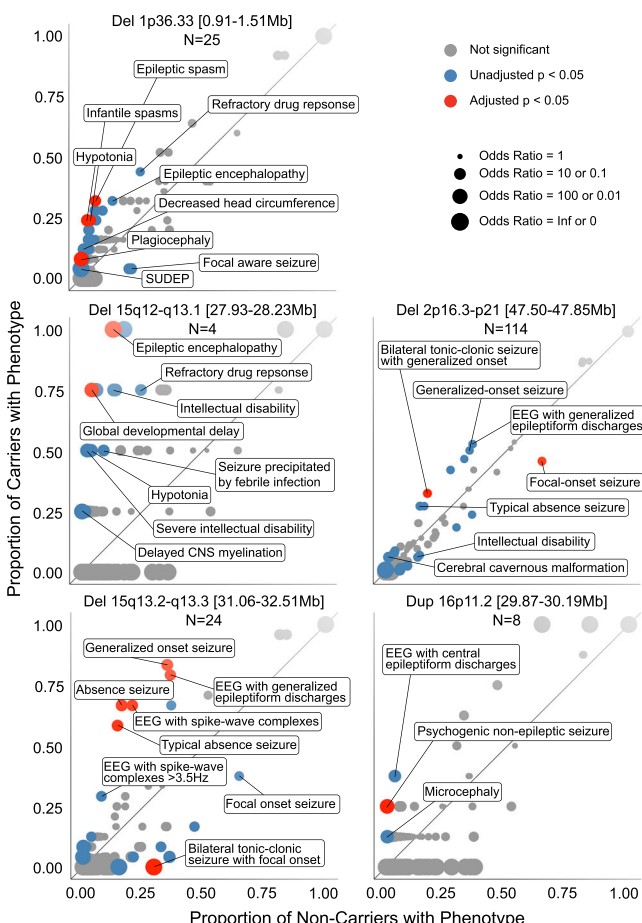

**Fig. 2 | Genotype-first phenomic analysis in 10,880 individuals with detailed clinical data.** For each CNV, the proportion of carriers and non-carriers annotated with each HPO concept is plotted. Those above the diagonal were enriched among carriers, and those below were depleted. Odds ratios are represented by dot size. The selected phenotypes labeled were prioritized according to statistical evidence and clinical breadth. Full results for all associations reaching unadjusted $P < 0.05$ are provided in Supplementary Data 3. SUDEP sudden unexpected death in epilepsy, CNS central nervous system, EEG electroencephalogram.

generalized-onset and focal-onset seizures that characterizes the combined generalized and focal epilepsy type[62] (Fig. 3 and Supplementary Fig. 3A–E). The most common CNV, deletion at 2p21-p16.3, appeared to modestly increase the likelihood of a carrier having generalized epilepsy. However, a few CNVs had a profile dominated by core electroclinical features of generalized (for example, deletions at 15q13.2–15q13.3) or focal epilepsy (duplications at 9q34.3 [139.89–140.12 Mb]), with comorbid features being rare. Conversely, carriers of other CNVs had relatively high frequencies of neurodevelopmental disorders, epileptic spasms, and drug resistance suggestive of developmental and epileptic encephalopathy (deletions at 1p36.33). However, no CNV was found exclusively in people with a particular seizure type, and carriers of some CNVs appeared to have broad clinical features at frequencies indistinguishable from the cohort's baseline (duplications at 19p13.3), suggesting some generic contribution to epilepsy risk across epilepsy types.

## Discussion

In this study, we leveraged a substantial increase in sample size to identify novel seizure-associated CNVs when jointly analyzing 26,699 individuals with various types of seizure disorders against 492,324 population controls. We identified 25 novel loci with genome-wide significance for seizure disorders. In addition, all three previously

reported epilepsy-associated loci at genome-wide level maintained genome-wide significance for seizure disorders in our meta-analysis that included the epilepsy cohort from the previous study[20]. Of the 25 seizure-associated loci, 16 were previously implicated in neurological and psychiatric disorders, including epilepsy. Five were flanked by known segmental duplications (SDs) or low copy number repeats (LCRs). Of note, our fine-mapping analysis confirmed the first and third known critical regions for seizures within the phenotype spectrum of the 1p36 deletion syndrome[38], *TBX1* as the (known) causal gene for the 22q11.21 deletion/DiGeorge syndrome[48], and suggested the *SNRPN* promoter/exon 1 region as the causal element for seizures within the larger BP2-BP3 15q11.2-q13 duplication region. However, our study design did not support the assessment of whether the imprinting status of the duplicated region itself plays an additional role besides the previously suggested role of *SNRPN* promoter/exon 1 region in regulating the imprinting of the Prader-Willi critical region. Future studies that also include genomic screens of parents will shed light on this open question.

In a high-resolution phenomic analysis in a subset of 10,880 individuals from our cohort with epilepsy (from the Epi25 cohort), we identified 622 suggestive and 19 significant clinical associations informative for epileptologists among CNV carriers. This observation indicates that beyond contributing to the generic risk of seizures, several CNVs contribute to specific epilepsy types. Carriers of some CNVs tended to have features typical of developmental and epileptic encephalopathies with neurodevelopmental and non-seizure phenotypes. Conversely, carriers of others had phenotypes restricted to the core epileptic features of seizures and electroencephalographic abnormalities (both generalized and focal). Interestingly, reciprocal CNVs involving 22q11.21 seemed to produce opposite epilepsy types, with deletion and duplication carriers tending to have generalized and focal epilepsies, respectively. Dose-dependent effects of KLHL22 on DEPDC5 degradation are a possible explanation[68]. Overall, the high degree of pleiotropy among seizure-associated CNVs implies that these CNVs likely impair neurodevelopmental processes rather generically and contribute to the broad spectrum of neurodevelopmental disorders. According to the oligo-/polygenic inheritance model, CNVs may interact with the genetic background or environmental factors to generate the final disease phenotype. Interaction between CNVs and the polygenic background was recently demonstrated in carriers of the schizophrenia-associated 22q11.2 deletion[69]. Support for an oligogenic-CNV disorder model was also recently published[70].

Genome-wide genetic screening for pathogenic CNVs is recommended as a first-tier approach for the postnatal evaluation of individuals with intellectual disability, developmental delay, autism spectrum disorder, multiple congenital anomalies, and prenatal evaluation of fetuses with structural anomalies observed by ultrasound[71–73]. It has previously been shown that CNVs confer significant risk towards epilepsy[1,2,4–8,10,13,74], particularly for individuals with comorbid neurodevelopmental disorders such as intellectual disability[21,74–76]. In contrast to single nucleotide polymorphism SNP GWASs for epilepsy or seizures, where the risk of identified variants is small (OR < 2)[77,78], the effect sizes of the 25 CNVs identified in this study are large (median OR = 11, range 2–53). Our high-resolution phenomic analysis of 10,880 individuals with epilepsy grouped by CNV carrier status illustrates the seizures, EEG and brain imaging findings, and neurodevelopmental and other co-morbidities associated with each CNV. This genotype-first approach complements the traditional single-phenotype, case-control paradigm by taking a simultaneous phenome-wide perspective in individuals deeply phenotyped according to standardized protocols before CNV discovery or genetic association tests. We found phenotypic evidence supporting associations between CNVs, broad markers of epilepsy types, and fine-grained phenotypes. The high-resolution phenotype associations that an epileptologist can recognize derived from the HPO phenotype association analysis and

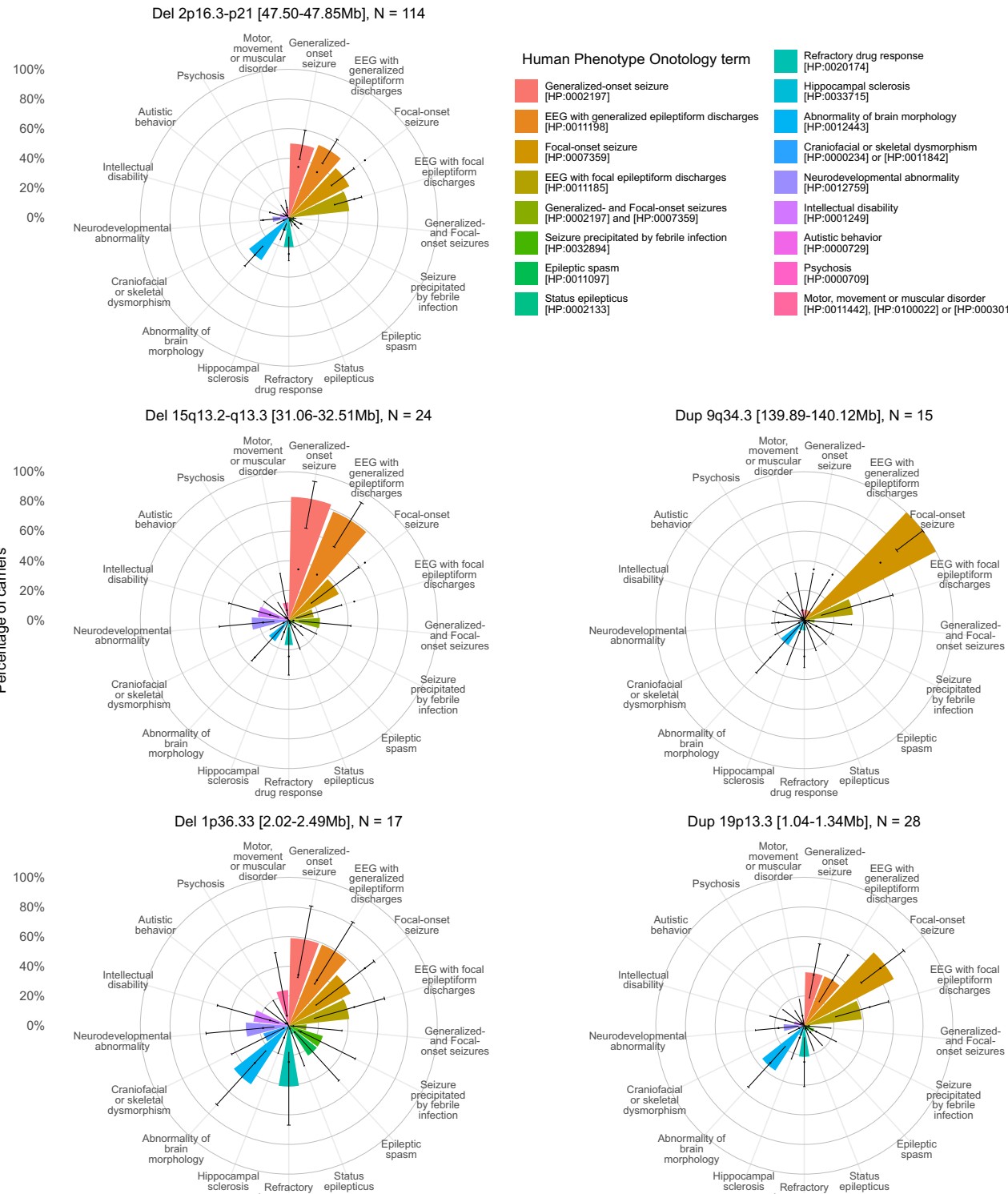

**Fig. 3 | Summary clinical signatures of CNVs in a deeply phenotyped epilepsy cohort.** The percentage of carriers of the CNV with each broad phenotype is shown by the height of bars arranged on a polar axis, with two-sided 95% confidence interval error bars for these percentages derived from the binomial distribution using *stats::binom.test()*. For reference, dots indicate the percentage of the entire Phenomic cohort of 10,880 people with each broad phenotype (representing the prior probability of a person having the phenotype without genetic stratification). The binomial distribution two-sided 95% confidence intervals for a cohort size of 10,880 are no wider than 1.9% (not shown for clarity). "Craniofacial or skeletal dysmorphism" includes individuals with either "Abnormality of the head [HP:

0000234]" (which excludes isolated brain structural abnormalities) or "Abnormal skeletal morphology [HP:0011842]". "Motor, movement or muscular disorder" includes individuals with any of "Abnormal central motor function [HP:0011442]", "Abnormality of movement [HP:0100022]" or "Abnormality of the musculature [HP:0003011]", but not "Motor delay [HP:0001270]", which is included in "Neuro-developmental abnormality". While "Neurodevelopmental abnormality" includes those with "Intellectual disability", the latter is shown additionally as it is a neuro-developmental outcome with particularly important socioeconomically important consequences. EEG electroencephalogram. Further CNV profiles are shown in Supplementary Fig. 2.

disease risk estimates from the meta-analysis for each CNV can enhance the interpretation of clinical relevance and pathogenicity following the American College for Genetics and Genomics Copy Number variant interpretation guidelines[24].

Our study has several limitations. First, many of the patients with seizures included in this study have comorbid neurological and psychiatric disorders. Therefore, some of the identified CNV loci may be associated with other clinical phenotypes present in a high percentage of all cases. Second, we did not detect robust associations with two important outcomes in our HPO analysis, refractory drug response and sudden unexpected death. Sudden unexpected death in epilepsy is poorly suited to cross-sectional studies: it was annotated to only 4 of 10,880 individuals, far fewer cases than expected to occur with follow-up of this cohort of individuals requiring tertiary center care[79]. This emphasizes the open-world interpretation required for our results: in any study that is cross-sectional and of a disorder that has inherently variable phenotyping depth (epilepsy presentations can often be classified only incompletely)[1,62], and which is characterized by some phenotypes that are age-dependent (such as some seizure types, autism, and intellectual disability), one should rarely assume that the absence of an annotation can be interpreted as the absence of that phenotype over the lifetime of the carrier. Thus, the proportion of individuals annotated with a phenotype is likely lower than the actual proportion manifesting it over their lifetimes[80]. Third, in contrast to conventional SNP-based GWASs, CNV-GWASs have major challenges in identifying the causal gene(s) impacted by the CNV. Among the 25 identified CNVs, deletions ranged from 230 kb to 5 Mb and duplications from 290 kb to 9 Mb, affecting 14.2 genes on average. CNV breakpoints in the current study are estimated from genotyped SNPs around the actual breakpoint. These breakpoint estimates are limited by the resolution of the genotyping platform used to call the CNVs. In fact, microarrays have many technical limitations, such as poor breakpoint resolution and limited sensitivity for small CNVs[81]. Newer technologies like whole-genome sequencing (WGS) will enable the assessment of a more comprehensive array of rare variants, including balanced rearrangements, small (exonic) CNVs[82], short tandem repeats, and other structural variants[83]. However, some genomic regions harbor complex deletion/duplication/inversion rearrangements (e.g., 22q11.21[84], 15q11.2[85]) that can even show population stratification (e.g., 16p11.2[86]). More accurate and complete (pangenome) references will be needed to determine the exact breakpoints of such complex rearrangements[87,88], even in the case of sequencing-based CNVs discovery. Lastly, we performed joint epilepsy/seizures and cross-disorder meta-analyses in individuals with minimal clinical information. Future studies with access to rich clinical metadata, such as electronic health records, will likely identify additional seizure-associated CNVs. It is important to consider the inclusion criteria for this cohort and the definition of cases and controls when interpreting associations and their relevance to a patient. Our phenomic analysis cohort was performed using the years 1–3 data of the Epi25 Collaborative, predominantly recruited from academic epilepsy centers and of European ancestry (92.9%, see Online Methods). Additionally, we screened cases to exclude those with brain trauma, meningitis, or encephalitis. Thus, our clinical associations should be considered most valid in individuals of European ancestry with likely genetic or unexplained epilepsies attending specialist epilepsy centers. Future data analyses from subsequent years of Epi25 will provide data more applicable to other populations.

Large-scale collaborations that enable the aggregation of massive datasets have greatly advanced epilepsy and the discovery of genetic factors through GWASs. Here, we have extended this framework to CNV discovery by meta-analyzing epilepsy and seizure disorders, followed by additional meta-analyses in neuropsychiatric disorders and traits to explore pleiotropy. We also identified fine-grained genotype-phenotype associations and clinical profiles for each CNV. Our results

will help refine promising candidate CNVs associated with specific epilepsy types and extend their clinical value. We are confident that applying this framework to even larger datasets has the potential to advance the discovery of all clinically relevant risk loci, ultra-rare high-risk CNVs missed by this study, and the underlying genes or functional elements.

## Methods
### Study cohorts
Each center's ethics committees/institutional review boards approved data collection and use. For the Epi25 cohort, patients or their legal guardians provided signed informed consent/assent according to local IRB requirements; as samples had been collected over 20 years in some centers, forms reflected standards at the time of collection. For Epi25 Consortium samples collected after 25th January 2015, forms required specific language according to the NIH Genomic Data Sharing Policy.

### Individuals with clinically defined epilepsy - Epi25 Collaborative
Individuals with ILAE-defined epilepsy ($N = 16,109$) were collected through the Epi25 Collaborative. The epilepsy diagnosis was performed according to clinical criteria (clinical interview, neurological examination, EEG, imaging data), following International League Against Epilepsy (ILAE) classifications[89]. All cohorts are detailed in Supplementary Table 1. All individuals of the Epi25 Collaborative cohort were selected to be of principal component analysis (PCA)-defined European ancestry. Ancestry-matched population controls ($N = 8545$) for the Epi25 arm of the study were recruited through (1) the Epi25 Collaborative, (2) a Broad Institute project on inflammatory bowel disease without reported epilepsy (part of the IBD Genetics Collaborative, IBDGC), (3) healthy individuals from the Genetics of Personality Collaborative (GPC), and (4) the THL Institute for Health and Welfare (subsample of the FINRISK study)[90]. Genotyping for all cases and controls was performed on the same genotyping array (Illumina Infinium Global Screening Array, GSA-MD v1.0) and at the same center (Broad Institute) as the epilepsy cases. For a detailed description, see ref. 20.

### CNV calling and quality control - Epi25 Collaborative
We restricted our analysis to only autosomal CNVs due to a higher quality of calls and followed the quality control (QC) pipeline developed in our previous study[20]. In detail, QC was performed in two major steps (1) pre- CNV calling QC and (2) post-CNV calling QC. For pre-CNV calling QC, we excluded samples with a call rate <0.96 or discordant sex status. To select individuals of European ancestry, we filtered autosomal SNPs for low genotyping rate (<0.98), a high difference in the SNP minor allele frequency between cases and controls (>0.05), deviation from Hardy-Weinberg equilibrium (HWE) with $P \le 0.001$, and pruned the remaining SNPs for linkage disequilibrium (−indep-pairwise 200 100 0.2) using PLINK v1.9[91]. We then performed a principal component analysis (PCA) of the Epi25 cases and controls using PLINK v1.9[91] and GCTA[92]. European individuals were defined as individuals clustering with the 1000 Genomes Project[93] European samples. We created GC wave-adjusted LRR (Log-R ratio) intensity files for all samples using PennCNV, generated a custom population B-allele frequency file, and employed PennCNV's CNV calling algorithms[2,94] to detect CNVs in our dataset. The post-CNV calling QC included the following steps: (1) CNV calls of the same type (deletion or duplication) were merged if the number of SNP/intensity markers between them was <20% of the total number when both segments were combined; (2) CNVs supported by <20 markers, <20 kb long, and with a SNP density <0.0001 were excluded from subsequent analyses; (3) CNVs that overlapped other CNVs in ≥1% of all samples within the Epi25 dataset were excluded to remove potential platform-specific artifacts, (4) CNVs with >50% overlap with telomeric, centromeric, and

immunoglobulin regions of the hg19 reference assembly were excluded; (5) CNVs with ≥50% overlap with reported common CNVs (allele frequency >1%) in two independent CNV reference catalogs (DGV Gold Standard Dataset[95]; DECIPHER Population Copy-Number Variation Frequencies[96]) were excluded. Finally, the probe-level intensity plots of all CNVs supporting the seizure-associated regions (Table 1) were visually inspected to exclude any remaining artifacts. The DGV Gold Standard and DECIPHER Population frequencies of the remaining CNVs are given in Supplementary Table 4.

### Individuals with seizures or neuropsychiatric phenotypes - neuropsychiatric disorders cohort

A large CNV dataset from individuals with a range of neuropsychiatric disorders (including seizure disorders) was aggregated from 17 different sources by Collins et al.[97]. The contributors of each cohort provided the specific clinical phenotypes. The aggregated individuals were grouped into 54 partially overlapping disease phenotypes standardized through the Human Phenome Ontology[98]. The 54 different phenotypes of Collins et al.[97] were obtained through a recursive hierarchical clustering that defined a minimal set of nonredundant primary phenotypes, each including a minimum of >300 samples in at least three independent cohorts, >3000 samples in total across all cohorts, and had less than 80% sample overlap with any other phenotype. Of the 54 phenotypes, we only selected neurological and psychiatric HPO-based phenotypes ($N = 23$, excluding Seizures, Supplementary Table 2). The architecture of these HPO-based phenotypes allows the identification of associations at different levels, from broad to narrow phenotypes, providing the opportunity to distill between pleiotropic and specific associations. This data set also included the Epi25 cohort from our previous CNV GWAS study[20]. This previous (outdated) Epi25 cohort was excluded from the neuropsychiatric cohort for cross-disorder meta-analyses in the present work. All the considered cohorts are listed in Supplementary Table 1. This aggregated CNV dataset comprised 248,751 individuals affected by at least one of 24 neuropsychiatric disorders, including 10,590 individuals with seizures and 483,779 population controls.

### Quality control - neuropsychiatric disorders cohort

The CNV harmonization procedure for the Neuropsychiatric cohort is described in the Supplementary Materials of Collins et al.[97] and included following steps: (1) CNV calls of the same type (deletion or duplication) were merged if their breakpoints were within ±25% of the size of their corresponding original CNV calls to avoid over-segmentation of large CNV calls; (2) CNVs not mapped to autosomes from the primary hg19 assembly were excluded; (3) Only CNVs between ≥100 kb and ≤20 Mb in size were considered; (4) CNVs that matched reported common CNVs (allele frequency >1%) in three independent CNV reference catalogs derived from genome sequencing (Abel et al.[99]; Collins et al.[100]; Sudmant et al.[81]) were excluded; (5) CNVs that overlapped other CNVs in ≥1% of samples within the same dataset or in any of the other array CNV datasets were excluded to remove potential platform specific artifacts; (6) We excluded all CNVs with ≥30% overlap with somatic hypermutable sites, segmental duplications, simple/low-complexity/satellite repeats, or N-masked bases of the hg19 reference assembly.

### Genome-wide association analysis

We performed segment-based CNV burden analyses to identify genomic regions with a significant increase of CNVs in epilepsy cases compared to controls, separated by CNV type (deletion or duplication). We adopted a sliding window approach as introduced by Collins et al.[26]. The sliding windows model allowed association testing of all autosomes through 267,237 sliding windows characterized by a window size of 200 kb and a step size of 10 kb, corresponding to 13,339.6 non-overlapping windows. Each of these windows was required to have

a low overlap with hypermutable sites, segmental duplications, simple/low-complexity/satellite repeats, and N-masked regions (>30%). For each of the genomic regions, we counted the number of overlapping CNVs separately for cases and controls for each CNV type (deletion or duplication). We required an overlap between the CNV and the genomic window of ≥10% to reveal the potential burden of small deletions or duplications (size ≥ 20 kb). We used the one-sided Fisher test as the test statistic for the CNVs collapsed for each segment. Cases/control CNV counts and the Fisher tests were performed using the CNV docker available at https://hub.docker.com/r/talkowski/rcnv and custom python (version 3.7.9) and R (version 3.6.1) scripts. The same procedure was applied to the cohorts of the neuropsychiatric disorder dataset, as detailed in Collins et al.[26].

### Meta-analysis and fine-mapping

Fixed-effects meta-analyses were performed using the *metafor* R (version 3.6.1) package with an empirical continuity correction[101] and a saddlepoint re-approximation of the null distribution used for inference. The meta-analysis procedure is detailed in Collins et al.[26]. We meta-analyzed the effect sizes from 7 GWAS derived from the 17 cohorts of the neuropsychiatric disorder dataset with each segment-based $P$-value of the Epi25 dataset. The threshold for genome-wide significance was set to $\alpha = 3.74 \times 10^{-6}$ after Bonferroni correction for multiples testing corresponding to the number of independent, non-overlapping 200 kb windows, calculated by merging all overlapping windows and dividing the sum of their sizes by 200 kb (effective $N = 13,339.6$ independent windows; $P = 3.74 \times 10^{-6}$). To account for possible cohort-specific biases, we expected each segment to fulfill the following additional criteria: (1) at least two cohorts featuring nominal significant $P$-values ($P < 0.05$) for the given segment, and (2) a meta-analysis $P < 0.05$ after excluding the single most significant cohort. We then used a Bayesian algorithm[102] to identify the minimal credible interval(s) that contained the causal element(s) or genes with 95% confidence, as in Collins et al.[97]. Finally, we explored the known biological function of all genes within the credible intervals and performed pathway analyses using Enrichr[103,104] (https://maayanlab.cloud/Enrichr/). All resources used to investigate the knowledge basis of all seizure-associated CNV regions are described in Supplementary Table 5.

### Detailed HPO characterization of Epi25 participants

To identify phenotypic associations with each of the CNVs within a cohort of individuals with epilepsy, we translated clinical data from years 1–3 of the deeply phenotyped Epi25 Collaborative international cohort into Human Phenotype Ontology (HPO, version released 2022-02-14) concepts, following our optimization of the HPO for epilepsy phenotypes[105]. We selected only individuals with CNV data and sufficiently detailed clinical data (as of 2022-01-25) to confirm the presence of seizures or epileptic encephalopathy with continuous spike-and-wave in sleep (EE-SWAS, an epilepsy syndrome in which overt clinical seizures may not always be observed). Categorical clinical data were mapped to HPO concepts using a data dictionary. Free text data were annotated with HPO terms manually (D.L.S. under the supervision of I.H. and R.H.T.)[25]. Quantitative data related to the gestational age, weight, and head circumference at birth were categorized to match HPO definitions using sex-stratified distributions from the INTERGROWTH-21th Project using the R *growthstandards* package (version 0.1.5)[106].

We inferred all HPO concepts applicable to each individual from those translated from the clinical data by propagation, following the *is_a* relationships between HPO concepts as previously described[107], using the R *ontologyIndex* package (version 2.7)[108]. We excluded HPO terms that carried no information in the context of this cohort (those that were annotated ubiquitously) and modified the relationships of others, tailoring them to this analysis (Supplementary Table 6).

Phenotypes were annotated as being explicitly present or not, without annotating any phenotypes as being explicitly absent. Taking this open-world perspective is conservative, meaning that the proportion of individuals in a group annotated with a particular phenotype should be considered a lower limit while still allowing statistical testing of phenotypic associations and mitigating the risk of explicitly annotating a phenotype as absent when it was present but not recorded or the individual will manifest the phenotype at some point in the future[80].

After excluding individuals with markers of acquired epilepsy that are unlikely to be part of the phenotype, such as significant brain trauma, encephalitis, or meningitis, 10,880 individuals from the genomic analysis had adequate phenotypic data available for analysis. Of these, 10,106 individuals are of European ancestry, 602 of East Asian ancestry, and 172 of African ancestry, according to PCA analysis. After propagation to infer generic phenotypic descriptors from specific ones, this cohort had 214,203 informative annotations (median = 17 per individual, range = 1–128), spanning a repertoire of 1667 phenotypic concepts. The frequency of annotation of all 1667 phenotypes is available in Supplementary Data 4.

### Phenome-wide association analysis of CNVs
All association analyses and phenomic visualizations were performed in R. Associations between CNVs, and HPO concepts were calculated using the Fisher's exact test (function *fisher.test* from the *stats* package). The tested phenotypes were all those 1667 HPO terms translated from clinical data that were informative (not ubiquitous) and are detailed in Supplementary Table 3. While this was a descriptive analysis, given a large number of tests performed ((29 groups of multiple individuals + 2 groups of a single individual) × 1667 HPO concepts = 51,677)), we sought to aid identification of the most robust associations. Bonferroni's single step and Holm's step-down adjustments are overly conservative given the dependence structure of propagated HPO annotations. For example, after full harmonization, annotations of Typical absence seizure [HP:0011147], Generalized non-motor (absence) seizure [HP:0002121], and Generalized-onset seizure [HP:0002197] will be highly correlated because an individual cannot have the first without the second or the second without the last as a result of there *is_a* relationships in the HPO. Therefore we applied the minP step-down procedure, which uses a permutation-based approach to control the family-wise error rate[61]. We selected 100,000 randomly generated groups of individuals from the Epi25 phenomic analysis cohort of size *N*, where *N* is the number of carriers of each CNV. Then for each of these groups, we calculated the two-sided Fisher's exact test *P*-values for every one of the 1667 HPO concepts. We used the *adj_Wstep* function from the *NRejections* package (version 1.2.0) in R to perform the step-down procedure. This generated *P*-values corrected for the correlation-adjusted number of tested HPO annotations. We did not adjust *P*-values across CNVs because we were interested only in identifying those associations that were most robust in this descriptive analysis.

### Reporting summary
Further information on research design is available in the Nature Portfolio Reporting Summary linked to this article.

## Data availability
All genome-wide CNV association summary statistics are available at Zenodo (https://zenodo.org/record/7939126#.ZGK7yi-B29Y with https://doi.org/10.5281/zenodo.7939126). Individual-level CNV data for epilepsy patients are available from the Epi25 Consortium (http://epi-25.org/) upon signing the Epi25 charter (See Epi25 page http://epi-25.org/) and submission and acceptance of a full research proposal. Furthermore, raw data is deposited at dbGAP https://www.ncbi.nlm.nih.gov/projects/gap/cgi-bin/study.cgi?study_id=phs001551.v1.p1. All HPO-based phenome-wide summary statistics are available in Supplementary Data 3 of this manuscript. Fine-mapping results are available in Supplementary Data 1 and 2 of this manuscript. The CNV data of the Neuropsychiatric cohort are described in the Supplementary Materials of Collins et al.[97]. They can be accessed from existing publications, public resources, or, upon request, from the authors of Collins et al.[97] (see "Key resources table" and Table S2 in Collins et al.[97]). The CNV data reported by GeneDx and Indiana University clinical testing sites were not consented for public release. All datasets used in this study are detailed in Supplementary Table 1 of our manuscript.

## Code availability
The code for the association and meta analysis is available and have been deposited at Zenodo (https://github.com/talkowski-lab/rCNV2/tree/v1.0, with https://doi.org/10.5281/zenodo.6647918). Also, we provided a Docker image hosted on DockerHub (https://hub.docker.com/r/talkowski/rcnv) and Google Container Registry (https://gcr.io/gnomad-wgs-v2-sv/rcnv), which provides a controlled container environment containing all dependencies necessary to execute the code identically as presented in this study.

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

## Acknowledgements

This research was funded in whole, or in part, by the Wellcome Trust [203914/Z/16/Z], supporting D.L.S. For the purpose of Open Access, the author has applied a CC BY public copyright license to any Author Accepted Manuscript version arising from this submission. I.H. was supported by The Hartwell Foundation (Individual Biomedical Research Award), the National Institute for Neurological Disorders and Stroke (K02 NS112600), the Eunice Kennedy Shriver National Institute of Child Health and Human Development through the Intellectual and Developmental Disabilities Research Center (IDDRC) at Children's Hospital of Philadelphia and the University of Pennsylvania (U54 HD086984), and by the German Research Foundation (HE5415/3-1, HE5415/5-1, HE5415/6-1, HE5415/7-1). Research reported in this publication was also supported by the National Center for Advancing Translational Sciences of the National Institutes of Health (UL1TR001878), by the Institute for Translational Medicine and Therapeutics' (ITMAT) at the Perelman School of Medicine of the University of Pennsylvania, and by Children's Hospital of Philadelphia through the Epilepsy NeuroGenetics Initiative (ENGIN). R.L.C. was supported by NHGRI T32HG002295 and NSF GRFP #2017240332. We thank the Epi25 principal investigators, local staff from individual cohorts, and all of the patients with epilepsy who participated in the study for making this global collaboration and resource possible to advance epilepsy genetics research. This work is part of the Centers for Common Disease Genomics (CCDG) program, funded by the National Human Genome Research Institute (NHGRI) and the National Heart, Lung, and Blood Institute (NHLBI). CCDG-funded Epi25 research activities at the Broad Institute, including genomic data generation in the Broad Genomics Platform, are supported by NHGRI grant UM1 HG008895 (PIs: Eric Lander, Stacey Gabriel, Mark Daly, Sekar Kathiresan). The Genome Sequencing Program efforts were also supported by NHGRI grant 5U01HG009088-02. The content is solely the responsibility of the authors and does not necessarily represent the official views of the National Institutes of Health. We thank the Stanley Center for Psychiatric Research at the Broad Institute for supporting the genomic data generation efforts and control sample aggregation.

## Author contributions

Management of the study and content of the manuscript was the responsibility of D.L., C.L., L.M., D.L.S. and I.H. D.L. and C.L. designed the project, mentored all steps, coordinated the contributions, and guaranteed scientific coherence. D.L., C.L., L.M., T.B., M.M., D.L.S., R.T., R.L.C. and I.H. interpreted the results. The Epi25 Collaborative provided raw genotypic data for ~18,000 epilepsy patients and ~11,000 controls (Epi25 Collaborative cohort). M.T. provided a clean aggregated dataset of ~260,000 patients and ~460,000 controls with CNV calls (neuropsychiatric disorders cohort). L.M.N., L.M., and C.L. carried out CNV calls and QC. L.M. performed the GWAS for the Epi25 dataset. R.L.C. performed the meta-analyses of the Epi25 GWAS with the neuropsychiatric disorders cohort. With the supervision of I.H. and R.T., D.L.S. developed the HPO-based phenome-wide association framework and applied it for each CNV. D.L.S. and S.G. cleaned the raw Epi25 phenotypic data. D.L.S., S.P., and J.X. illustrated the Epi25 phenome-wide association results. L.M., C.L., D.L., D.L.S., and R.L.C. wrote the manuscript. All authors including members of the Epi25 Collaborative, saw, commented, and approved the final draft.

## Competing interests

R.H.T. received honoraria from Arvelle/Angelini, Bial, Eisai, GW Pharma/ Jazz, Sanofi, UCB Pharma, and Zogenix, meeting support from LivaNova, Bial, Novartis, UCB Pharma, and unrestricted funding support from Arvelle/Angelini and UNEEG. M.E.T. receives research funding or reagents from Levo Therapeutics, Microsoft Inc., and Illumina Inc. All other authors report no competing interests.

## Additional information

[1]Genomic Medicine Institute, Lerner Research Institute, Cleveland Clinic, Cleveland, USA. [2]Translational and Clinical Research Institute, Newcastle University, Newcastle upon Tyne, UK. [3]Clinical Neurosciences, Newcastle upon Tyne Hospitals NHS Foundation Trust, Newcastle upon Tyne, UK. [4]The Epilepsy NeuroGenetics Initiative, Children's Hospital of Philadelphia, Philadelphia, PA, USA. [5]Department of Biomedical and Health Informatics (DBHi), Children's Hospital of Philadelphia, Philadelphia, PA, USA. [6]Center for Genomic Medicine, Massachusetts General Hospital, Boston, USA. [7]Program in Medical and Population Genetics, Broad Institute of Massachusetts Institute of Technology (M.I.T.) and Harvard, Cambridge, USA. [8]Cologne Center for Genomics, University of Cologne, Cologne, Germany. [9]Department of Neurology, University of Pennsylvania, Perelman School of Medicine, Philadelphia, PA, USA. [10]Department of Clinical and Experimental Epilepsy, Institute of Neurology, University College London, London, UK. [11]Stanley Center for Psychiatric Research, Broad Institute of Harvard and M.I.T, Cambridge, MA, USA. [12]Epilepsy Center, Neurological Institute, Cleveland Clinic, Cleveland, OH, US. [170]These authors

contributed equally: Ludovica Montanucci, David Lewis-Smith, Ryan L. Collins, Lisa-Marie Niestroj. [171]These authors jointly supervised this work: Costin Leu, Dennis Lal. ✉e-mail: costin.leu@uth.tmc.edu; Dennis.Lal@uth.tmc.edu

## Epi25 Collaborative

**Columbia University Institute for Genomic Medicine analysis group** Joshua E. Motelow[13], Gundula Povysil[13], Ryan S. Dhindsa[13], Kate E. Stanley[13], Andrew S. Allen[14] & David B. Goldstein[13]

**Epi25 sequencing, analysis, project management, and browser development at the Broad Institute** Yen-Chen Anne Feng[11,15,16,17], Daniel P. Howrigan[11,15,17], Liam E. Abbott[11,15,17], Katherine Tashman[11,15,17], Felecia Cerrato[11], Caroline Cusick[11], Tarjinder Singh[11,15,17], Henrike Heyne[11,15,17], Andrea E. Byrnes[11,15,17], Claire Churchhouse[11,15,17], Nick Watts[11,15,17], Matthew Solomonson[11,15,17], Dennis Lal[1,11], Namrata Gupta[17] & Benjamin M. Neale[11,15,17]

**Epi25 executive committee** Samuel F. Berkovic[18], Holger Lerche[19], David B. Goldstein[13] & Daniel H. Lowenstein[20]

**Epi25 strategy, phenotyping, analysis, informatics, and project management committees**
Samuel F. Berkovic[18], Holger Lerche[19], David B. Goldstein[13], Daniel H. Lowenstein[20], Gianpiero L. Cavalleri[21,22], Patrick Cossette[23], Chris Cotsapas[24], Peter De Jonghe[25,26,27], Tracy Dixon-Salazar[28], Renzo Guerrini[29], Hakon Hakonarson[30], Erin L. Heinzen[13], Ryan S. Dhindsa[13], Kate E. Stanley[13], Ingo Helbig[5,9,30], Patrick Kwan[31,32], Anthony G. Marson[33], Slavé Petrovski[32,34], Sitharthan Kamalakaran[13], Sanjay M. Sisodiya[35,36], Randy Stewart[37], Sarah Weckhuysen[25,26,27], Chantal Depondt[38], Dennis J. Dlugos[9,30], Ingrid E. Scheffer[18], Pasquale Striano[39,40], Catharine Freyer[20], Roland Krause[41], Patrick May[41], Kevin McKenna[20], Brigid M. Regan[18], Caitlin A. Bennett[18], Stephanie L. Leech[18], Costin Leu[1,11,35,36] & David Lewis-Smith[2,3,5,30]

**Authors from individual Epi25 cohorts:**

**Australia: Melbourne (AUSAUS)** Samuel F. Berkovic[18], Ingrid E. Scheffer[18], Brigid M. Regan[18], Caitlin A. Bennett[18] & Stephanie L. Leech[18]

**Australia: Royal Melbourne (AUSRMB)** Terence J. O'Brien[31,32], Slavé Petrovski[32,34], Marian Todaro[31,32] & Patrick Kwan[31,32]

**Belgium: Antwerp (BELATW)** Sarah Weckhuysen[25,26,27], Peter De Jonghe[25,26,27] & Hannah Stamberger[25,26,27]

**Belgium: Brussels (BELULB)** Chantal Depondt[38]

**Canada: Andrade (CANUTN)** Danielle M. Andrade[42,43], Quratulain Zulfiqar Ali[42] & Tara R. Sadoway[43]

**Switzerland: Bern (CHEUBB)** Heinz Krestel[24] & André Schaller[44]

**Cyprus (CYPCYP)** Savvas S. Papacostas[45], Ioanna Kousiappa[45], George A. Tanteles[46] & Christou Yiolanda[45]

**Czech Republic: Prague (CZEMTH)** Katalin Štěrbová[47], Markéta Vlčková[47], Lucie Sedláčková[47] & Petra Laššuthová[47]

**Germany: Frankfurt/Marburg (DEUPUM)** Karl Martin Klein[48,49,50,51], Felix Rosenow[48,49,50], Philipp S. Reif[48,49,50] & Susanne Knake[49,50]

**Germany: Giessen (DEUUGS)** Bernd A. Neubauer[52], Friedrich Zimprich[53], Martha Feucht[52] & Eva Reinthaler[53]

**Germany: Bonn (DEUUKB)** Wolfram S. Kunz[54,55], Gábor Zsurka[54,55], Rainer Surges[55], Tobias H. Baumgartner[55] & Randi von Wrede[55]

**Germany: Kiel (DEUUKL)** Ingo Helbig[30,56,57], Karl Martin Klein[48,49,50,51], Manuela Pendziwiat[56,57], Hiltrud Muhle[57], Annika Rademacher[57], Andreas van Baalen[57], Sarah von Spiczak[57,58], Ulrich Stephani[57], Zaid Afawi[59], Amos D. Korczyn[60], Moien Kanaan[61], Christina Canavati[61], Gerhard Kurlemann[62], Karen Müller-Schlüter[63], Gerhard Kluger[64,65], Martin Häusler[66] & Ilan Blatt[60,67]

**Germany: Leipzig (DEUULG)** Johannes R. Lemke[68] & Ilona Krey[68]

**Germany: Tuebingen (DEUUTB)** Holger Lerche[19], Yvonne G. Weber[19,69], Stefan Wolking[19], Felicitas Becker[19,70], Stephan Lauxmann[19], Christian Bosselmann[19], Josua Kegele[19], Christian Hengsbach[19], Sarah Rau[19], Bernhard J. Steinhoff[71], Andreas Schulze-Bonhage[72], Ingo Borggräfe[73,74], Christoph J. Schankin[75,76], Susanne Schubert-Bast[72], Herbert Schreiber[77], Thomas Mayer[78], Rudolf Korinthenberg[79], Knut Brockmann[80], Markus Wolff[81], Gerhard Kurlemann[82], Dieter Dennig[83], Rene Madeleyn[84] & Josua Kegele[19]

**Finland: Kuopio (FINKPH)** Reetta Kälviäinen[85,86], Anni Saarela[85] & Oskari Timonen[86]

**Finland: Helsinki (FINUVH)** Tarja Linnankivi[87] & Anna-Elina Lehesjoki[88]

**France: Lyon (FRALYU)** Sylvain Rheims[89,90], Gaetan Lesca[91], Philippe Ryvlin[92], Louis Maillard[93], Luc Valton[94], Philippe Derambure[95], Fabrice Bartolomei[96], Edouard Hirsch[97], Véronique Michel[98] & Francine Chassoux[99]

**Wales: Swansea (GBRSWU)** Mark I. Rees[100], Seo-Kyung Chung[100,101], William O. Pickrell[102], Robert H. W. Powell[103], Mark D. Baker[102], Beata Fonferko-Shadrach[102], Charlotte Lawthom[104] & Joe Anderson[104]

**UK: UCL (GBRUCL)** Sanjay M. Sisodiya[35,36], Natascha Schneider[35,36], Simona Balestrini[35,36], Sara Zagaglia[35,36] & Vera Braatz[35,36]

**UK: Imperial/Liverpool (GBRUNL)** Anthony G. Marson[33], Michael R. Johnson[105], Pauls Auce[106] & Graeme J. Sills[107]

**Hong Kong (HKGHKK)** Patrick Kwan[31,32,108], Larry W. Baum[109], Pak C. Sham[109], Stacey S. Cherny[110] & Colin H. T. Lui[111]

**Ireland: Dublin (IRLRCI)** Gianpiero L. Cavalleri[21,22], Norman Delanty[21,22,112], Colin P. Doherty[22,113], Arif Shukralla[22,112], Hany El-Naggar[21,22,112] & Peter Widdess-Walsh[21,22,112]

**Croatia (HRVUZG)** Nina Barišić[114]

**Italy: Milan (ITAICB)** Laura Canafoglia[115], Silvana Franceschetti[116], Barbara Castellotti[117], Tiziana Granata[118] & Francesca Ragona[118]

**Italy: Genova (ITAIGI)** Pasquale Striano[39,40], Federico Zara[39,40], Michele Iacomino[39], Antonella Riva[39,40], Francesca Madia[39,40], Maria Stella Vari[39], Vincenzo Salpietro[39], Marcello Scala[39], Maria Margherita Mancardi[39], Nobili Lino[39], Elisa Amadori[39] & Thea Giacomini[39]

**Italy: Bologna (ITAUBG)** Francesca Bisulli[119,120], Tommaso Pippucci[121], Laura Licchetta[119,120], Raffaella Minardi[119], Paolo Tinuper[119,120], Lorenzo Muccioli[119,120] & Barbara Mostacci[119]

**Italy: Catanzaro (ITAUMC)** Antonio Gambardella[122], Angelo Labate[122], Grazia Annesi[123], Lorella Manna[123] & Monica Gagliardi[123]

**Italy: Florence (ITAUMR)** Renzo Guerrini[29], Elena Parrini[29], Davide Mei[29], Annalisa Vetro[29], Claudia Bianchini[29], Martino Montomoli[29], Viola Doccini[29] & Carmen Barba[29]

**Japan: Fukuoka (JPNFKA)** Shinichi Hirose[124] & Atsushi Ishii[124]

**Japan: RIKEN Institute (JPNRKI)** Toshimitsu Suzuki[125,126], Yushi Inoue[127] & Kazuhiro Yamakawa[125,126]

**Lebanon: Beirut (LEBABM)** Ahmad Beydoun[128], Wassim Nasreddine[128] & Nathalie Khoueiry-Zgheib[129]

**Lithuania (LTUUHK)** Birute Tumiene[130,131] & Algirdas Utkus[130,131]

**New Zealand: Otago (NZLUTO)** Lynette G. Sadleir[132] & Chontelle King[132]

**Turkey: Bogazici (TURBZU)** S. Hande Caglayan[133], Mutluay Arslan[134], Zuhal Yapıcı[135], Pınar Topaloglu[135], Bulent Kara[136], Uluc Yis[136], Dilsad Turkdogan[136] & Aslı Gundogdu-Eken[133]

**Turkey: Istanbul (TURIBU)** Nerses Bebek[137,138], Sibel Uğur-İşeri[138], Betül Baykan[137], Barış Salman[138], Garen Haryanyan[137], Emrah Yücesan[139], Yeşim Kesim[137] & Çiğdem Özkara[140]

**Taiwan (TWNCGM)** Meng-Han Tsai[141], Chen-Jui Ho[141], Chih-Hsiang Lin[141], Kuang-Lin Lin[142] & I-Jun Chou[142]

**USA: BCH (USABCH)** Annapurna Poduri[143,144], Beth R. Shiedley[143,144] & Catherine Shain[143,144]

**USA: Baylor College of Medicine (USABLC)** Jeffrey L. Noebels[145] & Alicia Goldman[145]

**USA: Cleveland Clinic (USACCF)** Robyn M. Busch[1,12,146], Lara Jehi[12,146], Imad M. Najm[12,146], Dennis Lal[1,12], Lisa Ferguson[12] & Jean Khoury[12,146]

**USA: Cincinnati Children's Hospital Medical Center (USACCH)** Tracy A. Glauser[147] & Peggy O. Clark[147]

**USA: Philadelphia/CHOP (USACHP) and Philadelphia/Rowan (USACRW)** Russell J. Buono[30,148,149], Thomas N. Ferraro[148], Michael R. Sperling[149], Dennis J. Dlugos[30,150], Warren Lo[150], Michael Privitera[151], Jacqueline A. French[152], Patrick Cossette[23], Steven Schachter[153] & Hakon Hakonarson[30]

**USA: EPGP (USAEGP)** Daniel H. Lowenstein[20], Ruben I. Kuzniecky[154], Dennis J. Dlugos[9,30] & Orrin Devinsky[152]

**USA: NYU HEP (USAHEP)** Daniel H. Lowenstein[20], Ruben I. Kuzniecky[154], Jacqueline A. French[152] & Manu Hegde[20]

**USA: Nationwide Children's Hospital (USANCH)** David A. Greenberg[155]

**USA: Penn/CHOP (USAUPN)** Ingo Helbig[5,9,30], Colin A. Ellis[9], Ethan Goldberg[4,30], Katherine L. Helbig[4,30], David Lewis-Smith[2,3,5,30], Mahgenn Cosico[4,30], Priya Vaidiswaran[4,30] & Eryn Fitch[4,30]

**Kenya: Kilifi; South Africa: Aguincourt; Ghana: Kintampo (KENKIL, GHAKNT, ZAFAGN)** Charles R. J. C. Newton[156,157,158], Symon M. Kariuki[156,157,158], Ryan G. Wagner[159,160,161] & Seth Owusu-Agyei[162,163]

**USA: Massachusetts General Hospital (USAMGH)** Andrew J. Cole[164], Christopher M. McGraw[164] & S. Anthony Siena[165]

**USA: Vanderbilt University Medical Centre (USAVAN)** Lea Davis[166,167,168,169], Donald Hucks[166,169], Annika Faucon[169], David Wu[169], Bassel W. Abou-Khalil[169], Kevin Haas[169] & Randip S. Taneja[169]

[13]Institute for Genomic Medicine, Columbia University, New York, NY 10032, USA. [14]Division of Integrative Genomics, Department of Biostatistics and Bioinformatics, Duke University, Durham, NC 27710, USA. [15]Analytic and Translational Genetics Unit, Department of Medicine, Massachusetts General Hospital and Harvard Medical School, Boston, MA 02114, USA. [16]Psychiatric & Neurodevelopmental Genetics Unit, Department of Psychiatry, Massachusetts General Hospital and Harvard Medical School, Boston, MA 02114, USA. [17]Program in Medical and Population Genetics, Broad Institute of Harvard and MIT, 7 Cambridge Center, Cambridge, MA 02142, USA. [18]Epilepsy Research Centre, Department of Medicine, University of Melbourne, Victoria, Australia. [19]Department of Neurology and Epileptology, Hertie Institute for Clinical Brain Research, University of Tübingen, 72076 Tübingen, Germany. [20]Department of Neurology, University of California, San Francisco, CA 94110, USA. [21]School of Pharmacy and Biomolecular Sciences, The Royal College of Surgeons in Ireland, Dublin, Ireland. [22]The FutureNeuro Research Centre, Dublin, Ireland. [23]University of Montreal, Montreal, QC H3T 1J4, Canada. [24]School of Medicine, Yale University, New Haven, CT 06510, USA. [25]Neurogenetics Group, Center for Molecular Neurology, VIB Antwerp, Belgium. [26]Laboratory of Neurogenetics, Institute Born-Bunge, University of Antwerp, Antwerp, Belgium. [27]Division of Neurology, Antwerp University Hospital, Antwerp, Belgium. [28]LGS Foundation, NY New York 11716, USA. [29]Pediatric Neurology, Neurogenetics and Neurobiology Unit and Laboratories, Children's Hospital A. Meyer, University of Florence, Florence, Italy. [30]Division of Neurology, Children's Hospital of Philadelphia, Philadelphia, PA 19104, USA. [31]Department of Neuroscience, Central Clinical School, Monash University, Alfred Hospital, Melbourne, Australia. [32]Departments of Medicine and Neurology, University of Melbourne, Royal Melbourne Hospital, Parkville, Australia. [33]Department of Pharmacology and Therapeutics, University of Liverpool, Liverpool, UK. [34]Centre for Genomics Research, Precision Medicine and Genomics, IMED Biotech Unit, AstraZeneca, Cambridge, UK. [35]Department of Clinical and Experimental Epilepsy, UCL Queen Square Institute of Neurology, London, UK. [36]Chalfont Centre for Epilepsy, Chalfont St Peter, UK. [37]National Institute of Neurological Disorders and Stroke, Bethesda, MD 20852, USA. [38]Department of Neurology, Université Libre de Bruxelles, Brussels, Belgium. [39]IRCCS "G. Gaslini" Institute, Genova, Italy. [40]Department of Neurosciences, Rehabilitation, Ophthalmology, Genetics, Maternal and Child Health, University of Genoa, Genoa, Italy. [41]Luxembourg Centre for Systems Biomedicine, University Luxembourg, Esch-sur-Alzette, Luxembourg. [42]Department of Neurology, Toronto Western Hospital, Toronto, ON M5T 2S8, Canada. [43]University Health Network, University of Toronto, Toronto, ON, Canada. [44]Institute of Human Genetics, Bern University Hospital, Bern, Switzerland. [45]Department of Neurobiology and Clinical Neurophysiology (Epilepsy Center), The Cyprus Institute of Neurology and Genetics, 2370 Nicosia, Cyprus. [46]Department of Clinical Genetics, The Cyprus Institute of Neurology and Genetics, 2370 Nicosia, Cyprus. [47]Department of Paediatric Neurology, 2nd Faculty of Medicine, Charles University and Motol Hospital, Prague, Czech Republic. [48]Epilepsy Center Frankfurt Rhine-Main, Center of Neurology and Neurosurgery, Goethe University Frankfurt, Frankfurt, Germany. [49]Epilepsy Center Hessen-Marburg, Department of Neurology, Philipps University Marburg, Marburg, Germany. [50]LOEWE Center for Personalized Translational Epilepsy Research (CePTER), Goethe University Frankfurt, Frankfurt, Germany.

[51]Departments of Clinical Neurosciences, Medical Genetics and Community Health Sciences, Hotchkiss Brain Institute & Alberta Children's Hospital Research Institute, Cumming School of Medicine, University of Calgary, Calgary, Alberta, Canada. [52]Pediatric Neurology, Children's Hospital, Giessen, Germany. [53]Department of Neurology, Medical University of Vienna, Vienna, Austria. [54]Institute of Experimental Epileptology and Cognition Research, University Bonn, 53127 Bonn, Germany. [55]Department of Epileptology, University Bonn, 53127 Bonn, Germany. [56]Institute of Clinical Molecular Biology, Christian-Albrechts-University of Kiel, Kiel, Germany. [57]Department of Neuropediatrics, Christian-Albrechts-University of Kiel, 24105 Kiel, Germany. [58]DRK-Northern German Epilepsy Centre for Children and Adolescents, 24223 Schwentinental-Raisdorf, Germany. [59]Sackler School of Medicine, Tel-Aviv University, Ramat Aviv, Israel. [60]Tel-Aviv University Sackler Faculty of Medicine, Ramat Aviv 69978, Israel. [61]Hereditary Research Lab, Bethlehem University, Bethlehem, Palestine. [62]Bonifatius Hospital Lingen, Neuropediatrics Wilhelmstrasse 13, 49808 Lingen, Germany. [63]Epilepsy Center for Children, University Hospital Neuruppin, Brandenburg Medical School, Neuruppin, Germany. [64]Neuropediatric Clinic and Clinic for Neurorehabilitation, Epilepsy Center for Children and Adolescents, Vogtareuth, Germany. [65]Research Institute Rehabilitation / Transition / Palliation, PMU Salzburg, Austria. [66]Division of Neuropediatrics and Social Pediatrics, Department of Pediatrics, University Hospital, RWTH Aachen, Aachen, Germany. [67]Department of Neurology, Sheba Medical Center, Ramat Gan, Israel. [68]Institute of Human Genetics, Leipzig, Germany. [69]Department of Epileptology and Neurology, University of Aachen, Aachen, Germany. [70]Department of Neurology, University of Ulm, Ulm, Germany. [71]Epilepsy Center Kork, Kehl, Germany. [72]Department of Epileptology, University Hospital Freiburg, Freiburg, Germany. [73]Department of Pediatric Neurology, Ludwig Maximilians University, Munchen, Germany. [74]Dr von Hauner Childrens Hospital, Munich, Germany. [75]Department of Neurology, Ludwig Maximilians University, Munchen, Germany. [76]Department of Neurology, University Hospital Bern, Bern, Switzerland. [77]Neurological Practice Center & Neuropoint Patient Academy, Ulm, Germany. [78]Epilepsy Center Kleinwachau, Radeberg, Germany. [79]Department of Neuropediatrics and Muscular Disorders, University Hospital Freiburg, Freiburg, Germany. [80]University Children's Hospital, Göttingen, Germany. [81]Department of Pediatric Neurology & Developmental Medicine, University Childrens Hospital, Tubingen, Germany. [82]Department of Neuropediatrics, Westfalische Wilhelms University, Munster, Germany. [83]Private Neurological Practice, Stuttgart, Germany. [84]Department of Pediatrics, Filderklinik, Filderstadt, Germany. [85]Epilepsy Center, Kuopio University Hospital, Kuopio, Finland. [86]Institute of Clinical Medicine, University of Eastern Finland, Kuopio, Finland. [87]Child Neurology, New Children's Hospital and Pediatric Research Center, University of Helsinki and Helsinki University Hospital, Helsinki, Finland. [88]Medicum, University of Helsinki, Helsinki, Finland and Folkhälsan Research Center, Helsinki, Finland. [89]Department of Functional Neurology and Epileptology, Hospices Civils de Lyon and University of Lyon, Lyon, France. [90]Lyon's Neuroscience Research Center, INSERM U1028 / CNRS UMR 5292, Lyon, France. [91]Department of Clinical Genetics, Hospices Civils de Lyon and University of Lyon, Lyon, France. [92]Department of Clinical Neurosciences, Centre Hospitalo-Universitaire Vaudois, Lausanne, Switzerland. [93]Neurology Department, University Hospital of Nancy, UMR 7039, CNRS, Lorraine University, Nancy, France. [94]Clinical Neurophysiology and Epilepsy Unit, Department of Neurology, University Hospital of Toulouse, Toulouse, France. [95]Department of Clinical Neurophysiology, Lille University Medical Center, EA 1046, University of Lille, Lille, France. [96]Clinical Neurophysiology and Epileptology Department. Timone Hospital, Marseille, France. [97]Department of Neurology, University Hospital of Strasbourg, Strasbourg, France. [98]Department of Neurology, Hôpital Pellegrin, Bordeaux, France. [99]Epilepsy Unit, Department of Neurosurgery, Centre Hospitalier Sainte-Anne and University Paris Descartes, Paris, France. [100]Faculty of Medicine and Health, University of Sydney, Sydney, Australia. [101]Kids Neuroscience Centre, Kids Research, Children Hospital at Westmead, Sydney, NSW, Australia. [102]Swansea University Medical School, Swansea University, SA2 8PP Swansea, UK. [103]Department of Neurology, Morriston Hospital, Swansea Bay University Health Board, Swansea, Wales, UK. [104]Neurology Department, Aneurin Bevan University Health Board, Newport, Wales, UK. [105]Division of Brain Sciences, Imperial College London, London, UK. [106]Department of Neurology, Walton Centre NHS Foundation Trust, Liverpool, UK. [107]School of Life Sciences, University of Glasgow, Glasgow, UK. [108]Department of Medicine and Therapeutics, Chinese University of Hong Kong, Hong Kong, China. [109]Department of Psychiatry, University of Hong Kong, Hong Kong, China. [110]Department of Epidemiology and Preventive Medicine and Department of Anatomy and Anthropology, Sackler Faculty of Medicine, Tel Aviv University, Tel Aviv, Israel. [111]Department of Medicine, Tseung Kwan O Hospital, Hong Kong, China. [112]The Department of Neurology, Beaumont Hospital, Dublin, Ireland. [113]Neurology Department, St. James Hospital, Dublin, Ireland. [114]Department of Pediatric University Hospital centre Zagreb, Zagreb, Croatia. [115]Integrated Diagnostics for Epilepsy, Fondazione IRCCS Istituto Neurologico Carlo Besta, Milan, Italy. [116]Neurophysiology, Fondazione IRCCS Istituto Neurologico Carlo Besta, Milan, Italy. [117]Unit of Genetics of Neurodegenerative and Metabolic Diseases, Fondazione IRCCS Istituto Neurologico Carlo Besta, Milan, Italy. [118]Department of Pediatric Neuroscience, Fondazione IRCCS Istituto Neurologico Carlo Besta, Milan, Italy. [119]IRCCS, Institute of Neurological Sciences of Bologna, Bologna, Italy. [120]Department of Biomedical and Neuromotor Sciences, University of Bologna, Bologna, Italy. [121]Medical Genetics Unit, Polyclinic Sant'Orsola-Malpighi University Hospital, Bologna, Italy. [122]Institute of Neurology, Department of Medical and Surgical Sciences, University "Magna Graecia", Catanzaro, Italy. [123]Institute of Molecular Bioimaging and Physiology, CNR, Section of Germaneto, Catanzaro, Italy. [124]Department of Pediatrics, School of Medicine, Fukuoka University, Fukuoka, Japan. [125]Laboratory for Neurogenetics, RIKEN Center for Brain Science, Saitama, Japan. [126]Department of Neurodevelopmental Disorder Genetics, Institute of Brain Science, Nagoya City University Graduate School of Medical Science, Nagoya, Aichi, Japan. [127]National Epilepsy Center, Shizuoka Institute of Epilepsy and Neurological Disorder, Shizuoka, Japan. [128]Department of Neurology, American University of Beirut Medical Center, Beirut, Lebanon. [129]Department of Pharmacology and Toxicology, American University of Beirut Faculty of Medicine, Beirut, Lebanon. [130]Institute of Biomedical Sciences, Faculty of Medicine, Vilnius University, Vilnius, Lithuania. [131]Centre for Medical Genetics, Vilnius University Hospital Santaros Klinikos, Vilnius, Lithuania. [132]Department of Paediatrics and Child Health, University of Otago, Wellington, New Zealand. [133]Department of Molecular Biology and Genetics, Bogaziçi University, Istanbul, Turkey. [134]Department of Child Neurology, Gulhane Education and Research Hospital, Health Sciences University, Ankara, Turkey. [135]Department of Child Neurology, Istanbul Faculty of Medicine, Istanbul University, Istanbul, Turkey. [136]Department of Child Neurology, Medical School, Marmara University, Istanbul, Turkey. [137]Department of Neurology, Istanbul Faculty of Medicine, Istanbul University, Istanbul, Turkey. [138]Department of Genetics, Aziz Sancar Institute of Experimental Medicine, Istanbul University, Istanbul, Turkey. [139]Bezmialem Vakif University, Institute of Life Sciences and Biotechnology, Istanbul, Turkey. [140]Department of Neurology, Faculty of Medicine, Cerrahpaşa University Istanbul, Istanbul, Turkey. [141]Department of Neurology, Kaohsiung Chang Gung Memorial Hospital, Kaohsiung, Taiwan. [142]Department of Pediatric Neurology, Chang Gung Memorial Hospital, Taoyuan, Taiwan. [143]Epilepsy Genetics Program, Department of Neurology, Boston Children's Hospital, Boston, MA 02115, USA. [144]Department of Neurology, Harvard Medical School, Boston, MA 02115, USA. [145]Neurology, Baylor College of Medicine, Texas, US. [146]Department of Neurology, Neurological Institute, Cleveland Clinic, Cleveland, OH 44195, USA. [147]Cincinnati Children's Hospital Medical Center, Cincinnati, Ohio, USA. [148]Cooper Medical School of Rowan University, Camden, NJ 08103, USA. [149]Thomas Jefferson University, Philadelphia, PA 19107, USA. [150]Nationwide Children's Hospital, Columbus, OH 3205, USA. [151]University of Cincinnati, Cincinnati, OH 45220, USA. [152]Department of Neurology, New York University, New York, NY, USA. [153]Beth Israel Deaconess/ Harvard, Boston, MA 02115, USA. [154]Department of Neurology, Hofstra-Northwell Medical School, New York, NY 11549, USA. [155]Department of Pediatrics, Ohio State Wexner Medical Center, Battelle Center for Mathematical Medicine. Columbus, Ohio, USA. [156]Neuroscience Unit, KEMRI-Wellcome Trust Research Programme, Kilifi, Kenya. [157]Department of Public Health, Pwani University, Kilifi, Kenya. [158]Department of Psychiatry, University of Oxford, Oxford, UK. [159]MRC/Wits Rural Public Health & Health Transitions Research Unit (Agincourt), School of Public Health, Faculty of Health Sciences, University of the

Witwatersrand, Johannesburg, South Africa. [160]Department of Epidemiology and Global Health, Umeå University, Umeå, Sweden. [161]Department of Clinical Sciences, Neurosciences, Umeå University, Umeå, Sweden. [162]Kintampo Health Research Centre, Ghana Health Service, Kintampo, Ghana. [163]University of Health and Allied Science in Ho, Ho, Ghana. [164]Neurology, Massachusetts General Hospital, Boston, MA, USA. [165]Medical School, Nova Southeastern University, Fort Lauderdale, FL, USA. [166]Division of Genetic Medicine, Department of Medicine, Vanderbilt University Medical Center, Nashville, TN, USA. [167]Department of Psychiatry and Behavioral Sciences, Vanderbilt University Medical Center, Nashville, TN, USA. [168]Department of Biomedical Informatics, Vanderbilt University Medical Center, Nashville, TN, USA. [169]Vanderbilt Genetics Institute, Vanderbilt University Medical Center, Nashville, USA.

