## [Peer Review File · Nature Communications]

Genome-wide identification and phenotypic characterization of seizure-associated copy number variations in 741,075 individuals.REVIEWER COMMENTS

Reviewer #1 (Remarks to the Author):

The authors describe a CNV analysis of a large cohort of humans, including healthy controls, individuals who had experienced seizures, and individuals with other neuropsychiatric phenotypes. The manuscript is well-written and contains important data that advances our understanding of the molecular mechanisms underlying seizures. I have only a couple comments:

- "Landau-Kleffner syndrome" is spelled incorrectly on line 390. In any case, it would be better to use the 2022 ILAE syndromic classification (DEE-SWAS) here and throughout the paper.
- The list of neuropsychiatric phenotypes in Supplementary Table 3 is confusing. The classification "neurodevelopmental abnormality" (which the authors note was the most frequently co-associated phenotype with the 35 identified seizure risk CNVs) seems overly general. Some other phenotypes such as "CNS hypermyelination" are extremely rare and challenging to diagnose, so that I question whether there were really 3204 such patients in the study. In other categories such as "sleep disturbance," the number of patients seems far lower than would be expected. Could the authors more explicitly state in the Methods why these specific phenotypic definitions were chosen and who was responsible for making the classification/diagnosis?

Reviewer #2 (Remarks to the Author):

Montanucci et al., analyzed a large dataset of seizure and epilepsy disorders (10,590+16,109=26,699) against a background of 492,324 population controls, which led to identification of 35 genome-wide significant loci, 32 of which were novel for seizure and epilepsy. They then explored the pleiotropy of these 35 loci in 248,752 individuals with a collection of 23 neuropsychiatric phenotypes.

Although this is a comprehensive study, some additional analyses and clarifications are needed to fully elucidate the findings.

1. Since the data was a compilation from multiple sources, each with different Quality Control procedure, how did authors account for inconsistencies in genotyping platforms, different variant calling bioinformatic tools and false positive CNVs?
2. Rare variants were identified as those in <1% of controls, as indicated above, and <1% of cases. I wonder if they have used any other datasets like 1000 Genomes CNVs and/or Gold Standard Track in Database of Genomic Variants. The advantage of using controls beyond those identified by microarrays is that such data will overcome microarray and platform specific artifacts.
3. The list of associated CNVs in the Table 1 are mostly from a few loci across the genome. For example, multiple smaller CNVs are reported from 8p23.3 and 15q11.2 but these CNVs are only have a short distance apart from each other and are from regions of the genome with high proportion of homology, segmental duplications, and LCRs. I am wondering why this is the case. Are they larger CNVs but fragmented into smaller sub CNVs due to technology, i.e., arrays?
4. Some of the 35 loci presented in the Table 1 as CNVs with significant association with seizures are not such rare in the general populations (according to the Database for Genomic Variants). For example, 15q11.2 duplications can be found in a frequency of about 0.4%-0.5%. Whereas clearly defined pathogenic CNVs such as 1q21.1 was not listed as being relevant to seizure.
5. The authors compare the results to a single previous GWAS study, identifying 32 novel loci, and confirmation of the three that were previously described. They do not compare to any of the previously known regions and/or genes that have been implicated in epilepsy to give a proper summary of what is novel in this study. The study would benefit from comparing the known epilepsy genes and known CNVs and identify which were found in the loci identified in this study and to

highlight any that were not found in this relatively large cohort. Given the combination of multiple platforms and approaches in the NDD cohorts, they should have been able to better refine the breakpoints compared to earlier studies. In addition, for the loci that have already been described in previous studies, (15q11.2, 15q13.2, 16p11.2 etc.), it would be helpful to show how much the region has been refined in this study using CNV data across such a large set of samples (including the NDD cohort).

6. CNV calling in the Epi25 cohort was done only using a single algorithm pennCNV, and although some pre/post QC was done there will still be a higher false-positive rate without independent validation using a secondary program. They control for cohorts when combining the data across the NDD cohorts, but it doesn't seem like they control for platform which may have been inherently controlled for in the bin/windowing approach to identify the significant loci, but I'm not certain.

7. In Table 2, the authors should identify if the locus listed is a known "genomic disorder" (i.e. flanked by segmental duplication with high recurrent mutation rate). This should figure into the discussion.

Reviewer #3 (Remarks to the Author):

The authors perform a meta-analysis of copy number variation with epilepsy or seizure phenotypes across the Epi25 data (16k cases, 8.5k controls) and a subset of data from Collins et al., 2022 (10.5k cases, 484k controls).

The authors mostly use methods described in Collins et al., Cell 2022 for calling CNVs (key being that the CNVs are at least 200kb in size) and phenotyping the individuals (highest resolution at human phenotype ontology [HPO] terms). They ultimately identify 35 loci, expanding the number of known loci linked to epilepsy/seizures from 3 previously known loci in epilepsy.

These loci range from several hundred kb to over a 1Mb, and include deletions and duplications. Many of them are associated with well-characterized syndromic diseases where epilepsy is a comorbidity. Many of these have been known since karyotyping and cytogenetics have been used for understanding developmental delay. As one might expect from this cursory observation and the large size of the CNVs, the loci are highly pleiotropic with strong associations with 80% exhibiting strong associations with developmental delay.

At some level, the current manuscript largely repackages associations identified in Collins et al., prior epilepsy consortia, and/or large studies of CNVs in developmental delay (e.g. Coe et al. Nat Genet 2019; Cooper et al., Nat Genet 2011, those well cataloged by DECIPHER) with a real synthesis of the results.

Regardless of this work being a meta-analysis that adds very little unique data compared to prior publications, I see this manuscript as being impactful in two ways. First, it appears to identify some novel loci in epilepsy/seizures. Second, it could serve as a resource for the epilepsy research and clinical communities. However, both are limited by the current presentation of the work.

Major concerns

- there should be a column in table 1 or early supplementary tables reporting the number of carriers & meta-analysis statistics, replication info, etc. can be included in this also
- the title refers to seizure-associated copy number variants, but this seems to overlook the fact that almost all loci identified here (and similarly almost all patients) likely have co-morbid developmental delay. This should somehow be apparent in the title. These are large pleiotropic CNVs, not precise genetic hits.
- the authors use a fine-mapping technique to elucidate the credible sets underlying each locus in Collins et al., 2022. They claim this is not within the scope of this work. I disagree, I think this work's impact for a research audience would be significantly enhanced by having an idea of how confidently the data pinpoint specific genes within these CNV loci.

- to the point above, the pathway analyses could be improved by fine-mapping the CNV hits first.
- the impact for a clinical audience could be greatly enhanced by having a summary table that describes resources for each association, e.g. at OMIM, GeneReviews, or DECIPHER wherever such associations exist
- to the point above, such a table would make it clear which (if any) loci are truly novel in association to epilepsy and/or novel over all
- how do these CNV results compare with the ILAE common variant study published on medRxiv with 29k cases and 26 loci identified? Broad overlaps with these loci should be acknowledged, if any.

Minor concerns

- related to the lack of integration & context above, here is an example: 3q29 is annotated as having no other phenotypic associations in Table 1. DECIPHER shows a microduplication and microdeletion syndrome
- in table 1, N is easy to misinterpret as number of carriers
- the 15q region is quite complex, and yields a range of known syndromes. One of them is duplication 15q syndrome, which can involve an interstitial (typically 3 copies) or isodicentric (typically 4+ copies) duplication. Can the authors say anything about loci like this beyond "there is a copy gain"? If not, they ought to clarify they mean copy loss and copy gain instead of duplication/deletion throughout the manuscript.
- are the 3 loci replicated from prior replicated in independent data? It wasn't clear from the main text.

Overall, I think the authors need to place the results in a better context both for biologists and for clinicians. They should also highlight what, if anything, is novel here compared to previous epilepsy and neurodevelopmental genetic findings.

Reviewer #1 (Remarks to the Author):

The authors describe a CNV analysis of a large cohort of humans, including healthy controls, individuals who had experienced seizures, and individuals with other neuropsychiatric phenotypes. The manuscript is well-written and contains important data that advances our understanding of the molecular mechanisms underlying seizures. I have only a couple comments:

Response 1.1: We thank the reviewer for highlighting the value of our study in improving our understanding of the genetic basis of seizure causation.

- "Landau-Kleffner syndrome" is spelled incorrectly on line 390. In any case, it would be better to use the 2022 ILAE syndromic classification (DEE-SWAS) here and throughout the paper.

Response 1.2: We corrected the typo and used the 2022 ILAE syndromic classification throughout the paper.

- The list of neuropsychiatric phenotypes in Supplementary Table 3 is confusing. The classification "neurodevelopmental abnormality" (which the authors note was the most frequently co-associated phenotype with the 35 identified seizure risk CNVs) seems overly general. Some other phenotypes such as "CNS hypermyelination" are extremely rare and challenging to diagnose, so that I question whether there were really 3204 such patients in the study. In other categories such as "sleep disturbance," the number of patients seems far lower than would be expected. Could the authors more explicitly state in the Methods why these specific phenotypic definitions were chosen and who was responsible for making the classification/diagnosis?

Response 1.3: We agree with the reviewer that the distribution of phenotypes is heterogeneous and likely due to a mix ascertained from phenotype labels from diagnostic labs and population cohorts. The acquisition of the phenotypes described in Supplementary Table 3 is the same as in Collins *et al.* (2022) (see Table S2 of Collins *et al.*, 2022 *Cell*, PMID: 35917817). Each cohort's contributors, including large clinical genetic diagnostic labs, provided the specific clinical phenotypes. Table S2 of Collins *et al.* (2022) lists 54 disease phenotypes standardized using the hierarchical structure provided by the Human Phenome Ontology database (Kohler *et al.*, 2019, PMID: 30476213). In contrast to our PheWAS analysis, Collins *et al.*, used the HPO to group people by phenotype category. We selected all seizure and neuropsychiatric phenotypes from this list of 54 phenotypes of Collins *et al.* (2022). To improve the readability and to assess our results in light of the data ascertainment, we added a description of the hierarchical organization procedure to the **Methods**. We now point out that the results from the heterogeneity analysis focus on understanding each locus' pleiotropy. For genotype-phenotype associations in individuals with seizures, the reader should focus on the PheWAS analysis (**Table 3, Fig. 2, Fig. 3, Supplementary Fig. 2A-E, Supplementary Fig. 3A-E**). Here, we performed a genotype-first approach, comparing the enrichment of >15,000 HPO terms in people with epilepsy and a specific CNV vs. epilepsy patients without CNV at that locus. The HPO term assignment was expert-curated in the deeply phenotyped individuals from the Epi25 cohort (see **Results** section: "**Characterization of the clinical subphenotypes enriched in the carriers of each seizure-associated CNV in epilepsy patients with deep phenotypes**"). Notably, this dataset generated by our team and the Epi25 consortium and the HPO PheWAS approach was used for the first time in this study.

Page 16, Methods section: “**Individuals with seizures or neuropsychiatric phenotypes - neuropsychiatric disorders cohort**”, paragraph 1, was changed from: “A large CNV dataset from individuals with a range of neuropsychiatric disorders (including seizure disorders) was aggregated from 17 different sources by Collins et al. (2022)¹. This data set also included the Epi25 cohort from our previous CNV GWAS study². In the present work, the individuals from² were excluded from the neuropsychiatric cohort for cross-disorder meta-analyses. All the considered cohorts are listed in **Supplementary Table 1**. This aggregated CNV dataset comprised 248,751 individuals affected by at least one of 24 neuropsychiatric disorders, including 10,590 individuals with seizures and 483,779 population controls”.

Changed to: “A large CNV dataset from individuals with a range of neuropsychiatric disorders (including seizure disorders) was aggregated from 17 different sources by Collins et al. (2022)⁹⁷. *The contributors of each cohort provided the specific clinical phenotypes. The aggregated individuals were grouped into 54 partially overlapping disease phenotypes standardized through the Human Phenome Ontology⁹⁸. The 54 different phenotypes of Collins et al. (2022)⁹⁷ were obtained through a recursive hierarchical clustering that defined a minimal set of nonredundant primary phenotypes, each including a minimum of >300 samples in at least three independent cohorts, >3,000 samples in total across all cohorts, and had less than 80% sample overlap with any other phenotype. Of the 54 phenotypes, we only selected neurological and psychiatric HPO-based phenotypes (N=23, excluding Seizures, **Supplementary Table 4**). The architecture of these HPO-based phenotypes allows the identification of associations at different levels, from broad to narrow phenotypes, providing the opportunity to distill between pleiotropic and specific associations.* This data set also included the Epi25 cohort from our previous CNV GWAS study²⁰. *This previous (outdated) Epi25 cohort was excluded from the neuropsychiatric cohort for cross-disorder meta-analyses in the present work.* All the considered cohorts are listed in **Supplementary Table 1**. This aggregated CNV dataset comprised 248,751 individuals affected by at least one of 24 neuropsychiatric disorders, including 10,590 individuals with seizures and 483,779 population controls.”

Page 9, we updated the Results section “**Characterization of the clinical subphenotypes enriched in the carriers of each seizure-associated CNV in epilepsy patients with deep phenotypes**” as follows: “*We performed phenome-wide association analyses for each of the 33 credible intervals identified across the 25 CNV regions to characterize the high-resolution clinical manifestations associated with each CNV. This analysis was performed on a subset of the Epi25 Collaborative cohort (Phenomic cohort, **Supplementary Table 1**) comprising 10,880 individuals with non-acquired epilepsy and deep phenotypic data (the clinical presentation of this cohort of 10,880 individuals and the frequencies of selected common and characteristic epilepsy phenotypes are provided in **Supplementary Table 5**). In the Phenomic cohort, 562 individuals (5.2%) carried at least one seizure-associated credible interval (N=498 / 4.6% carried one credible interval, N=64 / 0.6% carried 2-5 credible intervals). The most common credible interval (deletion at 2p21-p16.3) was carried by 114 (1.0%) individuals, and 18 credible intervals were found in at least 0.1% of the cohort (≥ 11 carriers). One CNV was not found (deletion at 9p24.3, containing a single credible interval). Across the 32 detected credible intervals and 1,667 annotated HPO concepts, we identified 622 nominally significant associations (two-sided Fisher's exact test, **Supplementary Table 6**). Given the large number of associations tested and that HPO annotations describing the same clinical feature at different levels of precision are highly correlated, we applied the minP step-down procedure to aid interpretation⁶¹, yielding 19 associations robust to multiple testing within each genetically defined group (minP-adjusted $P < 0.05$, **Table 3, Fig. 2, Fig. 3, and Supplementary Fig. 2A-E**).*

Carriers of deletions at 1p36.33 [0.91-1.51Mb] (N=25, 0.23% of the Phenomic cohort), 1p36.33 [2.02-2.49Mb] (N=17, 0.16%), or 15q12-q13.1 (N=4, 0.037%), and carriers of duplications at

15q11.2-q13.3 (N=46, 0.42%) were enriched with clinical features suggestive of developmental and epileptic encephalopathies, such as epileptic spasms and tonic seizures, epileptic encephalopathy, and other neurodevelopmental disorders, sudden unexpected death in epilepsy, and morphological abnormalities⁶². Features characterizing genetic generalized epilepsy were associated with deletions at 2p21-p16.3 (N=114, 1.05%, generalized tonic-clonic and absence seizures), 15q11.2 (N=56, 0.52%, eyelid myoclonia and absence seizures), 16p13.11 (N=42, 0.39%, generalized tonic-clonic seizures), 15q13.2-q13.3 (N=24, 0.22%, absence seizures) or 22q11.21 [20.65-21.54Mb] (N=6, 0.055%, juvenile myoclonic epilepsy-like features). Duplications at 16p11.2 (N=8, 0.074%) were associated with non-epileptic seizures comorbid with epilepsy (OR=81.5, unadjusted P=4.82x10⁻⁴, minP-adjusted P=0.0297), and showed a nonsignificant greater frequency of microcephaly (OR=31.5, unadjusted P=3.62x10⁻², minP-adjusted P=0.92) that replicates the mirror microcephaly/macrocephaly phenotype of the reciprocal 16p11.2 CNVs⁶³.

We interrogated the phenotypic annotations of CNV carriers regarding the candidate genes prioritized in our fine-mapping analysis. MSH2 was prioritized as the candidate gene for the most common deletion in the Phenomic cohort (2p21-p16.3). Heterozygous loss of function variants of the haploinsufficient gene MSH2 cause Lynch syndrome⁶⁴, and complete knockout of paralog Msh2 in Ccm1^{+/-} mice causes multiple cavernoma through a presumed second hit⁶⁵. We found that carriers had a nonsignificant greater frequency of neoplasms (OR=2.35, unadjusted P=2.49x10⁻², minP-adjusted P=1.00) and cerebral cavernomata (OR=5.23, unadjusted P=6.58x10⁻⁴, minP-adjusted P=0.157) than non-carriers. Carriers of the 1p36.33 [2.02-2.49Mb] deletion overlapping the gene SKI had features (hypotonia, talipes equinovarus, abnormalities of the globe and nose, osteoporosis, global developmental delay, and Chiari malformation) concordant to the Shprintzen-Goldberg craniosynostosis syndrome caused by SKI⁴¹. All 15 individuals with duplication of 9q34.3 had focal-onset seizures that were rarely drug-resistant, without any individual annotated with a neurodevelopmental disorder or polymicrogyria despite the presence of the GRIN1, which can cause polymicrogyria when affected by gain-of-function variants⁴⁷. Sixteen of 24 individuals carrying deletions at 15q13.3 [31.06-32.51Mb] had generalized absence seizures (OR=10.5, unadjusted P=3.70x10⁻⁸, minP-adjusted P=1x10⁻⁵), in line with the primary seizure type reported in carriers of the 15q13.3 deletion⁶⁶. Finding generalized myoclonic seizures in half of the carriers of the 22q11.2 [19.67-19.96Mb] deletion further confirmed TBX1⁶⁷, the known causal gene for the 22q11.21 deletion/DiGeorge syndrome⁴⁸. Features suggestive of juvenile myoclonic epilepsy were also found among six people carrying deletions overlapping with the second credible interval at 22q11.2 [20.65-21.54Mb] spanning the Noonan syndrome 10 locus containing in which a single individual was reported with seizures⁴⁹. However, none of these six individuals had annotations beyond seizures and electroencephalography phenotypes that would support a multisystemic syndrome.

Finally, clinicians may want to know the frequency of broad clinical features among carriers of the CNV identified in their patients to improve the interpretation of its clinical relevance and to facilitate genetically stratified prognostication. Therefore, we prioritized 17 common, conceptually broad, and important epilepsy manifestations and comorbidities for visualization, including the co-occurrence of generalized-onset and focal-onset seizures that characterizes the combined generalized and focal epilepsy type⁶² (**Fig 3 and Supplementary Fig. 3A-E**). The most common CNV, deletion at 2p21-p16.3, appeared to modestly increase the likelihood of a carrier having generalized epilepsy. However, a few CNVs had a profile dominated by core electroclinical features of generalized (for example, deletions at 15q13.2-15q13.3) or focal epilepsy (duplications at 9q34.3 [139.89-140.12Mb]), with comorbid features being rare. Conversely, carriers of other CNVs had relatively high frequencies of neurodevelopmental disorders, epileptic spasms, and drug resistance suggestive of developmental and epileptic encephalopathy (deletions at 1p36.33). However, no CNV was found exclusively in people with a particular seizure type, and carriers of some CNVs appeared to have broad clinical features at

frequencies indistinguishable from the cohort's baseline (duplications at 19p13.3), suggesting some generic contribution to epilepsy risk across epilepsy types."

We have restructured the previous first paragraph of the **Discussion** to reflect on the new fine-mapping results, the overlap between seizure-associated CNVs and previously reported clinical phenotypes and the CNV-HPO analysis in the Phenome cohort.

Page 11, **Discussion**, the first paragraph was removed and replaced with: "In this study, we leveraged a substantial increase in sample size to identify novel seizure-associated CNVs when jointly analyzing 26,699 individuals with various types of seizure disorders against 492,324 population controls. We identified 25 novel loci with genome-wide significance for seizure disorders. *In addition, all three previously reported epilepsy-associated loci at genome-wide level maintained genome-wide significance for seizure disorders in our meta-analysis that included the epilepsy cohort from the previous study²⁰. Of the 25 seizure-associated loci, 16 were previously implicated in neurological and psychiatric disorders, including epilepsy. Five were flanked by known segmental duplications (SDs) or low copy number repeats (LCRs). Of note, our fine-mapping analysis confirmed the first and third known critical regions for seizures within the phenotype spectrum of the 1p36 deletion syndrome³⁸, TBX1 as the (known) causal gene for the 22q11.21 deletion/DiGeorge syndrome⁴⁸, and suggested the SNRPN promoter/exon 1 region as the causal element for seizures within the larger BP2-BP3 15q11.2-q13 duplication region.*

In a high-resolution phenomic analysis in a subset of 10,880 individuals from our cohort with epilepsy (from the Epi25 cohort), we identified 622 suggestive and 19 significant clinical associations informative for epileptologists among CNV carriers. This observation indicates that beyond contributing to the generic risk of seizures, several CNVs contribute to specific epilepsy types. Carriers of some CNVs tended to have features typical of developmental and epileptic encephalopathies with neurodevelopmental and non-seizure phenotypes. Conversely, carriers of others had phenotypes restricted to the core epileptic features of seizures and electroencephalographic abnormalities (both generalized and focal). Interestingly, reciprocal CNVs involving 22q11.21 seemed to produce opposite epilepsy types, with deletion and duplication carriers tending to have generalized and focal epilepsies, respectively. Dose-dependent effects of KLHL22 on DEPDC5 degradation are a possible explanation⁶⁸. Overall, the high degree of pleiotropy among seizure-associated CNVs implies that these CNVs likely impair neurodevelopmental processes rather generically and contribute to the broad spectrum of neurodevelopmental disorders. According to the oligo-/polygenic inheritance model, CNVs may interact with the genetic background or environmental factors to generate the final disease phenotype. Interaction between CNVs and the polygenic background was recently demonstrated in carriers of the schizophrenia-associated 22q11.2 deletion⁶⁹. Support for an oligogenic-CNV disorder model was also recently published⁷⁰."

Reviewer #2 (Remarks to the Author):

Montanucci et al., analyzed a large dataset of seizure and epilepsy disorders (10,590+16,109=26,699) against a background of 492,324 population controls, which led to identification of 35 genome-wide significant loci, 32 of which were novel for seizure and epilepsy. They then explored the pleiotropy of these 35 loci in 248,752 individuals with a collection of 23 neuropsychiatric phenotypes.

Although this is a comprehensive study, some additional analyses and clarifications are needed to fully elucidate the findings.

1. Since the data was a compilation from multiple sources, each with different Quality Control procedure, how did authors account for inconsistencies in genotyping platforms, different variant calling bioinformatic tools and false positive CNVs?

Response 2.1: Yes, we performed an extensive CNV quality control to account for various potential technical and other biases. We also agree with the reviewer that the procedure of data harmonization from different sources was not explicitly detailed in this manuscript. We improved the Quality Control procedure description, as detailed in Collins et al. (2022) (PMID: 35917817), for the Neuropsychiatric disorders cohort, which is data compilation from multiple sources. We also improved our description of the Quality Control procedure for the Epi25 Collaborative cohort aimed to minimize batch differences to the Neuropsychiatric cohort while retaining high sensitivity informed by visual inspection of probe-level intensity plots of all significant CNVs. (See also **Response 2.2** for additional data quality analyses based on suggestions by the reviewer.)

Page 17, the entire Methods section “**Quality control - neuropsychiatric disorders cohort**”, was replaced with the following text: “*The CNV harmonization procedure for the Neuropsychiatric cohort is described in the Supplementary Materials of Collins et al. (2022)⁹⁷ and included following steps: 1) CNV calls of the same type (deletion or duplication) were merged if their breakpoints were within $\pm 25\%$ of the size of their corresponding original CNV calls to avoid over-segmentation of large CNV calls; 2) CNVs not mapped to autosomes from the primary hg19 assembly were excluded; 3) Only CNVs between $\geq 100\text{kb}$ and $\leq 20\text{Mb}$ in size were considered; 4) CNVs that matched reported common CNVs (allele frequency $> 1\%$) in three independent CNV reference catalogs derived from genome sequencing (Abel et al., 2020⁹⁹; Collins et al., 2020¹⁰⁰; Sudmant et al., 2015⁸¹) were excluded; 5) CNVs that overlapped other CNVs in $\geq 1\%$ of samples within the same dataset or in any of the other array CNV datasets were excluded to remove potential platform specific artifacts; 6) We excluded all CNVs with $\geq 30\%$ overlap with somatic hypermutable sites, segmental duplications, simple/low-complexity/satellite repeats, or N-masked bases of the hg19 reference assembly.*”

Page 16, Methods section: “**CNV calling and quality control - Epi25 Collaborative**”, the end of the paragraph was changed from: “For the post-CNV calling QC, we merged adjacent CNVs if the number of intervening markers between them was $< 20\%$ of the total number when both segments were combined. CNVs supported by < 20 markers, $< 20\text{kb}$ long, and with a SNP density < 0.0001 were excluded from subsequent analyses”.

Changed to: “*The post-CNV calling QC included the following steps: 1) CNV calls of the same type (deletion or duplication) were merged if the number of SNP/intensity markers between them was $< 20\%$ of the total number when both segments were combined; 2) CNVs supported by < 20 markers, $< 20\text{kb}$ long, and with a SNP density < 0.0001 were excluded from subsequent analyses; 3) CNVs that overlapped other CNVs in $\geq 1\%$ of all samples within the Epi25 dataset were excluded to remove*”

potential platform-specific artifacts, 4) CNVs with >50% overlap with telomeric, centromeric, and immunoglobulin regions of the hg19 reference assembly were excluded; 5) CNVs with ≥50% overlap with reported common CNVs (allele frequency >1%) in two independent CNV reference catalogs (DGV Gold Standard Dataset⁹⁵; DECIPHER Population Copy-Number Variation Frequencies⁹⁶) were excluded. Finally, the probe-level intensity plots of all CNVs supporting the seizure-associated regions (**Table 1**) were visually inspected to exclude any remaining artifacts. The DGV Gold Standard and DECIPHER Population frequencies of the remaining CNVs are given in **Supplementary Table 7.**”

We added **Supplementary Table 7:** CNV frequencies in the cases & controls of the meta-analysis, DGV Gold Standard, and DECIPHER databases.

Cytoband	CNV type	Hg19 Start (Mb)	Hg19 End (Mb)	Cases carrier frequency [%]	Control carrier frequency [%]	Frequency in DGV Gold Standard [%]	Frequency in DECIPHER [%]
1p36.33	DEL	0.91	1.51	0.086	0.015	0	0
1p36.33	DEL	2.02	2.49	0.146	0.003	0	0
1q44	DEL	245.29	245.86	0.041	0.002	0	0
2p21-p16.3	DEL	47.5	47.85	0.678	0.002	0	0
2q13	DUP	110.77	111.06	0.139	0.108	0	0.27
3q29	DEL	195.76	196.24	0.034	0.002	0	0.12
8p23.3-p23.2	DEL	0.4	5.47	0.067	0.010	0	0
9p24.3	DEL	0.33	0.56	0.049	0.007	0	0.02
9q34.3	DUP	139.21	140.12	0.315	0.003	0	0
10q26.3	DEL	133.41	134.68	0.030	0.002	0	0
15q11.2	DEL	22.74	23.28	0.689	0.284	0.41	0.83
15q11.2-q13.3	DUP	22.98	32.15	0.258	0.005	0	0.12
15q12-q13.1	DEL	27.93	28.23	0.097	0.008	0	0
15q13.2-q13.3	DEL	31.06	32.51	0.243	0.012	0	0.02
16p13.3	DUP	0.6	0.89	0.573	0.009	0	0
16p13.11	DEL	15.42	16.35	0.363	0.031	0.03	0.07
16p12.2	DEL	21.88	22.5	0.191	0.054	0.09	0.08
16p11.2	DEL	29.56	30.19	0.165	0.024	0.05	0.12
16p11.2	DUP	29.87	30.19	0.127	0.026	0	0.03
17q12	DUP	34.76	36.25	0.187	0.014	0.02	0.12
17q21.31	DEL	41.08	41.45	0.461	0.004	0	0
19p13.3	DUP	1.04	1.34	0.427	0.003	0	0.08
20q13.33	DUP	62	62.35	0.479	0.006	0	0
22q11.21	DUP	18.99	21.54	0.199	0.067	0	0
22q11.21	DEL	18.99	21.54	0.120	0.009	0	0.19

The frequencies in the DGV Gold Standard and DECIPHER Population databases are given for CNVs with ≥50% overlap with the seizure-associated CNV regions.

2. Rare variants were identified as those in <1% of controls, as indicated above, and <1% of cases. I wonder if they have used any other datasets like 1000 Genomes CNVs and/or Gold Standard Track in Database of Genomic Variants. The advantage of using controls beyond those identified by microarrays is that such data will overcome microarray and platform specific artifacts.

Response 2.2: We agree with the reviewer that filtering the CNVs using external datasets is a good strategy. We have added similar filtering to the Epi25 Collaborative cohort, excluding one of the previously reported seizure-associated regions (Duplication at 1p36.33). To improve the ability of the readers to interpret the data behind the identified association, we added the new **Supplementary Table 7** detailing the frequencies of CNVs with >50% overlap with the identified seizure-associated

CNVs in the DGV Gold Standard and the DECIPHER Population Copy-Number Variation datasets. The updated **Methods** sections for both cohorts are given in **Response 2.1**.

Please see the updated Method sections “**Quality control - neuropsychiatric disorders cohort**” and “**CNV calling and quality control - Epi25 Collaborative**” in **Response 2.1**.

Please see the new **Supplementary Table 7** showing the DGV Gold Standard and DECIPHER Population CNV frequencies in **Response 2.1**.

3. The list of associated CNVs in the Table 1 are mostly from a few loci across the genome. For example, multiple smaller CNVs are reported from 8p23.3 and 15q11.2 but these CNVs are only have a short distance apart from each other and are from regions of the genome with high proportion of homology, segmental duplications, and LCRs. I am wondering why this is the case. Are they larger CNVs but fragmented into smaller sub CNVs due to technology, i.e., arrays?

Response 2.3: We agree with the reviewer that the apparent clustering of multiple smaller CNVs at specific loci is an interesting observation. Deconvoluting the CNV association signal at these complex regions has many challenges due to the genomic complexity of these loci that can have population-specific (<https://doi.org/10.1101/2022.07.09.499321>) or even person-specific configurations that even classical whole genome sequencing technology cannot resolve (PMID: 35357919). We added this information to the limitations section of the **Discussion**.

However, methodologies have been developed to refine the signal we observe and prioritize the association signal at a given locus. In the revised version of our manuscript, we incorporated the suggestion of the reviewers to improve the fine-mapping procedure, leading to revised start/stop positions of the associated regions and an updated number of 25 genome-wide significantly associated CNV regions. Briefly, we used a Bayesian algorithm (PMID: 18642345) to identify the minimal interval(s) that contained the causal element(s) with 95% confidence, as well as the entire “merged CNV region” that is supported by genome-wide association signals as in Collins et al. (2022) (PMID: 35917817). The resulting start/stop coordinates of the associated regions were accordingly updated in **Table 1**, and we report the coordinates of both the merged region and the credible intervals. The main changes are, for example, the four previously reported deletions at 8p23.2 now merged into one single seizure-associated region with three (credible) intervals that likely contain the causal element(s) with 95% confidence. Similarly, the previously reported duplications at 15q11-q13 were merged into one seizure-associated region as defined by the genome-wide association signals, with one credible interval within the known BP3-BP3 15q11-q13 microduplication syndrome region (PMID: 26022164). The identified credible interval is narrow and contains the imprinted *SNRPN* locus (promoter and first exons) to regulate the imprinting of the critical region for Prader-Willi syndrome (PMID: 9973278, 10802660). We updated the HPO enrichment / PheWAS analysis to be also based on the 33 identified credible intervals of each genome-wide significantly associated CNV region. We updated all corresponding **Methods**, **Results**, and **Discussion** sections.

Page 18, we changed the Methods section “**Meta-analysis**” to “**Meta-analysis and fine-mapping**” with the following addition at the end of the section: “*We then used a Bayesian algorithm¹⁰² to identify the minimal credible interval(s) that contained the causal element(s) or genes with 95% confidence, as in Collins et al. (2022)⁹⁷. Finally, we explored the known biological function of all genes within the credible intervals and performed pathway analyses using Enrichr^{103,104}. All resources used*

to investigate the knowledge basis of all seizure-associated CNV regions are described in Supplementary Table 8.”

Page 5, we changed the Results section “**Discovery of 35 genome-wide significant seizure-associated CNVs**” to “**Discovery of 25 genome-wide significant seizure-associated CNV regions**” and modified it as follows: “We performed a meta-analysis of 16,109 individuals with epilepsy and 8,545 population controls (the Epi25 Collaborative cohort) with 10,590 individuals with seizures (not explicitly meeting diagnostic criteria for epilepsy) and 483,779 population controls, derived from an aggregated CNV dataset of 17 cohorts (neuropsychiatric disorders cohort) (see all cohorts of this study in **Supplementary Table 1**). The genome was scanned using 267,237 genomic segments of 200kb size in a 10kb sliding window approach²⁶. After applying Bonferroni correction of the threshold for a significant association in the meta-analysis *and fine-mapping*, we identified 25 loci associated with seizures at genome-wide significance ($P \leq 3.74 \times 10^{-6}$). All 25 loci are shown in **Fig. 1** and detailed in **Table 1**. The 25 identified loci included 15 deletion CNVs (size range: 230kb to 5Mb) and ten duplication CNVs (size range: 290kb to 8.9Mb). *All the genome-wide associated deletions found in this study consisted of the loss of one copy, while all duplications consisted of the gain of one copy. Three of the 25 seizure-associated loci (15q11.2-q13.3 dup, 15q13.2-q13.3 del, 16p13.11 del) had previous genome-wide statistical support for an association with epilepsy from our previous study²⁰ that included 40% of the individuals with seizures of this study.* All other identified CNVs (22/25, 88%) represent new genome-wide significant loci for seizures, *with 10/22 (59%) loci previously implicated in neurological and psychiatric disorders, 6/22 (23%) specifically in epilepsy by studies without genome-wide statistical support, 2/22 (9%) reported in individuals without neurological or psychiatric disorders, and 4/22 (18%) not previously reported regions.* We detailed in **Table 2** all commonly reported disease phenotypes for the 25 identified seizure-associated loci. *Our meta-analysis in seizure disorders was likely not powered enough to identify some of the known CNVs implicated in epilepsy (without genome-wide statistical support) associated with seizures (e.g., 1q21.1 del/dup).* Reciprocal CNVs, defined by deletions and duplications associated with seizures involving overlapping genomic segments, were found at 15q11.2, 16p11.2, and 22q11.21. *No overlap existed between the seizure-associated CNV regions identified in this study and the most recent SNP-based GWAS study in epilepsy²⁷.*”

Updated **Table 1**: Genome-wide significantly associated CNV regions and credible intervals.

Cytoband	CNV type	Hg19 Start (Mb)	Hg19 End (Mb)	Lowest P-value in region	OR [95% CI]	Credible interval containing the causal element/gene with 95% confidence	Associations with neuropsychiatric phenotypes [N]	Highest odds ratio in neuropsychiatric disorder/seizure meta-analyses (95% CI)
1p36.33	DEL	0.91	1.51	4.65E-09	23 (10-54)	910000-1510000	5	11 (5-24) CNS abnormality
1p36.33	DEL	2.02	2.49	1.95E-07	44 (14-141)	2020000-2490000	3	48 (14-170) Abnormalities of Cognition
1q44	DEL	245.29	245.86	6.59E-07	41 (13-133)	245290000-245860000	6	62 (19-207) Abnormal brain morphology
2p21-p16.3	DEL	47.5	47.85	3.00E-13	12 (7-22)	47500000-47850000	23	13 (8-24) Behavioral abnormalities
2q13	DUP	110.77	111.06	1.33E-06	3 (2-5)	110770000-111060000	6	24 (9-63) Hyperactivity
3q29	DEL	195.76	196.24	2.01E-07	40 (13-122)	195760000-196240000	0	no significant association signal
8p23.3-p23.2	DEL	0.4	5.47	1.83E-08	12 (6-25)	400000-610000 3040000-3780000 4810000-5470000	4	16 (7-39) Intellectual disability
9p24.3	DEL	0.33	0.56	1.08E-07	13 (6-29)	330000-560000	1	7 (4-12) Neurodevelopmental abnormality
9q34.3	DUP	139.21	140.12	1.67E-06	12 (5-27)	139210000-139590000 139890000-140120000	21	12 (6-25) Schizophrenia
10q26.3	DEL	133.41	134.68	3.16E-07	40 (13-125)	133410000-133740000 134370000-134680000	2	32 (10-107) Sleep disorder
15q11.2	DEL	22.74	23.28	1.02E-13	3 (2-3)	22740000-23280000	12	3 (2-4) Abnormalities of Cognition
15q11.2-q13.3	DUP	22.98	32.15	3.68E-19	27 (14-52)	24750000-25080000	23	33 (16-67) Intellectual disability
15q12-q13.1	DEL	27.93	28.23	9.85E-07	14 (6-34)	27930000-28230000	3	24 (11-52) Intellectual disability
15q13.2-q13.3	DEL	31.06	32.51	6.71E-16	14 (8-24)	31060000-32510000	17	16 (6-44) Sleep disorder
16p13.3	DUP	0.6	0.89	1.98E-13	9 (5-14)	600000-890000	23	12 (6-26) Schizophrenia
16p13.11	DEL	15.42	16.35	3.53E-17	9 (6-14)	15420000-16350000	18	13 (4-44) CNS atrophy
16p12.2	DEL	21.88	22.5	1.14E-06	4 (2-5)	21880000-22500000	6	4 (2-6) Hyperactivity
16p11.2	DEL	29.56	30.19	1.25E-11	9 (5-15)	29560000-30190000	11	13 (8-21) Intellectual disability
16p11.2	DUP	29.87	30.19	2.45E-08	6 (3-10)	29870000-30190000	12	11 (5-24) Sleep disorder
17q12	DUP	34.76	36.25	1.12E-15	15 (9-27)	34760000-35510000 35960000-36250000	14	10 (5-20) Abnormal brain morphology
17q21.31	DEL	41.08	41.45	1.98E-08	5 (3-9)	41080000-41450000	20	6 (4-10) Abnormality of the nervous system
19p13.3	DUP	1.04	1.34	2.85E-10	7 (4-12)	1040000-1340000	23	7 (4-14) Schizophrenia
20q13.33	DUP	62	62.35	7.12E-11	6 (4-10)	62000000-62350000	23	8 (5-13) CNS abnormality
22q11.21	DUP	18.99	21.54	8.73E-11	5 (3-7)	18990000-19370000 20200000-21540000 18990000-19400000	9	5 (3-7) Hyperactivity
22q11.21	DEL	18.99	21.54	1.57E-08	26 (10-65)	19670000-19960000 20650000-21540000	12	26 (9-70) Central motor dysfunction

Column 1: Cytoband localization of the CNV. Column 2: CNV type, either deletion (DEL) or duplication (DUP). Columns 3 and 4: Genomic coordinates (in Mb) on the GRCh37 reference genome of the start and end position of the merged CNV region that is supported by genome-wide association signals. Columns 5 and 6: Lowest *P*-values in each CNV region and corresponding odds ratios (with 95% confidence interval) of the genome-wide CNV meta-analysis in 25,345 individuals with seizures and 492,324 controls. *Column 7: GRCh37 coordinates of the credible interval(s) that contained the causal element(s) with 95% confidence.* Column 8: N=Number of neuropsychiatric disorders that showed a significant genome-wide CNV association in this locus. Column 9: Highest odds ratio for each locus in any of the 23 cross-disorder meta-analyses. Deletions are shown in rows with a light blue background, and duplications are shown in rows with white background.

Page 9, we updated the Results section “**Characterization of the clinical subphenotypes enriched in the carriers of each seizure-associated CNV in epilepsy patients with deep phenotypes**” as follows: “*We performed phenome-wide association analyses for each of the 33 credible intervals identified across the 25 CNV regions to characterize the high-resolution clinical manifestations associated with each CNV. This analysis was performed on a subset of the Epi25 Collaborative cohort (Phenomic cohort, **Supplementary Table 1**) comprising 10,880 individuals with non-acquired epilepsy and deep phenotypic data (the clinical presentation of this cohort of 10,880 individuals and the frequencies of selected common and characteristic epilepsy phenotypes are provided in **Supplementary Table 5**). In the Phenomic cohort, 562 individuals (5.2%) carried at least one seizure-associated credible interval (N=498 / 4.6% carried one credible interval, N=64 / 0.6% carried 2-5 credible intervals). The most common credible interval (deletion at 2p21-p16.3) was carried by 114 (1.0%) individuals, and 18 credible intervals were found in at least 0.1% of the cohort (≥11 carriers). One CNV was not found (deletion at 9p24.3, containing a single credible interval). Across the 32 detected credible intervals and 1,667 annotated HPO concepts, we identified 622 nominally significant associations (two-sided Fisher's exact test, **Supplementary Table 6**). Given the large number of associations tested and that HPO annotations describing the same clinical feature at*

different levels of precision are highly correlated, we applied the *minP* step-down procedure to aid interpretation⁶¹, yielding 19 associations robust to multiple testing within each genetically defined group (*minP*-adjusted $P < 0.05$, **Table 3**, **Fig. 2**, **Fig. 3**, and **Supplementary Fig. 2A-E**).

Carriers of deletions at 1p36.33 [0.91-1.51Mb] ($N=25$, 0.23% of the Phenomic cohort), 1p36.33 [2.02-2.49Mb] ($N=17$, 0.16%), or 15q12-q13.1 ($N=4$, 0.037%), and carriers of duplications at 15q11.2-q13.3 ($N=46$, 0.42%) were enriched with clinical features suggestive of developmental and epileptic encephalopathies, such as epileptic spasms and tonic seizures, epileptic encephalopathy, and other neurodevelopmental disorders, sudden unexpected death in epilepsy, and morphological abnormalities⁶². Features characterizing genetic generalized epilepsy were associated with deletions at 2p21-p16.3 ($N=114$, 1.05%, generalized tonic-clonic and absence seizures), 15q11.2 ($N=56$, 0.52%, eyelid myoclonia and absence seizures), 16p13.11 ($N=42$, 0.39%, generalized tonic-clonic seizures), 15q13.2-q13.3 ($N=24$, 0.22%, absence seizures) or 22q11.21 [20.65-21.54Mb] ($N=6$, 0.055%, juvenile myoclonic epilepsy-like features). Duplications at 16p11.2 ($N=8$, 0.074%) were associated with non-epileptic seizures comorbid with epilepsy (OR=81.5, unadjusted $P=4.82 \times 10^{-4}$, *minP*-adjusted $P=0.0297$), and showed a nonsignificant greater frequency of microcephaly (OR=31.5, unadjusted $P=3.62 \times 10^{-2}$, *minP*-adjusted $P=0.92$) that replicates the mirror microcephaly/macrocephaly phenotype of the reciprocal 16p11.2 CNVs⁶³.

We interrogated the phenotypic annotations of CNV carriers regarding the candidate genes prioritized in our fine-mapping analysis. *MSH2* was prioritized as the candidate gene for the most common deletion in the Phenomic cohort (2p21-p16.3). Heterozygous loss of function variants of the haploinsufficient gene *MSH2* cause Lynch syndrome⁶⁴, and complete knockout of paralog *Msh2* in *Ccm1*^{+/-} mice causes multiple cavernoma through a presumed second hit⁶⁵. We found that carriers had a nonsignificant greater frequency of neoplasms (OR=2.35, unadjusted $P=2.49 \times 10^{-2}$, *minP*-adjusted $P=1.00$) and cerebral cavernomata (OR=5.23, unadjusted $P=6.58 \times 10^{-4}$, *minP*-adjusted $P=0.157$) than non-carriers. Carriers of the 1p36.33 [2.02-2.49Mb] deletion overlapping the gene *SKI* had features (hypotonia, talipes equinovarus, abnormalities of the globe and nose, osteoporosis, global developmental delay, and Chiari malformation) concordant to the Shprintzen-Goldberg craniosynostosis syndrome caused by *SKI*⁴¹. All 15 individuals with duplication of 9q34.3 had focal-onset seizures that were rarely drug-resistant, without any individual annotated with a neurodevelopmental disorder or polymicrogyria despite the presence of the *GRIN1*, which can cause polymicrogyria when affected by gain-of-function variants⁴⁷. Sixteen of 24 individuals carrying deletions at 15q13.3 [31.06-32.51Mb] had generalized absence seizures (OR=10.5, unadjusted $P=3.70 \times 10^{-8}$, *minP*-adjusted $P=1 \times 10^{-5}$), in line with the primary seizure type reported in carriers of the 15q13.3 deletion⁶⁶. Finding generalized myoclonic seizures in half of the carriers of the 22q11.2 [19.67-19.96Mb] deletion further confirmed *TBX1*⁶⁷, the known causal gene for the 22q11.21 deletion/DiGeorge syndrome⁴⁸. Features suggestive of juvenile myoclonic epilepsy were also found among six people carrying deletions overlapping with the second credible interval at 22q11.2 [20.65-21.54Mb] spanning the Noonan syndrome 10 locus containing in which a single individual was reported with seizures⁴⁹. However, none of these six individuals had annotations beyond seizures and electroencephalography phenotypes that would support a multisystemic syndrome.

Finally, clinicians may want to know the frequency of broad clinical features among carriers of the CNV identified in their patients to improve the interpretation of its clinical relevance and to facilitate genetically stratified prognostication. Therefore, we prioritized 17 common, conceptually broad, and important epilepsy manifestations and comorbidities for visualization, including the co-occurrence of generalized-onset and focal-onset seizures that characterizes the combined generalized and focal epilepsy type⁶² (**Fig 3** and **Supplementary Fig. 3A-E**). The most common CNV, deletion at 2p21-p16.3, appeared to modestly increase the likelihood of a carrier having generalized epilepsy. However, a few CNVs had a profile dominated by core electroclinical features of generalized (for

example, deletions at 15q13.2-15q13.3) or focal epilepsy (duplications at 9q34.3 [139.89-140.12Mb]), with comorbid features being rare. Conversely, carriers of other CNVs had relatively high frequencies of neurodevelopmental disorders, epileptic spasms, and drug resistance suggestive of developmental and epileptic encephalopathy (deletions at 1p36.33). However, no CNV was found exclusively in people with a particular seizure type, and carriers of some CNVs appeared to have broad clinical features at frequencies indistinguishable from the cohort's baseline (duplications at 19p13.3), suggesting some generic contribution to epilepsy risk across epilepsy types."

We have restructured the previous first paragraph of the **Discussion** to reflect on the new fine-mapping results, the overlap between seizure-associated CNVs and previously reported clinical phenotypes and the CNV-HPO analysis in the Phenome cohort.

Page 11, **Discussion**, the first paragraph was removed and replaced with: "In this study, we leveraged a substantial increase in sample size to identify novel seizure-associated CNVs when jointly analyzing 26,699 individuals with various types of seizure disorders against 492,324 population controls. We identified 25 novel loci with genome-wide significance for seizure disorders. *In addition, all three previously reported epilepsy-associated loci at genome-wide level maintained genome-wide significance for seizure disorders in our meta-analysis that included the epilepsy cohort from the previous study²⁰. Of the 25 seizure-associated loci, 16 were previously implicated in neurological and psychiatric disorders, including epilepsy. Five were flanked by known segmental duplications (SDs) or low copy number repeats (LCRs). Of note, our fine-mapping analysis confirmed the first and third known critical regions for seizures within the phenotype spectrum of the 1p36 deletion syndrome³⁸, TBX1 as the (known) causal gene for the 22q11.21 deletion/DiGeorge syndrome⁴⁸, and suggested the SNRPN promoter/exon 1 region as the causal element for seizures within the larger BP2-BP3 15q11.2-q13 duplication region.*

In a high-resolution phenomic analysis in a subset of 10,880 individuals from our cohort with epilepsy (from the Epi25 cohort), we identified 622 suggestive and 19 significant clinical associations informative for epileptologists among CNV carriers. This observation indicates that beyond contributing to the generic risk of seizures, several CNVs contribute to specific epilepsy types. Carriers of some CNVs tended to have features typical of developmental and epileptic encephalopathies with neurodevelopmental and non-seizure phenotypes. Conversely, carriers of others had phenotypes restricted to the core epileptic features of seizures and electroencephalographic abnormalities (both generalized and focal). Interestingly, reciprocal CNVs involving 22q11.21 seemed to produce opposite epilepsy types, with deletion and duplication carriers tending to have generalized and focal epilepsies, respectively. Dose-dependent effects of KLHL22 on DEPDC5 degradation are a possible explanation⁶⁸. Overall, the high degree of pleiotropy among seizure-associated CNVs implies that these CNVs likely impair neurodevelopmental processes rather generically and contribute to the broad spectrum of neurodevelopmental disorders. According to the oligo-/polygenic inheritance model, CNVs may interact with the genetic background or environmental factors to generate the final disease phenotype. Interaction between CNVs and the polygenic background was recently demonstrated in carriers of the schizophrenia-associated 22q11.2 deletion⁶⁹. Support for an oligogenic-CNV disorder model was also recently published⁷⁰."

Page 14, Limitation section of the **Discussion**, we added: "*Among the 25 identified CNVs, deletions ranged from 230kb to 5Mb and duplications from 290kb to 9Mb, affecting 14.2 genes on average.* CNV breakpoints in the current study are estimated from genotyped SNPs around the actual breakpoint. These breakpoint estimates are limited by the resolution of the genotyping platform used to call the CNVs. In fact, microarrays have many technical limitations, such as poor breakpoint resolution and limited sensitivity for small CNVs⁸¹. Newer technologies like whole-genome sequencing

(WGS) will enable the assessment of a more comprehensive array of rare variants, including balanced rearrangements, small (exonic) CNVs⁸², short tandem repeats, and other structural variants⁸³. *However, some genomic regions harbor complex deletion/duplication/inversion rearrangements (e.g., 22q11.21⁸⁴, 15q11.2⁸⁵) that can even show population stratification (e.g., 16p11.2⁸⁶). More accurate and complete (pangenome) references will be needed to determine the exact breakpoints of such complex rearrangements^{87,88}, even in the case of sequencing-based CNVs discovery.* Lastly, we performed joint epilepsy/seizures and cross-disorder meta-analyses in individuals with minimal clinical information.”

Updated **Fig. 2** shows the proportion of carriers and non-carriers annotated with each HPO concept for the 19 significant CNV-HPO associations in the Phenome cohort.

Fig. 2: Genotype-first phenomic analysis in 10,880 individuals with detailed clinical data. For each CNV, the proportion of carriers and non-carriers annotated with each HPO concept is plotted. Those above the diagonal were enriched among carriers, and those below were depleted. *Selected phenotypic concepts are labeled.* Full results for all associations reaching raw $p < 0.05$ are provided in **Supplementary Table 6**. SUDEP = sudden unexpected death in epilepsy, CNS = central nervous system, EEG = electroencephalogram.

Updated **Fig. 3** shows the clinical signatures of all CNVs with significant CNV-HPO associations in the Phenome cohort.

Fig. 3: Summary of clinical signatures of CNVs in a deeply phenotyped epilepsy cohort. Dots represent the prior frequency of each broad clinical manifestation in the Phenomic cohort. Binomial distribution-derived 95% confidence interval bars are given for the frequency of each phenotype among carriers of the CNV. "Craniofacial or skeletal dysmorphism" includes individuals with either "Abnormality of the head [HP:0000234]" (which excludes isolated brain structural abnormalities) or "Abnormal skeletal morphology [HP:0011842]". "Motor, movement or muscular disorder" includes individuals with any of "Abnormal central motor function [HP:0011442]", "Abnormality of movement [HP:0100022]" or "Abnormality of the musculature [HP:0003011]", but not "Motor delay [HP:0001270]", which is included in "Neurodevelopmental abnormality". While "Neurodevelopmental abnormality" includes those with "Intellectual disability", the latter is shown in addition as it is a neurodevelopmental outcome with particularly important socioeconomically important consequences. EEG = electroencephalogram. Further CNV profiles are shown in **Supplementary Fig. 3A-E**.

Updated **Table 3** shows the 19 significant CNV-HPO associations in the Phenome cohort.

Table 3: Significant individual CNV-HPO associations.

Locus	CNV	HPO	Odds ratio [95% CI]	Relative risk	P -value		CNV carriers			CNV non-carriers	
					Raw	Adjusted	Prop	N _{pheno}	N _{tot}	N _{pheno}	N _{tot}
15q13.2-q13.3 [31.06-32.51Mb]	DEL	Generalized non-motor (absence) seizure [HP:0002121]	10.5 [4.25-28.5]	4.18	3.70E-08	1.00E-05	0.667	16	24	1731	10856
15q13.2-q13.3 [31.06-32.51Mb]	DEL	Typical absence seizure [HP:0011147]	8.43 [3.48-21.3]	4.1	6.94E-07	1.10E-04	0.583	14	24	1545	10856
15q13.2-q13.3 [31.06-32.51Mb]	DEL	EEG with spike-wave complexes [HP:0010850]	7.84 [3.16-21.2]	3.28	1.18E-06	2.00E-04	0.667	16	24	2205	10856
15q13.2-q13.3 [31.06-32.51Mb]	DEL	Generalized-onset seizure [HP:0002197]	9.41 [3.15-37.9]	2.4	1.41E-06	2.20E-04	0.833	20	24	3766	10856
15q13.2-q13.3 [31.06-32.51Mb]	DEL	EEG with generalized epileptiform discharges [HP:0011198]	6.76 [2.44-23.2]	2.2	1.98E-05	0.00379	0.792	19	24	3905	10856
15q13.2-q13.3 [31.06-32.51Mb]	DEL	Bilateral tonic-clonic seizure with focal onset [HP:0007334]	0 [0-0.404]	0	4.07E-04	0.0484	0	0	24	3168	10856
1p36.33 [0.91-1.51Mb]	DEL	Hypotonia [HP:0001252]	12.2 [3.95-32]	9.51	3.23E-05	0.00674	0.24	6	25	274	10855
1p36.33 [0.91-1.51Mb]	DEL	Epileptic spasm [HP:0011097]	7.47 [2.78-18.4]	5.4	6.85E-05	0.0108	0.32	8	25	643	10855
1p36.33 [0.91-1.51Mb]	DEL	Abnormal muscle tone [HP:0003808]	8.65 [2.81-22.7]	6.82	1.97E-04	0.0287	0.24	6	25	382	10855
1p36.33 [0.91-1.51Mb]	DEL	Infantile spasms [HP:0012469]	8.34 [2.71-21.9]	6.58	2.39E-04	0.0324	0.24	6	25	396	10855
1p36.33 [0.91-1.51Mb]	DEL	Abnormal muscle physiology [HP:0011804]	8.21 [2.67-21.5]	6.48	2.59E-04	0.0339	0.24	6	25	402	10855
1p36.33 [0.91-1.51Mb]	DEL	Abnormality of the musculature [HP:0003011]	8.04 [2.61-21.1]	6.35	2.87E-04	0.038	0.24	6	25	410	10855
1p36.33 [0.91-1.51Mb]	DEL	Plagiocephaly [HP:0001357]	93.8 [9.48-482]	86.8	3.30E-04	0.045	0.08	2	25	10	10855
2p21-p16.3 [47.50-47.85Mb]	DEL	Focal-onset seizure [HP:0007359]	0.463 [0.313-0.681]	0.708	4.79E-05	0.0086	0.456	52	114	6939	10766
2p21-p16.3 [47.50-47.85Mb]	DEL	Bilateral tonic-clonic seizure with generalized onset [HP:0025190]	2.3 [1.5-3.46]	1.88	9.09E-05	0.0157	0.325	37	114	1861	10766
15q12-q13.1 [27.93-28.23Mb]	DEL	Global developmental delay [HP:0001263]	69.1 [5.55-3540]	18.1	2.80E-04	0.0127	0.75	3	4	451	10876
15q12-q13.1 [27.93-28.23Mb]	DEL	Epileptic encephalopathy [HP:0200134]	Inf [4.43-Inf]	7.72	2.83E-04	0.0127	1	4	4	1408	10876
15q12-q13.1 [27.93-28.23Mb]	DEL	Encephalopathy [HP:0001298]	Inf [4.41-Inf]	7.69	2.87E-04	0.0129	1	4	4	1414	10876
16p11.2 [29.87-30.19Mb]	DUP	Psychogenic non-epileptic seizure [HP:0033052]	81.5 [7.85-471]	61.8	4.82E-04	0.0297	0.25	2	8	44	10872

The first column reports the genomic band and coordinates of the considered CNV. The CNV type is reported in column 2. In column 3, the HPO term name and identifier are reported. In column 4, the odds ratio with raw/uncorrected 95% confidence interval is reported. In column 5, the relative risk is given to aid

interpretation. In column 6, the uncorrected P -values are reported. In column 7, the minP step-down P -value is given, which provides an adjustment for all 1,667 HPO term associations tested within each CNV group while accounting for the correlation between harmonized HPO annotations (see Online Methods). In column 8, the proportion of CNV carriers annotated with the phenotype is given. In columns 9-10 and 11-12, N_{pheno} and N_{tot} are the numbers of individuals presenting with the phenotype and the total number of individuals carrying and not-carrying the CNV, respectively.

4. Some of the 35 loci presented in the Table 1 as CNVs with significant association with seizures are not such rare in the general populations (according to the Database for Genomic Variants). For example, 15q11.2 duplications can be found in a frequency of about 0.4%-0.5%. Whereas clearly defined pathogenic CNVs such as 1q21.1 was not listed as being relevant to seizure.

Response 2.4: Our ultra-large-scale genome-wide CNV seizure association study has the strength to not only identify completely penetrant disease-causing CNVs but also copy number polymorphisms (CNVs that confer risk to develop epilepsy, and are not entirely absent in the general population, such as the well-described risk locus 15q11.2). We agree with the reviewer that we needed to point this out and extend the introduction.

Although our CNV screen is the largest to date for epilepsy and seizures, we are still underpowered to identify all CNVs associated with these disorders. Given that CNVs are rare, larger studies are needed to identify more (rare) associations. We expanded the outlook section of the **Discussion**. We only reported CNVs that showed a significant association with seizures in our meta-analysis. Our meta-analysis dataset may not have been powered enough to uncover an association of CNVs at 1q21.1 with seizures. Alternately, CNVs at 1q21.1 may be specifically associated with epilepsy and not with the broader phenotype of seizure disorders. We rephrased the **Results** section “**Discovery of 25 genome-wide significant seizure-associated CNV regions**” to address this observation. Also, in the new **Supplementary Table 7**, we detail the case/control frequencies in our meta-analysis for each significant CNV region and state, for comparison, the frequencies of CNVs in the DGV Gold Standard and DECIPHER Population Copy-Number Variation databases.

Page 5, last paragraph of the **Introduction**, we added: “To explore this hypothesis, we performed a meta-analysis of GWAS studies comprising 26,699 individuals with diagnosed epilepsy or seizures and 492,324 controls. Since both definitions are based on the presence of seizures, we refer to individuals affected by either condition as 'individuals with seizures' from here on forward. *The effective sample size of this study ($N_{\text{eff}}=101,302$) provides adequate power to identify significant associations of risk CNVs that are present in the general healthy population, therefore, do not exhibit complete penetrance. However, the analytic setup restricts the frequency in the general population to up to 1% for quality purposes.* We assessed the pleiotropy of any identified seizure-associated CNV in subsequent meta-analyses of epilepsy and 238,161 independent individuals affected by a range of 23 neuropsychiatric disorders. Finally, using a subset of the seizure cohort comprising 10,880 individuals with epilepsy detailed using 214,2903 Human Phenotype Ontology (HPO) annotations²⁵, we evaluated the clinical features characterizing carriers of each seizure-associated CNV.”

Please see the rephrased Results section “**Discovery of 25 genome-wide significant seizure-associated CNV regions**” in **Response 2.3**.

Page 15, last paragraph of the **Discussion**, we added: “Our results will help refine promising candidate CNVs associated with specific epilepsy types and extend their clinical value. We are confident that applying this framework to even larger datasets has the potential to advance the discovery of *all* clinically relevant risk loci, *ultra-rare high-risk CNVs missed by this study*, and the underlying genes or functional elements.”

Please see the updated **Table 1** in **Response 2.3**.

Please see the updated **Supplementary Table 7** in **Response 2.1**.

5. The authors compare the results to a single previous GWAS study, identifying 32 novel loci, and confirmation of the three that were previously described. They do not compare to any of the previously known regions and/or genes that have been implicated in epilepsy to give a proper summary of what is novel in this study. The study would benefit from comparing the known epilepsy genes and known CNVs and identify which were found in the loci identified in this study and to highlight any that were not found in this relatively large cohort.

Response 2.5: We agree with the reviewer that the manuscript would benefit from a complete overview of whether the identified seizure-associated CNVs have been previously reported in epilepsy or other neurological or psychiatric disorders. To this end, we generated a new **Table 2** to provide information on whether the novel associations overlap with previously reported pathogenic CNVs or genes known to cause brain-related disorders (the previous Table 2 is now **Table 3**). Additional candidate genes of lower confidence are detailed in **Supplementary Table 2**. We also included a new **Results** section, “**Fine-mapping and candidate genes**” and commented on the results in the **Discussion**.

Please see the rephrased Results section “**Discovery of 25 genome-wide significant seizure-associated CNV regions**” in **Response 2.3**.

Page 6, we replaced the second paragraph of the rephrased Results section “**Discovery of 25 genome-wide significant seizure-associated CNV regions**” with the new Results section “**Fine-mapping and candidate genes**” as follows: “*Out of the three CNV regions with previous genome-wide statistical support, our fine-mapping approach narrowed down the critical seizure-relevant region for the known 15q11-q13 duplication to the imprinted promoter/exon 1 region of SNRPN (Table 2, Supplementary Fig. 1). The SNRPN promoter/exon 1 region was suggested to regulate the imprinting of the critical region for Prader-Willi syndrome^{28,29}. Overexpression of SNRPN, corresponding to the seizure-associated duplication of the region, was found to cause abnormal neural development in cultured primary cortical neurons³⁰. Conversely, SNRPN knockdown was found in the same study to also cause subtle neuronal abnormalities, in line with reports of short SNRPN deletions in Prader-Willi syndrome³¹. For the other two CNV regions with previous genome-wide statistical support, we identified several genes with a brain phenotype in the minimal credible intervals. The 15q13.2-q13.3 deletion credible interval includes the haploinsufficient gene OTUD7A, shown to cause abnormal development of cortical dendritic spines and dendrite outgrowth in *Otud7a*^{DEL/+} mice³², and KLF13, shown to cause a layer-specific decrease of cortical interneurons in *Klf13*^{DEL/+} mice³³. The 16p13.11 deletion credible interval includes two haploinsufficient genes: MYH11, implicated in cerebrovascular disorders^{34,35} that are a risk factor for seizures³⁶, and MARF1, involved in cortical neurogenesis³⁷.*

Out of the six seizure-associated CNV regions previously implicated in epilepsy without genome-wide statistical support, we mapped the credible intervals of the two seizure-associated deletions at 1p36 to the first and third known critical regions for seizures within the phenotype spectrum of the 1p36 deletion syndrome³⁸. Known disease genes in the credible intervals at 1p36 are DVL1 (Robinow syndrome³⁹), TMEM240 (Spinocerebellar ataxia 21⁴⁰), and SKI (Shprintzen-Goldberg syndrome⁴¹). In the credible intervals of the remaining CNV regions, we identified the following known disease genes: i) the haploinsufficient KIF26B gene (Pontocerebellar hypoplasia⁴²) as the only gene affected by the 1q44 deletion, and ii) PRRT2 (self-limited familial infantile epilepsy, paroxysmal dyskinesia⁴³) and the haploinsufficient TAOK2 gene (Autism⁴⁴) at the 16p11.2 BP4-BP5 deletion syndrome locus. Of note, single nucleotide variants in PRRT2 are among the most frequent findings in clinical genetic testing of epilepsy⁴⁵.

Among the ten seizure-associated CNV regions previously reported in other neurological and psychiatric disorders, we identified one credible interval suggesting a different causal gene than previously reported: an interstitial 9q34.3 duplication not encompassing EHMT1 that is considered as the causal gene based on one out of 22 reported 9q34.3 duplication carrier⁴⁶. The top candidate gene within the credible interval identified by our meta-analysis is GRIN1, affected by 9q34.3 duplications in 21 of all reported carriers⁴⁶. GRIN1 gain of function variants are known to cause a developmental epileptic encephalopathy, often with polymicrogyria⁴⁷. In contrast, our fine-mapping analysis confirms TBX1 as the (known) causal gene for the 22q11.21 deletion/DiGeorge syndrome⁴⁸. We also found LZTR1 (Noonan syndrome⁴⁹) within the credible 22q11.21 deletion intervals. Other known disease genes in the credible intervals of the remaining CNV regions implicated in neurological and psychiatric disorders were: NPHP1 inside a 2q13 duplication (Autism and global developmental delay^{50,51}), KANK1 (Cerebral palsy spastic quadriplegic 2⁵²) inside a small 9p24.3 DOCK8/KANK1 deletion, and NIPA1 (Autosomal dominant spastic paraplegia 6⁵³) inside the 15q11.2 BP1-BP2 deletion syndrome region.

Finally, we identified four novel CNV regions associated with seizures. Three out of four harbored known disease genes. The credible region of a non-canonical 16p13.3 duplication included STUB1. STUB1 gain of function was reported to cause early onset dementia syndrome⁵⁴ and autosomal dominant ataxia with cognitive decline and autism⁵⁵. The credible region of a non-canonical 17q21.31 deletion included BRCA1. BRCA1 mutations are well-known in cancer⁵⁶, with BRCA1 as a possible mediator of glioma cell proliferation, migration, and glioma stem cell self-renewal⁵⁷. The credible region of a novel 20q13.33 duplication included KCNQ2 and EEF1A2. KCNQ2 gain of function is known to cause neurodevelopmental disability and neonatal encephalopathy^{58,59}. EEF1A2 gain of function was shown to cause neurodevelopmental disorders, including epilepsy and intellectual disability⁶⁰.

Significantly enriched Gene ontology (GO) Biological Processes among all known brain-related disease genes in the credible intervals were: chordate embryonic development (GO:0043009), sensory organ morphogenesis (GO:0090596), mitotic G2 DNA damage checkpoint signaling (GO:0007095), neural tube closure (GO:0001843), negative regulation of Ras protein signal transduction (GO:0046580), dendrite morphogenesis (GO:0048813), and mitotic G2/M transition checkpoint (GO:0044818). No GO Biological Process was significantly enriched when considering all genes inside all credible intervals, pointing to likely heterogeneous disease mechanisms of the 25 seizure-associated CNV regions. All credible intervals and known brain-related disease genes are detailed in **Table 2**, additional candidate genes of lower confidence are detailed in **Supplementary Table 2**, and all genes inside the credible intervals are detailed in **Supplementary Table 3**."

The new **Supplementary Fig. 1** shows our fine-mapping result for the known BP-II to BP-III 15q11-q13 duplication syndrome region, suggesting *SNRPN* as the candidate gene.

Supplementary Fig. 1: Regional CNV distribution plot and fine-mapping of the 15q11-q13 duplication. Known breakpoints are labeled BP-I to BP-V. The dashed line in the first plot represents the Bonferroni-corrected threshold for genome-wide significance, $\alpha=3.74 \times 10^{-6}$. The credible interval containing the causal element/gene with 95% confidence is highlighted in yellow. The dark orange arrow in the lower plot points to the candidate gene of the interval.

Please see the new **Table 2**: Known disease genes in the credible intervals of the seizure-associated CNV regions. Reported clinical phenotypes are highlighted in color.

Cytoband	CNV type	Best overlapping syndrome	Credible interval containing the causal element/gene with 95% confidence	Brain-related disease genes (high confidence)	PMID
15q11.2-q13.3	DUP	15q11-q13 duplication syndrome (Prader-Willi/Angelman critical region)	15:24750000-25080000	SNRPN overexpression (Neurodevelopmental phenotype)	27430727
				SNRPN deletion (Prader-Willi) - if the CNV is gene disrupting	35956251
15q13.2-q13.3	DEL	15q13.3 deletion syndrome	15:31060000-32510000	-	-
16p13.11	DEL	16p13.11 deletion syndrome	16:15420000-16350000	MYH11 (Moyamoya-like cerebrovascular disease, cerebral artery aneurysm)	29263223, 27367753
1p36.33	DEL	1p36 deletion syndrome (Seizures critical region 1)	1:910000-1510000	DVL1 (Robinow syndrome)	25817016
1p36.33	DEL	1p36 deletion syndrome (Seizures critical region 3)	1:2020000-2490000	TMEM240 (Spinocerebellar ataxia 21)	25070513
1q44	DEL	KIF26B deletion	1:245290000-245860000	KIF26B (Pontocerebellar hypoplasia)	30151950
16p12.2	DEL	16p12.1 deletion syndrome	16:21880000-22500000	-	-
16p11.2	DEL	16p11.2 deletion syndrome (BP4-BP5)	16:29560000-30190000	PRRT2 (Benign familial infantile seizures)	33746883
				TAOK2 (Autism spectrum disorder)	29467497
17q12	DUP	17q12 duplication syndrome	17:34760000-35510000	-	-
			17:35960000-36250000	-	-
2q13	DUP	NPHP1 duplication	2:110770000-111060000	NPHP1 duplication (Autism spectrum disorder, global developmental delay)	25126106, 16892302
3q29	DEL	3q29 deletion syndrome	3:195760000-196240000	-	-
8p23.3-p23.2	DEL	8p23.2-pter deletion syndrome	8:400000-610000	-	-
			8:3040000-3780000	-	-
			8:4810000-5470000	-	-
			-	-	-
9p24.3	DEL	9p24.3 DOCK8 / KANK1 deletion	9:330000-560000	KANK1 (Cerebral palsy spastic quadriplegic 2)	16301218
9q34.3	DUP	interstitial 9q34.3 duplication (not encompassing EHMT1)	9:139210000-139590000	-	-
			9:139890000-140120000	GRIN1 gain of function (Polymicrogyria)	29365063
10q26.3	DEL	10q26 deletion syndrome	10:133410000-133740000	-	-
			10:134370000-134680000	-	-
15q11.2	DEL	15q11.2 deletion syndrome (BP1-BP2)	15:22740000-23280000	NIPA1 (Spastic paraplegia 6, autosomal dominant)	23897027
16p11.2	DUP	16p11.2 duplication syndrome (BP4-BP5)	16:29870000-30190000	-	-
22q11.21	DUP	22q11.21 deletion syndrome (LCRA-LCRD)	22:18990000-19370000	-	-
			22:20200000-21540000	-	-
22q11.21	DEL	22q11.21 deletion syndrome (LCRA-LCRD)	22:18990000-19400000	-	-
			22:19670000-19960000	TBX1 (22q11.21 deletion syndrome)	14585638
22:20650000-21540000	LZTR1 (Noonan syndrome)	30368668			
2p21-p16.3	DEL	Lynch syndrome locus	2:47500000-47850000	-	-
19p13.3	DUP	non-canonical 19p13.3 duplication	19:1040000-1340000	-	-
15q12-q13.1	DEL	OCA2 deletion	15:27930000-28230000	-	-
16p13.3	DUP	non-canonical 16p13.3 duplication	16:600000-890000	STUB1 gain of function (early onset dementia syndrome, autosomal dominant ataxia with cognitive decline and autism)	35493319, 32211513
17q21.31	DEL	non-canonical 17q21.31 deletion	17:41080000-41450000	BRCA1 (Cancer)	35393462
20q13.33	DUP	novel 20q13.33 duplication	20:62000000-62350000	KCNQ2 gain of function (Neurodevelopmental disability, neonatal encephalopathy)	35780567, 28139826
				EEF1A2 gain of function (Neurodevelopmental disorders)	32160274

Highlighted are 1.) Purple: three CNV regions with previous genome-wide statistical support for epilepsy (PMID: 32568404), 2.) Light purple: six CNV regions previously implicated in epilepsy without genome-wide statistical support, 3.) Light blue: ten CNV regions previously reported in other neurological and psychiatric disorders, and 4.) Light green: four novel CNV regions never reported in neurological or psychiatric disorders.

Please see the new **Supplementary Table 2** detailing additional candidate genes of lower confidence:

Cytoband	CNV type	95% confidence interval(s) of the causal element/gene	Best overlapping syndrome	Genes segregating with phenotype in families	PMID for segregating genes	Genes with reported brain- or seizure-relevant function	PMID for genes with brain- or seizure-relevant function	Dosage sensitive genes (Haploinsufficient for deletions; Triplosensitive for duplications)	GWAS-mapped genes for neuropsychiatric disorders	PMID for GWAS-mapped genes
1p36.33	DEL	1:910000-1510000	1p36 deletion syndrome (Seizures critica	-	-	-	-	-	-	-
1p36.33	DEL	1:2020000-2490000	1p36 deletion syndrome (Seizures critica	-	-	-	-	SKI, PANK4	PLCH2 (Schizophrenia)	35396580
1q44	DEL	1:245290000-245860000	KIF26B deletion	-	-	-	-	KIF26B	KIF26B (Brain morphok	34910505
2p21-p16.3	DEL	2:47500000-47850000	Lynch syndrome locus	-	-	-	-	MSH2	-	-
2q13	DUP	2:110770000-111060000	NPHP1 duplication	-	-	-	-	-	-	-
3q29	DEL	3:195760000-196240000	3q29 deletion syndrome	-	-	-	-	UBXN7	-	-
8p23.3-p23.2	DEL	8:400000-610000 8:3040000-3780000 8:4810000-5470000	8p23.2-pter deletion syndrome 8p23.2-pter deletion syndrome 8p23.2-pter deletion syndrome	FBXO25 (ADHD) CSMD1 (familial f CSMD1 (familial f	31849056 28808687 28808687	-	-	-	CSMD1 CSMD1	CSMD1 (Schizophrenia 21439553, 29 CSMD1 (Schizophrenia 21439553, 29
9p24.3	DEL	9:330000-560000	9p24.3 DOCK8 / KANK1 deletion	KANK1 (Cerebral	16301218	-	-	-	-	-
9q34.3	DUP	9:139210000-139590000 9:139890000-140120000	interstitial 9q34.3 duplication (not encom interstitial 9q34.3 duplication (not encom	-	-	NOTCH1 gain of functio	23999872	NOTCH1 GRIN1, ABCA2	-	-
10q26.3	DEL	10:133410000-133740000 10:134370000-134680000	10q26 deletion syndrome 10q26 deletion syndrome	-	-	-	-	INPP5A	-	-
15q11.2	DEL	15:22740000-23280000	15q11.2 deletion syndrome (BP1-BP2)	-	-	CYFIP1 (decreased my	31371726	CYFIP1	-	-
15q11.2-q13.3	DUP	15:24750000-25080000	15q11-q13 duplication syndrome (Prader	-	-	-	-	-	-	-
15q12-q13.1	DEL	15:27930000-28230000	OCA2 deletion	-	-	-	-	-	-	-
15q13.2-q13.3	DEL	15:31060000-32510000	15q13.3 deletion syndrome	-	-	OTUD7A (abnormal de KLF13 (layer-specific d	29395074 34838304	OTUD7A	-	-
16p13.3	DUP	16:600000-890000	non-canonical 16p13.3 duplication	-	-	-	-	-	-	-
16p13.11	DEL	16:15420000-16350000	16p13.11 deletion syndrome	-	-	MARF1 (cortical neuro	28442784	MYH11, MARF1	ABCC1 (Deafness, auti	31273342
16p12.2	DEL	16:21880000-22500000	16p12.1 deletion syndrome	-	-	-	-	-	-	-
16p11.2	DEL	16:29560000-30190000	16p11.2 deletion syndrome (BP4-BP5)	-	-	-	-	TAOK2	TAOK2 (Schizophrenia, 35396580, 35 MAPK3 (cortical thickne DOC2A (Alzheimer's dis TMEM219 (Schizophre	34910505 35379992 29483656
16p11.2	DUP	16:29870000-30190000	16p11.2 duplication syndrome (BP4-BP5)	-	-	PRRT2 overexpression	34133925, 34685	-	TAOK2 (Schizophrenia, 35396580, 35 MAPK3 (cortical thickne DOC2A (Alzheimer's dis TMEM219 (Schizophre	34910505 35379992 29483656
17q12	DUP	17:34760000-35510000 17:35960000-36250000	17q12 duplication syndrome 17q12 duplication syndrome	-	-	-	-	ACACA	-	-
17q21.31	DEL	17:41080000-41450000	non-canonical 17q21.31 deletion	-	-	RND2 (neuronal migrat	18690213	-	-	-
19p13.3	DUP	19:1040000-1340000	non-canonical 19p13.3 duplication	-	-	-	-	-	ABCA7 (Alzheimer's dis ARHGAP45 (Alzheimer'	33589840 35379992
20q13.33	DUP	20:62000000-62350000	novel 20q13.33 duplication	-	-	-	-	-	-	-
22q11.21	DUP	22:18990000-19370000 22:20200000-21540000	22q11.21 duplication syndrome (proxima 22q11.21 duplication syndrome (LCRB-L	-	-	SLC25A1 (autistic-like	35203088	-	PI4KA (Cortical thickne	34910505
22q11.21	DEL	22:18990000-19400000 22:19670000-19960000 22:20650000-21540000	22q11.21 deletion syndrome (proximal) 22q11.21 deletion syndrome (proximal) 22q11.21 duplication syndrome (LCRB-L	-	-	HIRA (brain developme	33417013, 28515	HIRA	-	-
				-	-	-	-	SCARF2, MED15	PI4KA (Cortical thickne	34910505

Please see the new additions to the **Discussion in Response 2.3**.

Given the combination of multiple platforms and approaches in the NDD cohorts, they should have been able to better refine the breakpoints compared to earlier studies. In addition, for the loci that have already been described in previous studies, (15q11.2, 15q13.2, 16p11.2 etc.), it would be helpful to show how much the region has been refined in this study using CNV data across such a large set of samples (including the NDD cohort).

Response 2.6: We followed the referee's suggestion and carried out a fine-mapping approach of the identified CNVs to better refine breakpoints using a Bayesian algorithm (PMID: 18642345) to identify the minimal credible interval(s) that contained the causal element(s) with 95% confidence, as in Collins et al. (2022) PMID: 35917817). We then compared the results with previously reported ranges and candidate genes of the corresponding regions in association with neurological or psychiatric phenotypes. The refined breakpoints are now reported in **Table 1** in the "Credible interval containing the causal element/gene with 95% confidence" column. We detailed our findings in the new **Results** section "**Fine-mapping and candidate genes**", supported by a new **Table 2** detailing known disease genes in all credible intervals (former Table 2 is now Table 3), and the new **Supplementary Table 2**.

Please see the new Results section "**Fine-mapping and candidate genes**" in **Response 2.5**.

Please see the updated **Table 1** in **Response 2.3**.

Please see the new **Table 2** detailing reported disease phenotypes and known disease genes for each seizure-associated region and **Supplementary Table 2** with additional candidate genes in **Response 2.5**.

6. CNV calling in the Epi25 cohort was done only using a single algorithm pennCNV, and although some pre/post QC was done there will still be a higher false-positive rate without independent validation using a secondary program. They control for cohorts when combining the data across the NDD cohorts, but it doesn't seem like they control for platform which may have been inherently controlled for in the bin/windowing approach to identify the significant loci, but I'm not certain.

Response 2.7: We controlled for platform-specific artifacts in the first place by filtering the analyzed CNVs to be present at a maximum frequency of <1% in both meta-analysis cohorts (Epi25 and neuropsychiatric disorders cohort, see revised Quality Control sections in **Response 2.1**). Relying on PennCNV only plus extensive quality control is, at the moment, best practice (Sanders et al. Nat Commun. 2022; PMID: 35031607). Also, all individuals of the Epi25 cohort were genotyped with the same array (Global Screening Array) on the same platform (Broad Institute of Harvard and MIT), further reducing the likelihood of platform artifacts. Control vs. control association testing produced no relevant signals in our previous study (Niestroj et al., 2020, PMID: 32568404). We also visually inspected the probe-level intensity plots of all CNVs supporting the seizure-associated regions to exclude any remaining artifacts. We rephrased the corresponding parts of the **Methods** accordingly.

Please see the rephrased Methods sections “**Quality control - neuropsychiatric disorders cohort**” and “**CNV calling and quality control - Epi25 Collaborative**” in **Response 2.1**.

7. In Table 2, the authors should identify if the locus listed is a known "genomic disorder" (i.e. flanked by segmental duplication with high recurrent mutation rate). This should figure into the Discussion.

Response 2.8: We agree this information is essential, which we have detailed in the updated Result sections “**Discovery of 25 genome-wide significant seizure-associated CNV regions**” and “**Fine-mapping and candidate genes**”. Any documented breakpoints corresponding to the identified seizure-associated CNV region are detailed in **Table 2**. Such elements flanked 5/25 CNV regions.

Please see the new **Table 2** detailing previously reported CNV syndromes overlapping with the seizure-associated region in **Response 2.5**.

Please see an addition to the first paragraph of the **Discussion** in **Response 2.3**.

Reviewer #3 (Remarks to the Author):

The authors perform a meta-analysis of copy number variation with epilepsy or seizure phenotypes across the Epi25 data (16k cases, 8.5k controls) and a subset of data from Collins et al., 2022 (10.5k cases, 484k controls).

The authors mostly use methods described in Collins et al., Cell 2022 for calling CNVs (key being that the CNVs are at least 200kb in size) and phenotyping the individuals (highest resolution at human phenotype ontology [HPO] terms). They ultimately identify 35 loci, expanding the number of known loci linked to epilepsy/seizures from 3 previously known loci in epilepsy.

These loci range from several hundred kb to over a 1Mb, and include deletions and duplications. Many of them are associated with well-characterized syndromic diseases where epilepsy is a comorbidity. Many of these have been known since karyotyping and cytogenetics have been used for understanding developmental delay. As one might expect from this cursory observation and the large size of the CNVs, the loci are highly pleiotropic with strong associations with 80% exhibiting strong associations with developmental delay.

At some level, the current manuscript largely repackages associations identified in Collins et al., prior epilepsy consortia, and/or large studies of CNVs in developmental delay (e.g. Coe et al. Nat Genet 2019; Cooper et al., Nat Genet 2011, those well cataloged by DECIPHER) with a real synthesis of the results.

Response 3.1: Our study has significant additions to the Collins et al., Cell 2022 study: 1.) We provide 16,109 new individuals with epilepsy (4,056 new and 12,053 with updated phenotypes) globally collected through the largest clinical project in epilepsy research ‘Epi25’. In addition, all patients had clinical diagnoses and were classified to the latest version of the ILAE guidelines by the submitting neurologist and a phenotype committee of the epi25 (<https://epi-25.org>); 2.) We provide never-published ultra-deep phenotype data for epilepsy patients curated by experts and enhanced with >15,000 HPO terms. It is important to stress that the PheWAS analysis, based on this data, performs a genotype-first approach using the HPO terms to discover new high-resolution CNV-HPO associations to narrow down the clinical spectrum. We observed many associations of clinical relevance for epileptologists that will likely spin off treatment-related research studies.

Regardless of this work being a meta-analysis that adds very little unique data compared to prior publications, I see this manuscript as being impactful in two ways. First, it appears to identify some novel loci in epilepsy/seizures. Second, it could serve as a resource for the epilepsy research and clinical communities. However, both are limited by the current presentation of the work.

Response 3.2: We thank the reviewer for highlighting the study's strengths and the need to improve our work's presentation. In response to comments from all reviewers, we expanded our methods/analyses, improved the readability of the manuscript, and point now out that:

1. This study found 25 CNV loci as genome-wide associated with seizures. This phenotype was only partially investigated as part of epilepsy but never across all seizure disorders (usually not considered for epilepsy studies). Thus, all 25 CNVs are “novel” risk loci or represent phenotype expansions. 3/25 had previous genome-wide statistical support in epilepsy, 6/25 were reported in epilepsy by studies without genome-wide statistical support, 10/25 in

- neurological and psychiatric disorders, 2/25 reported in non-neurological disorders, and 4/25 were never reported (new **Table 2**).
2. The fine-mapping analysis identified 33 intervals within those 25 CNV loci that likely contain the causal element/gene with 95% confidence (updated Table 1). In these 33 (credible) intervals, we report 18 genes known to cause a seizure-relevant phenotype corresponding to the effect of the CNV (copy loss vs. copy gain) (new **Table 2**) and 11 genes reported to have relevant effects on brain morphology or function, also concordant to the effect of the CNV (new **Supplementary Table 2**).
 3. In meta-analyses of the seizure CNV-GWAS with 23 other neurological and psychiatric disorders, we found strong evidence for pleiotropy for 24/25 CNV loci. This observation was in line with our pathway analyses across all genes in the 33 credible intervals, not filtered for function/known caused phenotype, which was negative, suggesting a lack of power or highly heterogeneous disease mechanisms of the 25 seizure-associated CNV regions.
 4. Finally and importantly, we performed a novel type of CNV-HPO PheWAS analysis developed specifically for this study. The basis of this analysis was carriers of CNVs overlapping with any of the 25 identified CNV regions with deep phenotypes (a subset of the Epi25 cohort, **Supplementary Table 1**). The CNV-HPO PheWAS analysis successfully identified 19 significant epilepsy type-specific associations of CNVs with deep phenotypes (updated **Table 3**) and 605 nominal associations (updated **Supplementary Table 6**). The 19 significant associations are clinically useful and can inform treatment decisions, clinical genetic testing, and utilization of ACMG CNV guidelines that require quantifiable CNV pathogenicity assessment criteria (PMID: 31690835). Also, the biological annotations of the HPO-associated loci will promote future studies.

We then made accordingly several changes to the manuscript. We have added additional information to improve the value of our work as a resource for epilepsy research and clinical communities: 1.) we improved and expanded the fine-mapping analysis to narrow down the associated regions to the minimal credible interval(s) that likely contain the causal element(s) with 95% confidence, as in Collins et al. (2022), 2.) we compared the overlap of all identified seizure-associated CNV regions with previously reported pathogenic elements (CNVs or genes) leading to any disorder, and 3.) identified candidate genes based on previous findings, biological function, and pathway analyses. We have reworded the **Results** section “**Discovery of 25 genome-wide significant seizure-associated CNV regions**” and added a new **Results** section, “**Fine-mapping and candidate genes**”, to detail our fine-mapping and give, where possible, the biological context in which the identified CNVs / candidate genes may lead to seizures. We updated the CNV-HPO PheWAS analysis in the Phenome cohort to represent the newly identified credible intervals (updated **Results** section “**Characterization of the clinical subphenotypes enriched in the carriers of each seizure-associated CNV in epilepsy patients with deep phenotypes**”). Also, we have included a new **Supplementary Table 7** that states the frequencies of overlapping CNVs from two population databases (DGV Gold Standard and DECIPHER Population frequencies) along with the case/control CNV frequencies in our meta-analysis dataset. The new **Table 2** details previously reported disease phenotypes for each seizure-associated CNV region (the former Table 2 is now Table 3). All resources used to collect information about each CNV and the affected genes are detailed in a new **Supplementary Table 8**.

Page 18, we changed the Methods section “**Meta-analysis**” to “**Meta-analysis and fine-mapping**” with the following addition at the end of the section: “*We then used a Bayesian algorithm¹⁰² to identify the minimal credible interval(s) that contained the causal element(s) or genes with 95% confidence, as in Collins et al. (2022)⁹⁷. Finally, we explored the known biological function of all genes*”

*within the credible intervals and performed pathway analyses using Enrichr^{103,104}. All resources used to investigate the knowledge basis of all seizure-associated CNV regions are described in **Supplementary Table 8**.”*

Page 5, we changed the Results section “**Discovery of 35 genome-wide significant seizure-associated CNVs**” to “**Discovery of 25 genome-wide significant seizure-associated CNV regions**” and modified it as follows: “We performed a meta-analysis of 16,109 individuals with epilepsy and 8,545 population controls (the Epi25 Collaborative cohort) with 10,590 individuals with seizures (not explicitly meeting diagnostic criteria for epilepsy) and 483,779 population controls, derived from an aggregated CNV dataset of 17 cohorts (neuropsychiatric disorders cohort) (see all cohorts of this study in **Supplementary Table 1**). The genome was scanned using 267,237 genomic segments of 200kb size in a 10kb sliding window approach²⁶. After applying Bonferroni correction of the threshold for a significant association in the meta-analysis *and fine-mapping*, we identified **25** loci associated with seizures at genome-wide significance ($P \leq 3.74 \times 10^{-6}$). All **25** loci are shown in **Fig. 1** and detailed in **Table 1**. The **25** identified loci included **15** deletion CNVs (size range: 230kb to 5Mb) and **ten** duplication CNVs (size range: 290kb to 8.9Mb). *All the genome-wide associated deletions found in this study consisted of the loss of one copy, while all duplications consisted of the gain of one copy. Three of the 25 seizure-associated loci (15q11.2-q13.3 dup, 15q13.2-q13.3 del, 16p13.11 del) had previous genome-wide statistical support for an association with epilepsy from our previous study²⁰ that included 40% of the individuals with seizures of this study.* All other identified CNVs (22/25, 88%) represent new genome-wide significant loci for seizures, *with 10/22 (59%) loci previously implicated in neurological and psychiatric disorders, 6/22 (23%) specifically in epilepsy by studies without genome-wide statistical support, 2/22 (9%) reported in individuals without neurological or psychiatric disorders, and 4/22 (18%) not previously reported regions.* We detailed in **Table 2** all commonly reported disease phenotypes for the 25 identified seizure-associated loci. *Our meta-analysis in seizure disorders was likely not powered enough to identify some of the known CNVs implicated in epilepsy (without genome-wide statistical support) associated with seizures (e.g., 1q21.1 del/dup).* Reciprocal CNVs, defined by deletions and duplications associated with seizures involving overlapping genomic segments, were found at 15q11.2, 16p11.2, and 22q11.21. *No overlap existed between the seizure-associated CNV regions identified in this study and the most recent SNP-based GWAS study in epilepsy²⁷.”*

Updated **Table 1**: Genome-wide significantly associated CNV regions and credible intervals.

Cytoband	CNV type	Hg19 Start (Mb)	Hg19 End (Mb)	Lowest P-value in region	OR [95% CI]	Credible interval containing the causal element/gene with 95% confidence	Associations with neuropsychiatric phenotypes [N]	Highest odds ratio in neuropsychiatric disorder/seizure meta-analyses (95% CI)
1p36.33	DEL	0.91	1.51	4.65E-09	23 (10-54)	910000-1510000	5	11 (5-24) CNS abnormality
1p36.33	DEL	2.02	2.49	1.95E-07	44 (14-141)	2020000-2490000	3	48 (14-170) Abnormalities of Cognition
1q44	DEL	245.29	245.86	6.59E-07	41 (13-133)	245290000-245860000	6	62 (19-207) Abnormal brain morphology
2p21-p16.3	DEL	47.5	47.85	3.00E-13	12 (7-22)	47500000-47850000	23	13 (8-24) Behavioral abnormalities
2q13	DUP	110.77	111.06	1.33E-06	3 (2-5)	110770000-111060000	6	24 (9-63) Hyperactivity
3q29	DEL	195.76	196.24	2.01E-07	40 (13-122)	195760000-196240000	0	no significant association signal
8p23.3-p23.2	DEL	0.4	5.47	1.83E-08	12 (6-25)	400000-610000 3040000-3780000 4810000-5470000	4	16 (7-39) Intellectual disability
9p24.3	DEL	0.33	0.56	1.08E-07	13 (6-29)	330000-560000	1	7 (4-12) Neurodevelopmental abnormality
9q34.3	DUP	139.21	140.12	1.67E-06	12 (5-27)	139210000-139590000 139890000-140120000	21	12 (6-25) Schizophrenia
10q26.3	DEL	133.41	134.68	3.16E-07	40 (13-125)	133410000-133740000 134370000-134680000	2	32 (10-107) Sleep disorder
15q11.2	DEL	22.74	23.28	1.02E-13	3 (2-3)	22740000-23280000	12	3 (2-4) Abnormalities of Cognition
15q11.2-q13.3	DUP	22.98	32.15	3.68E-19	27 (14-52)	24750000-25080000	23	33 (16-67) Intellectual disability
15q12-q13.1	DEL	27.93	28.23	9.85E-07	14 (6-34)	27930000-28230000	3	24 (11-52) Intellectual disability
15q13.2-q13.3	DEL	31.06	32.51	6.71E-16	14 (8-24)	31060000-32510000	17	16 (6-44) Sleep disorder
16p13.3	DUP	0.6	0.89	1.98E-13	9 (5-14)	600000-890000	23	12 (6-26) Schizophrenia
16p13.11	DEL	15.42	16.35	3.53E-17	9 (6-14)	15420000-16350000	18	13 (4-44) CNS atrophy
16p12.2	DEL	21.88	22.5	1.14E-06	4 (2-5)	21880000-22500000	6	4 (2-6) Hyperactivity
16p11.2	DEL	29.56	30.19	1.25E-11	9 (5-15)	29560000-30190000	11	13 (8-21) Intellectual disability
16p11.2	DUP	29.87	30.19	2.45E-08	6 (3-10)	29870000-30190000	12	11 (5-24) Sleep disorder
17q12	DUP	34.76	36.25	1.12E-15	15 (9-27)	34760000-35510000 35960000-36250000	14	10 (5-20) Abnormal brain morphology
17q21.31	DEL	41.08	41.45	1.98E-08	5 (3-9)	41080000-41450000	20	6 (4-10) Abnormality of the nervous system
19p13.3	DUP	1.04	1.34	2.85E-10	7 (4-12)	1040000-1340000	23	7 (4-14) Schizophrenia
20q13.33	DUP	62	62.35	7.12E-11	6 (4-10)	62000000-62350000	23	8 (5-13) CNS abnormality
22q11.21	DUP	18.99	21.54	8.73E-11	5 (3-7)	18990000-19370000 20200000-21540000 18990000-19400000	9	5 (3-7) Hyperactivity
22q11.21	DEL	18.99	21.54	1.57E-08	26 (10-65)	19670000-19960000 20650000-21540000	12	26 (9-70) Central motor dysfunction

Column 1: Cytoband localization of the CNV. Column 2: CNV type, either deletion (DEL) or duplication (DUP). Columns 3 and 4: Genomic coordinates (in Mb) on the GRCh37 reference genome of the start and end position of the merged CNV region that is supported by genome-wide association signals. Columns 5 and 6: Lowest *P*-values in each CNV region and corresponding odds ratios (with 95% confidence interval) of the genome-wide CNV meta-analysis in 25,345 individuals with seizures and 492,324 controls. *Column 7: GRCh37 coordinates of the credible interval(s) that contained the causal element(s) with 95% confidence.* Column 8: N=Number of neuropsychiatric disorders that showed a significant genome-wide CNV association in this locus. Column 9: Highest odds ratio for each locus in any of the 23 cross-disorder meta-analyses. Deletions are shown in rows with a light blue background, and duplications are shown in rows with white background.

Page 6, we replaced the second paragraph of the rephrased Results section “**Discovery of 25 genome-wide significant seizure-associated CNV regions**” with the new Results section “**Fine-mapping and candidate genes**” as follows: “*Out of the three CNV regions with previous genome-wide statistical support, our fine-mapping approach narrowed down the critical seizure-relevant region for the known 15q11-q13 duplication to the imprinted promoter/exon 1 region of SNPRN (Table 2, Supplementary Fig. 1). The SNRPN promoter/exon 1 region was suggested to regulate the imprinting of the critical region for Prader-Willi syndrome^{28,29}. Overexpression of SNRPN, corresponding to the seizure-associated duplication of the region, was found to cause abnormal neural development in cultured primary cortical neurons³⁰. Conversely, SNRPN knockdown was found in the same study to also cause subtle neuronal abnormalities, in line with reports of short SNRPN deletions in Prader-Willi syndrome³¹. For the other two CNV regions with previous genome-wide statistical support, we identified several genes with a brain phenotype in the minimal credible intervals. The 15q13.2-q13.3 deletion credible interval includes the haploinsufficient gene OTUD7A, shown to cause abnormal development of cortical dendritic spines and dendrite outgrowth in *Otud7a*^{DEL/+} mice³², and KLF13, shown to cause a layer-specific decrease of cortical interneurons in *Klf13*^{DEL/+} mice³³. The 16p13.11*

deletion credible interval includes two haploinsufficient genes: MYH11, implicated in cerebrovascular disorders^{34,35} that are a risk factor for seizures³⁶, and MARF1, involved in cortical neurogenesis³⁷.

Out of the six seizure-associated CNV regions previously implicated in epilepsy without genome-wide statistical support, we mapped the credible intervals of the two seizure-associated deletions at 1p36 to the first and third known critical regions for seizures within the phenotype spectrum of the 1p36 deletion syndrome³⁸. Known disease genes in the credible intervals at 1p36 are DVL1 (Robinow syndrome³⁹), TMEM240 (Spinocerebellar ataxia 21⁴⁰), and SKI (Shprintzen-Goldberg syndrome⁴¹). In the credible intervals of the remaining CNV regions, we identified the following known disease genes: i) the haploinsufficient KIF26B gene (Pontocerebellar hypoplasia⁴²) as the only gene affected by the 1q44 deletion, and ii) PRRT2 (self-limited familial infantile epilepsy, paroxysmal dyskinesia⁴³) and the haploinsufficient TAOK2 gene (Autism⁴⁴) at the 16p11.2 BP4-BP5 deletion syndrome locus. Of note, single nucleotide variants in PRRT2 are among the most frequent findings in clinical genetic testing of epilepsy⁴⁵.

Among the ten seizure-associated CNV regions previously reported in other neurological and psychiatric disorders, we identified one credible interval suggesting a different causal gene than previously reported: an interstitial 9q34.3 duplication not encompassing EHMT1 that is considered as the causal gene based on one out of 22 reported 9q34.3 duplication carrier⁴⁶. The top candidate gene within the credible interval identified by our meta-analysis is GRIN1, affected by 9q34.3 duplications in 21 of all reported carriers⁴⁶. GRIN1 gain of function variants are known to cause a developmental epileptic encephalopathy, often with polymicrogyria⁴⁷. In contrast, our fine-mapping analysis confirms TBX1 as the (known) causal gene for the 22q11.21 deletion/DiGeorge syndrome⁴⁸. We also found LZTR1 (Noonan syndrome⁴⁹) within the credible 22q11.21 deletion intervals. Other known disease genes in the credible intervals of the remaining CNV regions implicated in neurological and psychiatric disorders were: NPHP1 inside a 2q13 duplication (Autism and global developmental delay^{50,51}), KANK1 (Cerebral palsy spastic quadriplegic 2⁵²) inside a small 9p24.3 DOCK8/KANK1 deletion, and NIPA1 (Autosomal dominant spastic paraplegia 6⁵³) inside the 15q11.2 BP1-BP2 deletion syndrome region.

Finally, we identified four novel CNV regions associated with seizures. Three out of four harbored known disease genes. The credible region of a non-canonical 16p13.3 duplication included STUB1. STUB1 gain of function was reported to cause early onset dementia syndrome⁵⁴ and autosomal dominant ataxia with cognitive decline and autism⁵⁵. The credible region of a non-canonical 17q21.31 deletion included BRCA1. BRCA1 mutations are well-known in cancer⁵⁶, with BRCA1 as a possible mediator of glioma cell proliferation, migration, and glioma stem cell self-renewal⁵⁷. The credible region of a novel 20q13.33 duplication included KCNQ2 and EEF1A2. KCNQ2 gain of function is known to cause neurodevelopmental disability and neonatal encephalopathy^{58,59}. EEF1A2 gain of function was shown to cause neurodevelopmental disorders, including epilepsy and intellectual disability⁶⁰.

Significantly enriched Gene ontology (GO) Biological Processes among all known brain-related disease genes in the credible intervals were: chordate embryonic development (GO:0043009), sensory organ morphogenesis (GO:0090596), mitotic G2 DNA damage checkpoint signaling (GO:0007095), neural tube closure (GO:0001843), negative regulation of Ras protein signal transduction (GO:0046580), dendrite morphogenesis (GO:0048813), and mitotic G2/M transition checkpoint (GO:0044818). No GO Biological Process was significantly enriched when considering all genes inside all credible intervals, pointing to likely heterogeneous disease mechanisms of the 25 seizure-associated CNV regions. All credible intervals and known brain-related disease genes are detailed in **Table 2**, additional candidate genes of lower confidence are detailed in **Supplementary Table 2**, and all genes inside the credible intervals are detailed in **Supplementary Table 3**."

The new **Supplementary Fig. 1** shows our fine-mapping result for the known BP-II to BP-III 15q11-q13 duplication syndrome region, suggesting SNRPN as the candidate gene.

Supplementary Fig. 1: Regional CNV distribution plot and fine-mapping of the 15q11-q13 duplication. Known breakpoints are labeled BP-I to BP-V. The dashed line in the first plot represents the Bonferroni-corrected threshold for genome-wide significance, $\alpha=3.74 \times 10^{-6}$. The credible interval containing the causal element/gene with 95% confidence is highlighted in yellow. The dark orange arrow in the lower plot points to the candidate gene of the interval.

Please see the new **Table 2**: Known disease genes in the credible intervals of the seizure-associated CNV regions. Reported clinical phenotypes are highlighted in color.

Cytoband	CNV type	Best overlapping syndrome	Credible interval containing the causal element/gene with 95% confidence	Brain-related disease genes (high confidence)	PMID
15q11.2-q13.3	DUP	15q11-q13 duplication syndrome (Prader-Willi/Angelman critical region)	15:24750000-25080000	SNRPN overexpression (Neurodevelopmental phenotype)	27430727
				SNRPN deletion (Prader-Willi) - if the CNV is gene disrupting	35956251
15q13.2-q13.3	DEL	15q13.3 deletion syndrome	15:31060000-32510000	-	-
16p13.11	DEL	16p13.11 deletion syndrome	16:15420000-16350000	MYH11 (Moyamoya-like cerebrovascular disease, cerebral artery aneurysm)	29263223, 27367753
1p36.33	DEL	1p36 deletion syndrome (Seizures critical region 1)	1:910000-1510000	DVL1 (Robinow syndrome)	25817016
1p36.33	DEL	1p36 deletion syndrome (Seizures critical region 3)	1:2020000-2490000	TMEM240 (Spinocerebellar ataxia 21)	25070513
				SKI (Shprintzen-Goldberg syndrome)	23023332
1q44	DEL	KIF26B deletion	1:245290000-245860000	KIF26B (Pontocerebellar hypoplasia)	30151950
16p12.2	DEL	16p12.1 deletion syndrome	16:21880000-22500000	-	-
16p11.2	DEL	16p11.2 deletion syndrome (BP4-BP5)	16:29560000-30190000	PRRT2 (Benign familial infantile seizures)	33746883
				TAOK2 (Autism spectrum disorder)	29467497
17q12	DUP	17q12 duplication syndrome	17:34760000-35510000	-	-
			17:35960000-36250000	-	-
2q13	DUP	NPHP1 duplication	2:110770000-111060000	NPHP1 duplication (Autism spectrum disorder, global developmental delay)	25126106, 16892302
3q29	DEL	3q29 deletion syndrome	3:195760000-196240000	-	-
8p23.3-p23.2	DEL	8p23.2-pter deletion syndrome	8:400000-610000	-	-
			8:3040000-3780000	-	-
			8:4810000-5470000	-	-
			-	-	-
9p24.3	DEL	9p24.3 DOCK8 / KANK1 deletion	9:330000-560000	KANK1 (Cerebral palsy spastic quadriplegic 2)	16301218
9q34.3	DUP	interstitial 9q34.3 duplication (not encompassing EHMT1)	9:139210000-139590000	-	-
			9:139890000-140120000	GRIN1 gain of function (Polymicrogyria)	29365063
10q26.3	DEL	10q26 deletion syndrome	10:133410000-133740000	-	-
			10:134370000-134680000	-	-
15q11.2	DEL	15q11.2 deletion syndrome (BP1-BP2)	15:22740000-23280000	NIPA1 (Spastic paraplegia 6, autosomal dominant)	23897027
16p11.2	DUP	16p11.2 duplication syndrome (BP4-BP5)	16:29870000-30190000	-	-
22q11.21	DUP	22q11.21 deletion syndrome (LCRA-LCRD)	22:18990000-19370000	-	-
			22:20200000-21540000	-	-
22q11.21	DEL	22q11.21 deletion syndrome (LCRA-LCRD)	22:18990000-19400000	-	-
			22:19670000-19960000	TBX1 (22q11.21 deletion syndrome)	14585638
			22:20650000-21540000	LZTR1 (Noonan syndrome)	30368668
2p21-p16.3	DEL	Lynch syndrome locus	2:47500000-47850000	-	-
19p13.3	DUP	non-canonical 19p13.3 duplication	19:1040000-1340000	-	-
15q12-q13.1	DEL	OCA2 deletion	15:27930000-28230000	-	-
16p13.3	DUP	non-canonical 16p13.3 duplication	16:600000-890000	STUB1 gain of function (early onset dementia syndrome, autosomal dominant ataxia with cognitive decline and autism)	35493319, 32211513
17q21.31	DEL	non-canonical 17q21.31 deletion	17:41080000-41450000	BRCA1 (Cancer)	35393462
20q13.33	DUP	novel 20q13.33 duplication	20:62000000-62350000	KCNQ2 gain of function (Neurodevelopmental disability, neonatal encephalopathy)	35780567, 28139826
				EEF1A2 gain of function (Neurodevelopmental disorders)	32160274

Highlighted are 1.) Purple: three CNV regions with previous genome-wide statistical support for epilepsy (PMID: 32568404), 2.) Light purple: six CNV regions previously implicated in epilepsy without genome-wide statistical support, 3.) Light blue: ten CNV regions previously reported in other neurological and psychiatric disorders, and 4.) Light green: four novel CNV regions never reported in neurological or psychiatric disorders.

Please see the new **Supplementary Table 2** detailing additional candidate genes of lower confidence:

Cytoband	CNV type	95% confidence interval(s) of the causal element/gene	Best overlapping syndrome	Genes segregating with phenotype in families	PMID for segregating genes	Genes with reported brain- or seizure-relevant function	PMID for genes with brain- or seizure-relevant function	Dosage sensitive genes (Haploinsufficient for deletions; Triplosensitive for duplications)	GWAS-mapped genes for neuropsychiatric disorders	PMID for GWAS-mapped genes
1p36.33	DEL	1:910000-1510000	1p36 deletion syndrome (Seizures critica	-	-	-	-	-	-	-
1p36.33	DEL	1:2020000-2490000	1p36 deletion syndrome (Seizures critica	-	-	-	-	SKI, PANK4	PLCH2 (Schizophrenia)	35396580
1q44	DEL	1:245290000-245860000	KIF26B deletion	-	-	-	-	KIF26B	KIF26B (Brain morphok	34910505
2p21-p16.3	DEL	2:47500000-47850000	Lynch syndrome locus	-	-	-	-	MSH2	-	-
2q13	DUP	2:110770000-111060000	NPHP1 duplication	-	-	-	-	-	-	-
3q29	DEL	3:195760000-196240000	3q29 deletion syndrome	-	-	-	-	UBXN7	-	-
8p23.3-p23.2	DEL	8:400000-610000 8:3040000-3780000 8:4810000-5470000	8p23.2-pter deletion syndrome 8p23.2-pter deletion syndrome 8p23.2-pter deletion syndrome	FBXO25 (ADHD) CSMD1 (familial f CSMD1 (familial f	31849056 28808687 28808687	- - -	- - -	- CSMD1 CSMD1	CSMD1 (Schizophrenia 21439553, 29 CSMD1 (Schizophrenia 21439553, 29	- - -
9p24.3	DEL	9:330000-560000	9p24.3 DOCK8 / KANK1 deletion	KANK1 (Cerebral	16301218	-	-	-	-	-
9q34.3	DUP	9:139210000-139590000 9:139890000-140120000	interstitial 9q34.3 duplication (not encom interstitial 9q34.3 duplication (not encom	-	-	NOTCH1 gain of functi	23999872	NOTCH1 GRIN1, ABCA2	-	-
10q26.3	DEL	10:133410000-133740000 10:134370000-134680000	10q26 deletion syndrome 10q26 deletion syndrome	-	-	-	-	INPP5A	-	-
15q11.2	DEL	15:22740000-23280000	15q11.2 deletion syndrome (BP1-BP2)	-	-	CYFIP1 (decreased my	31371726	CYFIP1	-	-
15q11.2-q13.3	DUP	15:24750000-25080000	15q11-q13 duplication syndrome (Prader	-	-	-	-	-	-	-
15q12-q13.1	DEL	15:27930000-28230000	OCA2 deletion	-	-	-	-	-	-	-
15q13.2-q13.3	DEL	15:31060000-32510000	15q13.3 deletion syndrome	-	-	OTUD7A (abnormal de KLF13 (layer-specific d	29395074 34838304	OTUD7A	-	-
16p13.3	DUP	16:600000-890000	non-canonical 16p13.3 duplication	-	-	-	-	-	-	-
16p13.11	DEL	16:15420000-16350000	16p13.11 deletion syndrome	-	-	MARF1 (cortical neuro	28442784	MYH11, MARF1	ABCC1 (Deafness, aut	31273342
16p12.2	DEL	16:21880000-22500000	16p12.1 deletion syndrome	-	-	-	-	-	-	-
16p11.2	DEL	16:29560000-30190000	16p11.2 deletion syndrome (BP4-BP5)	-	-	-	-	TAKO2	TAKO2 (Schizophrenia, 35396580, 35 MAPK3 (cortical thickne 34910505 DOC2A (Alzheimer's di 35379992 TMEM219 (Schizophre 29483656	- - - -
16p11.2	DUP	16:29870000-30190000	16p11.2 duplication syndrome (BP4-BP5)	-	-	PRRT2 overexpression 34133925, 34685	-	-	TAKO2 (Schizophrenia, 35396580, 35 MAPK3 (cortical thickne 34910505 DOC2A (Alzheimer's di 35379992 TMEM219 (Schizophre 29483656	- - - -
17q12	DUP	17:34760000-35510000 17:35960000-36250000	17q12 duplication syndrome 17q12 duplication syndrome	-	-	-	-	ACACA	-	-
17q21.31	DEL	17:41080000-41450000	non-canonical 17q21.31 deletion	-	-	RND2 (neuronal migrat	18690213	-	-	-
19p13.3	DUP	19:1040000-1340000	non-canonical 19p13.3 duplication	-	-	-	-	-	ABCA7 (Alzheimer's di 33589840 ARHGAP45 (Alzheimer 35379992	- -
20q13.33	DUP	20:62000000-62350000	novel 20q13.33 duplication	-	-	-	-	-	-	-
22q11.21	DUP	22:18990000-19370000 22:20200000-21540000	22q11.21 duplication syndrome (proxima 22q11.21 duplication syndrome (LCRB-L	-	-	SLC25A1 (autistic-like 35203088	-	-	PI4KA (Cortical thickne	34910505
22q11.21	DEL	22:18990000-19400000 22:19670000-19960000 22:20650000-21540000	22q11.21 deletion syndrome (proximal) 22q11.21 deletion syndrome (proximal) 22q11.21 duplication syndrome (LCRB-L	-	-	HIRA (brain developme 33417013, 28515	-	HIRA	-	-
				-	-	-	-	SCARF2, MED15	PI4KA (Cortical thickne	34910505

Page 9, we updated the Results section “**Characterization of the clinical subphenotypes enriched in the carriers of each seizure-associated CNV in epilepsy patients with deep phenotypes**” as follows: “*We performed phenome-wide association analyses for each of the 33 credible intervals identified across the 25 CNV regions to characterize the high-resolution clinical manifestations associated with each CNV. This analysis was performed on a subset of the Epi25 Collaborative cohort (Phenomic cohort, **Supplementary Table 1**) comprising 10,880 individuals with non-acquired epilepsy and deep phenotypic data (the clinical presentation of this cohort of 10,880 individuals and the frequencies of selected common and characteristic epilepsy phenotypes are provided in **Supplementary Table 5**). In the Phenomic cohort, 562 individuals (5.2%) carried at least one seizure-associated credible interval (N=498 / 4.6% carried one credible interval, N=64 / 0.6% carried 2-5 credible intervals). The most common credible interval (deletion at 2p21-p16.3) was carried by 114 (1.0%) individuals, and 18 credible intervals were found in at least 0.1% of the cohort (≥11 carriers). One CNV was not found (deletion at 9p24.3, containing a single credible interval). Across the 32 detected credible intervals and 1,667 annotated HPO concepts, we identified 622 nominally significant associations (two-sided Fisher's exact test, **Supplementary Table 6**). Given the large number of associations tested and that HPO annotations describing the same clinical feature at different levels of precision are highly correlated, we applied the minP step-down procedure to aid interpretation⁶¹, yielding 19 associations robust to multiple testing within each genetically defined group (minP-adjusted P<0.05, **Table 3, Fig. 2, Fig. 3, and Supplementary Fig. 2A-E**).*

Carriers of deletions at 1p36.33 [0.91-1.51Mb] (N=25, 0.23% of the Phenomic cohort), 1p36.33 [2.02-2.49Mb] (N=17, 0.16%), or 15q12-q13.1 (N=4, 0.037%), and carriers of duplications at

15q11.2-q13.3 (N=46, 0.42%) were enriched with clinical features suggestive of developmental and epileptic encephalopathies, such as epileptic spasms and tonic seizures, epileptic encephalopathy, and other neurodevelopmental disorders, sudden unexpected death in epilepsy, and morphological abnormalities⁶². Features characterizing genetic generalized epilepsy were associated with deletions at 2p21-p16.3 (N=114, 1.05%, generalized tonic-clonic and absence seizures), 15q11.2 (N=56, 0.52%, eyelid myoclonia and absence seizures), 16p13.11 (N=42, 0.39%, generalized tonic-clonic seizures), 15q13.2-q13.3 (N=24, 0.22%, absence seizures) or 22q11.21 [20.65-21.54Mb] (N=6, 0.055%, juvenile myoclonic epilepsy-like features). Duplications at 16p11.2 (N=8, 0.074%) were associated with non-epileptic seizures comorbid with epilepsy (OR=81.5, unadjusted P=4.82x10⁻⁴, minP-adjusted P=0.0297), and showed a nonsignificant greater frequency of microcephaly (OR=31.5, unadjusted P=3.62x10⁻², minP-adjusted P=0.92) that replicates the mirror microcephaly/macrocephaly phenotype of the reciprocal 16p11.2 CNVs⁶³.

We interrogated the phenotypic annotations of CNV carriers regarding the candidate genes prioritized in our fine-mapping analysis. MSH2 was prioritized as the candidate gene for the most common deletion in the Phenomic cohort (2p21-p16.3). Heterozygous loss of function variants of the haploinsufficient gene MSH2 cause Lynch syndrome⁶⁴, and complete knockout of paralog Msh2 in Ccm1^{+/-} mice causes multiple cavernoma through a presumed second hit⁶⁵. We found that carriers had a nonsignificant greater frequency of neoplasms (OR=2.35, unadjusted P=2.49x10⁻², minP-adjusted P=1.00) and cerebral cavernomata (OR=5.23, unadjusted P=6.58x10⁻⁴, minP-adjusted P=0.157) than non-carriers. Carriers of the 1p36.33 [2.02-2.49Mb] deletion overlapping the gene SKI had features (hypotonia, talipes equinovarus, abnormalities of the globe and nose, osteoporosis, global developmental delay, and Chiari malformation) concordant to the Shprintzen-Goldberg craniosynostosis syndrome caused by SKI⁴¹. All 15 individuals with duplication of 9q34.3 had focal-onset seizures that were rarely drug-resistant, without any individual annotated with a neurodevelopmental disorder or polymicrogyria despite the presence of the GRIN1, which can cause polymicrogyria when affected by gain-of-function variants⁴⁷. Sixteen of 24 individuals carrying deletions at 15q13.3 [31.06-32.51Mb] had generalized absence seizures (OR=10.5, unadjusted P=3.70x10⁻⁸, minP-adjusted P=1x10⁻⁵), in line with the primary seizure type reported in carriers of the 15q13.3 deletion⁶⁶. Finding generalized myoclonic seizures in half of the carriers of the 22q11.2 [19.67-19.96Mb] deletion further confirmed TBX1⁶⁷, the known causal gene for the 22q11.21 deletion/DiGeorge syndrome⁴⁸. Features suggestive of juvenile myoclonic epilepsy were also found among six people carrying deletions overlapping with the second credible interval at 22q11.2 [20.65-21.54Mb] spanning the Noonan syndrome 10 locus containing in which a single individual was reported with seizures⁴⁹. However, none of these six individuals had annotations beyond seizures and electroencephalography phenotypes that would support a multisystemic syndrome.

Finally, clinicians may want to know the frequency of broad clinical features among carriers of the CNV identified in their patients to improve the interpretation of its clinical relevance and to facilitate genetically stratified prognostication. Therefore, we prioritized 17 common, conceptually broad, and important epilepsy manifestations and comorbidities for visualization, including the co-occurrence of generalized-onset and focal-onset seizures that characterizes the combined generalized and focal epilepsy type⁶² (**Fig 3 and Supplementary Fig. 3A-E**). The most common CNV, deletion at 2p21-p16.3, appeared to modestly increase the likelihood of a carrier having generalized epilepsy. However, a few CNVs had a profile dominated by core electroclinical features of generalized (for example, deletions at 15q13.2-15q13.3) or focal epilepsy (duplications at 9q34.3 [139.89-140.12Mb]), with comorbid features being rare. Conversely, carriers of other CNVs had relatively high frequencies of neurodevelopmental disorders, epileptic spasms, and drug resistance suggestive of developmental and epileptic encephalopathy (deletions at 1p36.33). However, no CNV was found exclusively in people with a particular seizure type, and carriers of some CNVs appeared to have broad clinical features at

frequencies indistinguishable from the cohort's baseline (duplications at 19p13.3), suggesting some generic contribution to epilepsy risk across epilepsy types."

Updated **Fig. 2** shows the proportion of carriers and non-carriers annotated with each HPO concept for the 19 significant CNV-HPO associations in the Phenome cohort.

Fig. 2: Genotype-first phenomic analysis in 10,880 individuals with detailed clinical data. For each CNV, the proportion of carriers and non-carriers annotated with each HPO concept is plotted. Those above the diagonal were enriched among carriers, and those below were depleted. *Selected phenotypic concepts are labeled.* Full results for all associations reaching raw $p < 0.05$ are provided in **Supplementary Table 6**. SUDEP = sudden unexpected death in epilepsy, CNS = central nervous system, EEG = electroencephalogram.

Updated **Fig. 3** shows the clinical signatures of all CNVs with significant CNV-HPO associations in the Phenome cohort.

Fig. 3: Summary of clinical signatures of CNVs in a deeply phenotyped epilepsy cohort. Dots represent the prior frequency of each broad clinical manifestation in the Phenomic cohort. Binomial distribution-derived 95% confidence interval bars are given for the frequency of each phenotype among carriers of the CNV. "Craniofacial or skeletal dysmorphism" includes individuals with either "Abnormality of the head [HP:0000234]" (which excludes isolated brain structural abnormalities) or "Abnormal skeletal morphology [HP:0011842]". "Motor, movement or muscular disorder" includes individuals with any of "Abnormal central motor function [HP:0011442]", "Abnormality of movement [HP:0100022]" or "Abnormality of the musculature [HP:0003011]", but not "Motor delay [HP:0001270]", which is included in "Neurodevelopmental abnormality". While "Neurodevelopmental abnormality" includes those with "Intellectual disability", the latter is shown in addition as it is a neurodevelopmental outcome with particularly important socioeconomically important consequences. EEG = electroencephalogram. Further CNV profiles are shown in **Supplementary Fig. 3A-E**.

Updated **Table 3** shows the 19 significant CNV-HPO associations in the Phenome cohort.

Table 3: Significant individual CNV-HPO associations.

Locus	CNV	HPO	Odds ratio [95% CI]	Relative risk	P -value		CNV carriers			CNV non-carriers	
					Raw	Adjusted	Prop	N _{pheno}	N _{tot}	N _{pheno}	N _{tot}
15q13.2-q13.3 [31.06-32.51Mb]	DEL	Generalized non-motor (absence) seizure [HP:0002121]	10.5 [4.25-28.5]	4.18	3.70E-08	1.00E-05	0.667	16	24	1731	10856
15q13.2-q13.3 [31.06-32.51Mb]	DEL	Typical absence seizure [HP:0011147]	8.43 [3.48-21.3]	4.1	6.94E-07	1.10E-04	0.583	14	24	1545	10856
15q13.2-q13.3 [31.06-32.51Mb]	DEL	EEG with spike-wave complexes [HP:0010850]	7.84 [3.16-21.2]	3.28	1.18E-06	2.00E-04	0.667	16	24	2205	10856
15q13.2-q13.3 [31.06-32.51Mb]	DEL	Generalized-onset seizure [HP:0002197]	9.41 [3.15-37.9]	2.4	1.41E-06	2.20E-04	0.833	20	24	3766	10856
15q13.2-q13.3 [31.06-32.51Mb]	DEL	EEG with generalized epileptiform discharges [HP:0011198]	6.76 [2.44-23.2]	2.2	1.98E-05	0.00379	0.792	19	24	3905	10856
15q13.2-q13.3 [31.06-32.51Mb]	DEL	Bilateral tonic-clonic seizure with focal onset [HP:0007334]	0 [0-0.404]	0	4.07E-04	0.0484	0	0	24	3168	10856
1p36.33 [0.91-1.51Mb]	DEL	Hypotonia [HP:0001252]	12.2 [3.95-32]	9.51	3.23E-05	0.00674	0.24	6	25	274	10855
1p36.33 [0.91-1.51Mb]	DEL	Epileptic spasm [HP:0011097]	7.47 [2.78-18.4]	5.4	6.85E-05	0.0108	0.32	8	25	643	10855
1p36.33 [0.91-1.51Mb]	DEL	Abnormal muscle tone [HP:0003808]	8.65 [2.81-22.7]	6.82	1.97E-04	0.0287	0.24	6	25	382	10855
1p36.33 [0.91-1.51Mb]	DEL	Infantile spasms [HP:0012469]	8.34 [2.71-21.9]	6.58	2.39E-04	0.0324	0.24	6	25	396	10855
1p36.33 [0.91-1.51Mb]	DEL	Abnormal muscle physiology [HP:0011804]	8.21 [2.67-21.5]	6.48	2.59E-04	0.0339	0.24	6	25	402	10855
1p36.33 [0.91-1.51Mb]	DEL	Abnormality of the musculature [HP:0003011]	8.04 [2.61-21.1]	6.35	2.87E-04	0.038	0.24	6	25	410	10855
1p36.33 [0.91-1.51Mb]	DEL	Plagiocephaly [HP:0001357]	93.8 [9.48-482]	86.8	3.30E-04	0.045	0.08	2	25	10	10855
2p21-p16.3 [47.50-47.85Mb]	DEL	Focal-onset seizure [HP:0007359]	0.463 [0.313-0.681]	0.708	4.79E-05	0.0086	0.456	52	114	6939	10766
2p21-p16.3 [47.50-47.85Mb]	DEL	Bilateral tonic-clonic seizure with generalized onset [HP:0025190]	2.3 [1.5-3.46]	1.88	9.09E-05	0.0157	0.325	37	114	1861	10766
15q12-q13.1 [27.93-28.23Mb]	DEL	Global developmental delay [HP:0001263]	69.1 [5.55-3540]	18.1	2.80E-04	0.0127	0.75	3	4	451	10876
15q12-q13.1 [27.93-28.23Mb]	DEL	Epileptic encephalopathy [HP:0200134]	Inf [4.43-Inf]	7.72	2.83E-04	0.0127	1	4	4	1408	10876
15q12-q13.1 [27.93-28.23Mb]	DEL	Encephalopathy [HP:0001298]	Inf [4.41-Inf]	7.69	2.87E-04	0.0129	1	4	4	1414	10876
16p11.2 [29.87-30.19Mb]	DUP	Psychogenic non-epileptic seizure [HP:0033052]	81.5 [7.85-471]	61.8	4.82E-04	0.0297	0.25	2	8	44	10872

The first column reports the genomic band and coordinates of the considered CNV. The CNV type is reported in column 2. In column 3, the HPO term name and identifier are reported. In column 4, the odds ratio with raw/uncorrected 95% confidence interval is reported. In column 5, the relative risk is given to aid

interpretation. In column 6, the uncorrected P -values are reported. In column 7, the minP step-down P -value is given, which provides an adjustment for all 1,667 HPO term associations tested within each CNV group while accounting for the correlation between harmonized HPO annotations (see Online Methods). In column 8, the proportion of CNV carriers annotated with the phenotype is given. In columns 9-10 and 11-12, N_{pheno} and N_{tot} are the numbers of individuals presenting with the phenotype and the total number of individuals carrying and not-carrying the CNV, respectively.

We have restructured the previous first paragraph of the **Discussion** to reflect on the new fine-mapping results, the overlap between seizure-associated CNVs and previously reported clinical phenotypes, and the CNV-HPO analysis in the Phenome cohort.

Page 11, **Discussion**, the first paragraph was removed and replaced with: “In this study, we leveraged a substantial increase in sample size to identify novel seizure-associated CNVs when jointly analyzing 26,699 individuals with various types of seizure disorders against 492,324 population controls. We identified 25 novel loci with genome-wide significance for seizure disorders. *In addition, all three previously reported epilepsy-associated loci at genome-wide level maintained genome-wide significance for seizure disorders in our meta-analysis that included the epilepsy cohort from the previous study²⁰. Of the 25 seizure-associated loci, 16 were previously implicated in neurological and psychiatric disorders, including epilepsy. Five were flanked by known segmental duplications (SDs) or low copy number repeats (LCRs). Of note, our fine-mapping analysis confirmed the first and third known critical regions for seizures within the phenotype spectrum of the 1p36 deletion syndrome³⁸, TBX1 as the (known) causal gene for the 22q11.21 deletion/DiGeorge syndrome⁴⁸, and suggested the SNRPN promoter/exon 1 region as the causal element for seizures within the larger BP2-BP3 15q11.2-q13 duplication region.*

In a high-resolution phenomic analysis in a subset of 10,880 individuals from our cohort with epilepsy (from the Epi25 cohort), we identified 622 suggestive and 19 significant clinical associations informative for epileptologists among CNV carriers. This observation indicates that beyond contributing to the generic risk of seizures, several CNVs contribute to specific epilepsy types. Carriers of some CNVs tended to have features typical of developmental and epileptic encephalopathies with neurodevelopmental and non-seizure phenotypes. Conversely, carriers of others had phenotypes restricted to the core epileptic features of seizures and electroencephalographic abnormalities (both generalized and focal). Interestingly, reciprocal CNVs involving 22q11.21 seemed to produce opposite epilepsy types, with deletion and duplication carriers tending to have generalized and focal epilepsies, respectively. Dose-dependent effects of KLHL22 on DEPDC5 degradation are a possible explanation⁶⁸. Overall, the high degree of pleiotropy among seizure-associated CNVs implies that these CNVs likely impair neurodevelopmental processes rather generically and contribute to the broad spectrum of neurodevelopmental disorders. According to the oligo-/polygenic inheritance model, CNVs may interact with the genetic background or environmental factors to generate the final disease phenotype. Interaction between CNVs and the polygenic background was recently demonstrated in carriers of the schizophrenia-associated 22q11.2 deletion⁶⁹. Support for an oligogenic-CNV disorder model was also recently published⁷⁰.”

Major concerns

- there should be a column in table 1 or early supplementary tables reporting the number of carriers & meta-analysis statistics, replication info, etc. can be included in this also

Response 3.3: We added a new **Supplementary Table 7** detailing: 1) the frequency of case/control CNVs in the meta-analysis cohort (Epi25 and Neuropsychiatric disorders cohorts), 2) the frequency of CNVs with ≥50% overlap in the DGV Gold Standard and DECIPHER Population frequencies datasets.

We added **Supplementary Table 7:** CNV frequencies in the cases & controls of the meta-analysis, DGV Gold Standard, and DECIPHER databases.

Cytoband	CNV type	Hg19 Start (Mb)	Hg19 End (Mb)	Cases carrier frequency [%]	Control carrier frequency [%]	Frequency in DGV Gold Standard [%]	Frequency in DECIPHER [%]
----------	----------	-----------------	---------------	-----------------------------	-------------------------------	------------------------------------	---------------------------

1p36.33	DEL	0.91	1.51	0.086	0.015	0	0
1p36.33	DEL	2.02	2.49	0.146	0.003	0	0
1q44	DEL	245.29	245.86	0.041	0.002	0	0
2p21-p16.3	DEL	47.5	47.85	0.678	0.002	0	0
2q13	DUP	110.77	111.06	0.139	0.108	0	0.27
3q29	DEL	195.76	196.24	0.034	0.002	0	0.12
8p23.3-p23.2	DEL	0.4	5.47	0.067	0.010	0	0
9p24.3	DEL	0.33	0.56	0.049	0.007	0	0.02
9q34.3	DUP	139.21	140.12	0.315	0.003	0	0
10q26.3	DEL	133.41	134.68	0.030	0.002	0	0
15q11.2	DEL	22.74	23.28	0.689	0.284	0.41	0.83
15q11.2-q13.3	DUP	22.98	32.15	0.258	0.005	0	0.12
15q12-q13.1	DEL	27.93	28.23	0.097	0.008	0	0
15q13.2-q13.3	DEL	31.06	32.51	0.243	0.012	0	0.02
16p13.3	DUP	0.6	0.89	0.573	0.009	0	0
16p13.11	DEL	15.42	16.35	0.363	0.031	0.03	0.07
16p12.2	DEL	21.88	22.5	0.191	0.054	0.09	0.08
16p11.2	DEL	29.56	30.19	0.165	0.024	0.05	0.12
16p11.2	DUP	29.87	30.19	0.127	0.026	0	0.03
17q12	DUP	34.76	36.25	0.187	0.014	0.02	0.12
17q21.31	DEL	41.08	41.45	0.461	0.004	0	0
19p13.3	DUP	1.04	1.34	0.427	0.003	0	0.08
20q13.33	DUP	62	62.35	0.479	0.006	0	0
22q11.21	DUP	18.99	21.54	0.199	0.067	0	0
22q11.21	DEL	18.99	21.54	0.120	0.009	0	0.19

The frequencies in the DGV Gold Standard and DECIPHER Population databases are given for CNVs with $\geq 50\%$ overlap with the seizure-associated CNV regions.

- the title refers to seizure-associated copy number variants, but this seems to overlook the fact that almost all loci identified here (and similarly almost all patients) likely have comorbid developmental delay. This should somehow be apparent in the title. These are large pleiotropic CNVs, not precise genetic hits.

Response 3.4: We think our manuscript's title is appropriate because “seizures” was the only clinical symptom present in 100% of all cases of our GWAS meta-analysis cohort. Therefore, the identified CNVs are, in the first place, associated with seizures. We agree with the reviewer that we cannot exclude that some of the identified loci are associated with other clinical phenotypes present in a high percentage of all cases. For transparency, we included a corresponding statement in the limitation section of the **Discussion**. Also, we expanded and reworded the **Results** section “**Discovery of 25 genome-wide significant seizure-associated CNV regions**” to improve the clarity of our observation that most seizure-associated CNV regions are highly pleiotropic and often reported as pathogenic in other disorders (10/25 in neurological or psychiatric disorders excluding epilepsy). Previously reported disease phenotypes for each seizure-associated region are detailed in the new **Table 2**. However, our CNV-HPO PheWAS analysis provides evidence that the identified 25 seizure-associated loci are not driven by a random distribution of neurological and psychiatric phenotypes across the meta-analysis cohort. Using our expert-curated data set from Epi25 and >15,000 symptom-level HPO terms, we discovered clear genotype-phenotype associations for several CNVs that only became apparent when using the standardized HPO terms. Thus, phenotypic homogeneity may become only apparent once high-resolution data beyond crude disease labels are available for even larger cohorts.

Please see the rephrased Results section “**Discovery of 25 genome-wide significant seizure-associated CNV regions**” and **Table 2** in **Response 3.2**.

Page 13, **Discussion**, the beginning of paragraph four, we added: “Our study has several limitations. *First, many of the patients with seizures included in this study have comorbid neurological and psychiatric disorders. Therefore, some of the identified CNV loci may be associated with other clinical phenotypes present in a high percentage of all cases.*”

- *the authors use a fine-mapping technique to elucidate the credible sets underlying each locus in Collins et al., 2022. They claim this is not within the scope of this work. I disagree, I think this work's impact for a research audience would be significantly enhanced by having an idea of how confidently the data pinpoint specific genes within these CNV loci.*

Response 3.5: As requested by the referee, we have added a two-stage fine-mapping analysis to our study. First, we used a Bayesian algorithm (PMID: 18642345) to identify the minimal credible interval(s) that contained the causal element(s) with 95% confidence, as in Collins et al. (2022) (PMID: 35917817). The list of seizure-associated CNV regions in **Table 1** was updated to represent the fine-mapping analysis, leading to revised start/stop of the associated regions and an updated number of 25 genome-wide significantly associated CNV regions with 33 credible intervals for the causal elements. Second, we explored each candidate gene's known biological function, performed pathway analyses, and provided, where possible, the biological context of how each gene may be involved in the causation of seizures in the new **Results** section “**Fine-mapping and candidate genes**”. The HPO enrichment / PheWAS analysis (**Results** section: “**Characterization of the clinical subphenotypes enriched in the carriers of each seizure-associated CNV in epilepsy patients with deep phenotypes**”) was accordingly updated to correspond to the 33 identified credible intervals. We also amended the corresponding sections of the **Methods** and the **Discussion**.

Please see the updated Methods section “**Meta-analysis and fine-mapping**” in **Response 3.2**.

Please see the new Results section “**Fine-mapping and candidate genes**”, the updated Results section “**Characterization of the clinical subphenotypes enriched in the carriers of each seizure-associated CNV in epilepsy patients with deep phenotypes**”, **Tables 1 to 3** in **Response 3.2**.

Please see the first modified first two paragraphs of the **Discussion** in **Response 3.2**.

- *to the point above, the pathway analyses could be improved by fine-mapping the CNV hits first.*

Response 3.6: Many thanks for the suggestion. Using the results of the updated fine-mapping, we performed pathway analyses that generally point to heterogeneity among the pathways affected by the 25 identified seizure-associated regions. We amended the corresponding sections of the **Methods** and **Results**.

The results of the pathway analysis using the fine-mapped intervals containing the causal elements/genes with 95% confidence are given in the last paragraph of the new Results section “**Fine-mapping and candidate genes**”: “*Significantly enriched Gene ontology (GO) Biological Processes among all known brain-related disease genes in the credible intervals were: chordate embryonic*”

development (GO:0043009), sensory organ morphogenesis (GO:0090596), mitotic G2 DNA damage checkpoint signaling (GO:0007095), neural tube closure (GO:0001843), negative regulation of Ras protein signal transduction (GO:0046580), dendrite morphogenesis (GO:0048813), and mitotic G2/M transition checkpoint (GO:0044818). No GO Biological Process was significantly enriched when considering all genes inside all credible intervals, pointing to likely heterogeneous disease mechanisms of the 25 seizure-associated CNV regions. All credible intervals and known brain-related disease genes are detailed in **Table 2**, additional candidate genes of lower confidence are detailed in **Supplementary Table 2**, and all genes inside the credible intervals are detailed in **Supplementary Table 3**.”

Please see the full new Results section “**Fine-mapping and candidate genes**” in **Response 3.2**.

- the impact for a clinical audience could be greatly enhanced by having a summary table that describes resources for each association, e.g. at OMIM, GeneReviews, or DECIPHER wherever such associations exist

Response 3.7: We added the new **Supplementary Table 8** that describes all resources we used to investigate the seizure-associated CNV regions and refer to it in the revised **Methods** section “**Meta-analysis and fine-mapping**”.

A new **Supplementary Table 8** was added detailing all resources used to collect information about each CNV and the affected genes:

Name	Web address	Used for
DGV	http://dgv.tcag.ca/dgv/app/downloads?ref=	Annotation of CNV frequencies from external data
DECIPHER	https://www.deciphergenomics.org/about/downloads/data	Localization of reported phenotypes, curated genes, dosage sensitivity
	https://www.deciphergenomics.org	Reported CNV Syndromes
ClinGen	https://www.clinicalgenome.org	Localization of reported phenotypes, curated genes, dosage sensitivity
GARD	https://rarediseases.info.nih.gov	Detailed phenotypes of disorders caused by CNVs or genes overlapping with the seizure-associated regions
OMIM	https://www.omim.org	Molecular Genetics and reported patients for any gene
PubMed	https://pubmed.ncbi.nlm.nih.gov	Screening of published literature
UCSC Genome Browser on Human (GRCh37/hg19)	https://genome.ucsc.edu/	Visualization of the credible intervals
GWAS Catalog	https://www.ebi.ac.uk/gwas/home	Reported GWAS hits in credible intervals
Enrichr	https://maayanlab.cloud/Enrichr/	Pathway analyses

- to the point above, such a table would make it clear which (if any) loci are truly novel in association to epilepsy and/or novel over all

Response 3.8: We agree with the reviewer and added a new **Table 2** that details previously reported disease phenotypes for each seizure-associated region (the former Table 2 is now Table 3). The reader can now identify which of the 25 seizure-associated CNV loci have been: 1.) specifically implicated in epilepsy with and without formal association statistics (9/25), 2.) implicated in neurological or psychiatric disorders where seizures do not occur in the majority of individuals (10/25), 3.) implicated in other disorders (2/25), and 4.) never reported (4/25). We also updated the **Results** section “**Discovery of 25 genome-wide significant seizure-associated CNV regions**” accordingly.

Please see the rephrased Results section “**Discovery of 25 genome-wide significant seizure-associated CNV regions**” in **Response 3.2**.

Please see the new **Table 2** detailing reported disease phenotypes for each seizure-associated region in **Response 3.2**.

- how do these CNV results compare with the ILAE common variant study published on medRxiv with 29k cases and 26 loci identified? Broad overlaps with these loci should be acknowledged, if any.

Response 3.9: Many thanks for this suggestion. There was no overlap between the ILAE SNP-based GWAS loci for epilepsy as published on medRxiv and the seizure-associated CNV regions in our study. We have added a brief comment on this observation to the rephrased **Results** section “**Discovery of 25 genome-wide significant seizure-associated CNV regions**”.

Please see the rephrased Results section “**Discovery of 25 genome-wide significant seizure-associated CNV regions**” in **Response 3.2**.

Minor concerns

- related to the lack of integration & context above, here is an example:3q29 is annotated as having no other phenotypic associations in Table 1. DECIPHER shows a microduplication and microdeletion syndrome

Response 3.10: Many thanks for the example. CNVs at 3q29 are known to cause a wide spectrum of disorder phenotypes. In Table 1, we reported only the CNVs which were genome-wide significantly associated with seizures in our study. For each of these CNVs, we stated in the column “Highest odds ratio in neuropsychiatric disorder/seizure meta-analyses (95% CI)” the observed effect size when the epilepsy/seizure phenotype was meta-analyzed with other neurological and psychiatric phenotypes from Collins *et al.* (2022) to help assess the level of pleiotropy. The 3q29 deletion was indeed not found associated with any of the 23 tested neurological and psychiatric phenotypes. However, a 3q29 deletion syndrome was previously reported. We have updated the pleiotropy analysis results in **Table 1** and improved the clarity of the table. The new **Table 2** details all previously reported disease phenotypes for each seizure-associated region, including the 3q29 deletion syndrome overlapping with the identified seizure-associated region.

Please see the updated **Table 1** and the new **Table 2** detailing reported disease phenotypes for each seizure-associated region in **Response 3.2**.

- in table 1, N is easy to misinterpret as number of carriers

Response 3.11: We changed the name of column “N” in **Table 1** to “*Association with neuropsychiatric phenotypes [N]*”.

Please see the updated **Table 1** in **Response 3.2**.

- the 15q region is quite complex, and yields a range of known syndromes. One of them is duplication 15q syndrome, which can involve an interstitial (typically 3 copies) or isodicentric (typically 4+ copies) duplication. Can the authors say anything about loci like this beyond “there is a copy gain”? If not, they ought to clarify they mean copy loss and copy gain instead of duplication/deletion throughout the manuscript.

Response 3.12: The SNP array-based technology used in this study does not provide high-confidence exact copy number estimation beyond 1 copy (deletions) or 3 copies (duplications). In the first Results section, we included a corresponding definition for deletions/ duplications. Deconvoluting the CNV association signal at these complex regions has many challenges due to the genomic complexity of these loci that can have population-specific (<https://doi.org/10.1101/2022.07.09.499321>) or even person-specific configurations that even classical whole-genome sequencing technology cannot resolve (PMID: 35357919). Long-range whole-genome sequencing paired with improved pangenome references will be needed to fully resolve complex rearrangements such as the 15q region. We added a corresponding statement to the limitations section of the **Discussion**.

In Results, page 5, section “**Discovery of 25 genome-wide significant seizure-associated CNVs**” we added “*All the genome-wide associated deletions found in this study consisted in the loss of one copy, while all duplications consisted of the gain of one copy.*”

Please see the full Results section “**Discovery of 25 genome-wide significant seizure-associated CNV regions**” in **Response 3.2**.

Page 14, Limitation section of the **Discussion**, we added: “*Among the 25 identified CNVs, deletions ranged from 230kb to 5Mb and duplications from 290kb to 9Mb, affecting 14.2 genes on average.* CNV breakpoints in the current study are estimated from genotyped SNPs around the actual breakpoint. These breakpoint estimates are limited by the resolution of the genotyping platform used to call the CNVs. In fact, microarrays have many technical limitations, such as poor breakpoint resolution and limited sensitivity for small CNVs⁸¹. Newer technologies like whole-genome sequencing (WGS) will enable the assessment of a more comprehensive array of rare variants, including balanced rearrangements, small (exonic) CNVs⁸², short tandem repeats, and other structural variants⁸³. *However, some genomic regions harbor complex deletion/duplication/inversion rearrangements (e.g., 22q11.21⁸⁴, 15q11.2⁸⁵) that can even show population stratification (e.g., 16p11.2⁸⁶). More accurate and complete (pangenome) references will be needed to determine the exact breakpoints of such complex rearrangements^{87,88}, even in the case of sequencing-based CNVs discovery.* Lastly, we

performed joint epilepsy/seizures and cross-disorder meta-analyses in individuals with minimal clinical information.”

- are the 3 loci replicated from prior replicated in independent data? It wasn't clear from the main text.

Response 3.13: Yes, we replicated the previous findings. However, the three loci were previously found as genome-wide associated in our previous study (Niestroj et al., 2020, PMID: 32568404), which used a subset of the individuals with epilepsy from this study (40% of all cases, 66% of the individuals with epilepsy, N=10,712). Thus technically, the association of the three loci with seizures represents a phenotype expansion from epilepsy to the broader phenotype of seizure disorders. We amended the corresponding parts of the **Results** and **Discussion** to improve clarity.

Please see the rephrased Results section “**Discovery of 25 genome-wide significant seizure-associated CNV regions**” in **Response 3.2**.

Page 12, **Discussion**, paragraph 1, line 4, was changed from: “In addition, our seizure disorders meta-analysis replicated all three CNVs previously reported as associated with epilepsy at genome-wide level² even if the seizure cohort of this study has a more coarse-grained phenotype definition.”

Changed to: “*In addition, all three previously reported epilepsy-associated loci at genome-wide level maintained genome-wide significance for seizure disorders in our meta-analysis that included the epilepsy cohort from the previous study²⁰.*”

Overall, I think the authors need to place the results in a better context both for biologists and for clinicians. They should also highlight what, if anything, is novel here compared to previous epilepsy and neurodevelopmental genetic findings.

Response 3.14: We agree with the reviewer that a better biological context and highlighting of the novelty of our findings are necessary. While also addressing previous comments of reviewer #2 (Responses 2.5, 2.6, 2.8) and we rephrased the corresponding **Results** section “**Discovery of 25 genome-wide significant seizure-associated CNV regions**”, modified **Table 1** to incorporate fine-mapping analysis, and added a new **Results** section “**Fine-mapping and candidate genes**” in which we explore candidate genes. The new **Results** section is supported by a new **Table 2**, **Supplementary Fig. 1**, and **Supplementary Tables 2 and 3**. We also integrated a summary into the **Discussion**.

Please see the rephrased Results section “**Discovery of 25 genome-wide significant seizure-associated CNV regions**” and the updated **Table 1** in **Response 3.2**.

Please see the new Results section “**Fine-mapping and candidate genes**”, the new **Table 2**, **Supplementary Fig. 1**, **Supplementary Tables 2 and 3**, and an addition to the beginning of the **Discussion** in **Response 3.2**.

REVIEWERS' COMMENTS

Reviewer #1 (Remarks to the Author):

I thank the authors for addressing my concerns. I have no further comments.

Reviewer #2 (Remarks to the Author):

The paper presents in a much stronger way now. I would still like to see the Abstract be a bit more informative (very general now) including the most significant loci findings listed. This will help it get more pick-ups in keyword searches. Also for the title...you may want to use copy number variations (variations is the original term...it later became variants)...into the future with so much scrutiny on wording in phenomics/genomics "variations" may be a more mindful presentation (at least for the title). I did not have time to check on eventual data availability but I assume this is in place. This is a very important data resource for the community. Great work.

Reviewer #3 (Remarks to the Author):

This manuscript is greatly improved due to: 1) better integration of prior findings, 2) a clear message that 4/25 CNV-affected regions are novel across all phenotype while many other regions are novel for the seizure association, and 3) fine-mapping results that suggests causal regions and genes and enable future molecular studies. I do have two further questions related to the fine-mapping findings.

1) regarding the SNRPN result for the 15q11.2-13.2 region, there is a paternal (I think usually interstitial) and maternal form of the duplication (often isodicentric with more than 4 copies). Maternal vs paternal imprinting may affect which gene might be causal. For the maternal form, many implicate UBE3A (PMC3356696), while for the paternal NECDIN has been implicated (PMC8249516). All of this is with mouse models, so it would be great to better clarify the causal gene, and whether the causal gene might differ for maternal vs paternal duplication in this region, with human genetics. Within their diagnostic lab results, do the authors have enough information to stratify maternal vs paternal and fine-map them separately to see if there are different genes nominated for this region?

2) regarding the BRCA1 finding, is there a chance that the seizures are secondary to metastatic breast cancer in these individuals? Can the authors use their extensive phenotype data to resolve this?

Overall, I think this is a really improved manuscript that will be valuable to both molecular biologists and clinical geneticists.

REVIEWERS' COMMENTS

Reviewer #1 (Remarks to the Author):

I thank the authors for addressing my concerns. I have no further comments.

We thank the reviewer for the time devoted to the critical review of our manuscript, which allowed us to improve it.

Reviewer #2 (Remarks to the Author):

The paper presents in a much stronger way now. I would still like to see the Abstract be a bit more informative (very general now) including the most significant loci findings listed. This will help it get more pick-ups in keyword searches.

We modified the abstract to improve the presentation of the most important findings.

“Copy number variants (CNV) are established risk factors for neurodevelopmental disorders with seizures or epilepsy. With the hypothesis that seizure disorders share genetic risk factors, we pooled CNV data from 10,590 individuals with seizure disorders, 16,109 individuals with clinically validated epilepsy, and 492,324 population controls and identified 25 genome-wide significant loci, 22 of which are novel for seizure disorders, such as deletions at 1p36.33, 1q44, 2p21-p16.3, 3q29, 8p23.3-p23.2, 9p24.3, 10q26.3, 15q11.2, 15q12-q13.1, 16p12.2, 17q21.31, duplications at 2q13, 9q34.3, 16p13.3, 17q12, 19p13.3, 20q13.33, and reciprocal CNVs at 16p11.2, and 22q11.21. Using genetic data from additional 248,751 individuals with 23 neuropsychiatric phenotypes, we explored the pleiotropy of these 25 loci. Finally, in a subset of individuals with epilepsy and detailed clinical data available, we performed phenome-wide association analyses between individual CNVs and clinical annotations categorized through the Human Phenotype Ontology (HPO). For six CNVs, we identified 19 significant associations with specific HPO terms and generated, for all CNVs, phenotype signatures across 17 clinical categories relevant for epileptologists. This is the most comprehensive investigation of CNVs in epilepsy and related seizure disorders, with potential implications for clinical practice.”

Also for the title...you may want to use copy number variations (variations is the original term...it later became variants)...into the future with so much scrutiny on wording in phenomics/genomics "variations" may be a more mindful presentation (at least for the title).

We agree with the reviewer and changed the title accordingly to: “Genome-wide identification and phenotypic characterization of seizure-associated copy number variations in 741,075 individuals”

I did not have time to check on eventual data availability but I assume this is in place. This is a very important data resource for the community. Great work.

Our data availability statement is as follows:

“All genome-wide CNV association summary statistics are available at Zenodo (<https://zenodo.org/record/7939126#.ZGK7yi-B29Y> with DOI 10.5281/zenodo.7939126). Individual-level CNV data for epilepsy patients are available from the Epi25 Consortium (<http://epi-25.org/>) upon reasonable request and approved ethics protocol. Furthermore, raw data is deposited at dbGAP https://www.ncbi.nlm.nih.gov/projects/gap/cgi-bin/study.cgi?study_id=phs001551.v1.p1. All HPO-based phenome-wide summary statistics are available in Supplementary Table 6 of this manuscript. Fine-mapping results are available in Table 2 and Supplementary Tables 2-3 of this manuscript. The CNV data of the Neuropsychiatric cohort are described in the Supplementary Materials of Collins et al. (2022)⁹⁷. They can be accessed from existing publications, public resources, or, upon request, from the authors of Collins et al. (2022)⁹⁷ (see “Key resources table” and Table S2 in Collins et al.⁹⁷). The CNV data reported by GeneDx and Indiana University clinical testing sites were not consented for public release. All datasets used in this study are detailed in Supplementary Table 1 of our manuscript.”

Reviewer #3 (Remarks to the Author):

This manuscript is greatly improved due to: 1) better integration of prior findings, 2) a clear message that 4/25 CNV-affected regions are novel across all phenotype while many other regions are novel for the seizure association, and 3) fine-mapping results that suggests causal regions and genes and enable future molecular studies. I do have two further questions related to the fine-mapping findings.

1) regarding the *SNRPN* result for the 15q11.2-13.2 region, there is a paternal (I think usually interstitial) and maternal form of the duplication (often isodicentric with more than 4 copies). Maternal vs paternal imprinting may affect which gene might be causal. For the maternal form, many implicate *UBE3A* (PMC3356696), while for the paternal *NECDIN* has been implicated (PMC8249516). All of this is with mouse models, so it would be great to better clarify the causal gene, and whether the causal gene might differ for maternal vs paternal duplication in this region, with human genetics. Within their diagnostic lab results, do the authors have enough information to stratify maternal vs paternal and fine-map them separately to see if there are different genes nominated for this region?

We only had access to patient and not to parental DNA. Our genotyping data does not allow stratifying maternal versus paternal copy number change. We were, therefore, unable to assess whether the duplicated region is of maternal or paternal origin. Without stratifying between maternal vs. paternal duplications, our fine-mapping analysis identifies *SNRPN* as the only candidate gene of the BP2-BP3 15q11.2-q13 duplication. We have added to the discussion that the imprinting status of the duplicated region was unknown in this study.

Page 12, Discussion, end of the first paragraph, we added: “Of note, our fine-mapping analysis confirmed ... and suggested the *SNRPN* promoter/exon 1 region as the causal element for seizures within the larger BP2-BP3 15q11.2-q13 duplication region. However, our study design did not support the assessment of whether the imprinting status of the duplicated region itself plays an additional role besides the previously suggested role of *SNRPN* promoter/exon 1 region in regulating the imprinting of the Prader-Willi critical region. Future studies that also include genomic screens of parents will shed light on this open question.”

2) regarding the *BRCA1* finding, is there a chance that the seizures are secondary to metastatic breast cancer in these individuals? Can the authors use their extensive phenotype data to resolve this?

Excellent point, and we investigated this possibility with our newly developed CNV-HPO PheWAS framework. We only found nonsignificant greater frequencies of six tumor-related HPO terms (Cavernous hemangioma, Hemangioma, Vascular neoplasm, Pituitary adenoma, Neoplasm of the anterior pituitary, Neoplasm of the pituitary gland) in carriers of the 17q21.31 deletion. This information is included in Supplementary Table 6. Given that this result is only suggestive and we have many more results, we did not mention this observation in the discussion.

Overall, I think this is a really improved manuscript that will be valuable to both molecular biologists and clinical geneticists.

We thank the reviewer for the time devoted to reviewing our manuscript critically.